# Phosphatidylinositol 4,5-bisphosphate optical uncaging potentiates exocytosis

Alexander M Walter[1,2†]*, Rainer Müller[3†], Bassam Tawfik[1†], Keimpe DB Wierda[1], Paulo S Pinheiro[1], André Nadler[3,4], Anthony W McCarthy[2], Iwona Ziomkiewicz[1,5], Martin Kruse[6], Gregor Reither[3], Jens Rettig[7], Martin Lehmann[2], Volker Haucke[2], Bertil Hille[6], Carsten Schultz[3]*, Jakob Balslev Sørensen[1]*

[1]Neurosecretion group, Center for Neuroscience, Faculty of Health and Medical Sciences, University of Copenhagen, Copenhagen, Denmark; [2]Leibniz-Forschungsinstitut für Molekulare Pharmakologie, Berlin, Germany; [3]Cell Biology and Biophysics Unit, European Molecular Biology Laboratory, Heidelberg, Germany; [4]Max Planck Institute of Molecular Cell Biology and Genetics, Dresden, Germany; [5]Discovery Sciences, AstraZeneca, Cambridge, United Kingdom; [6]Department of Physiology and Biophysics, School of Medicine, University of Washington, Seattle, United States; [7]Cellular Neurophysiology, Center for Integrative Physiology and Molecular Medicine, Saarland University, Homburg, Germany

*For correspondence:
awalter@fmp-berlin.de (AMW);
schultz@embl.de (CS);
jakobbs@sund.ku.dk (JBSø)

†These authors contributed equally to this work

**Abstract** Phosphatidylinositol-4,5-bisphosphate [PI(4,5)$P_2$] is essential for exocytosis. Classical ways of manipulating PI(4,5)$P_2$ levels are slower than its metabolism, making it difficult to distinguish effects of PI(4,5)$P_2$ from those of its metabolites. We developed a membrane-permeant, photoactivatable PI(4,5)$P_2$, which is loaded into cells in an inactive form and activated by light, allowing sub-second increases in PI(4,5)$P_2$ levels. By combining this compound with electrophysiological measurements in mouse adrenal chromaffin cells, we show that PI(4,5)$P_2$ uncaging potentiates exocytosis and identify synaptotagmin-1 (the $Ca^{2+}$ sensor for exocytosis) and Munc13-2 (a vesicle priming protein) as the relevant effector proteins. PI(4,5)$P_2$ activation of exocytosis did not depend on the PI(4,5)$P_2$-binding CAPS-proteins, suggesting that PI(4,5)$P_2$ uncaging may bypass CAPS-function. Finally, PI(4,5)$P_2$ uncaging triggered the rapid fusion of a subset of readily-releasable vesicles, revealing a rapid role of PI(4,5)$P_2$ in fusion triggering. Thus, optical uncaging of signaling lipids can uncover their rapid effects on cellular processes and identify lipid effectors.
DOI: https://doi.org/10.7554/eLife.30203.001

## Introduction

Signal transduction between cells depends on the regulated exocytosis of vesicles to liberate neurotransmitters, neuropeptides and hormones. Neuronal exocytosis is driven by an evolutionarily conserved machinery that targets vesicles to the plasma membrane, attaches them to the membrane (sometimes referred to as vesicle 'docking'), and molecularly matures ('primes') them to a fusion-competent state. Primed vesicles reside in the so-called Readily Releasable Pool (RRP) whose fusion triggering leads to transmitter/hormone release. The assembly of the thermodynamically stable neuronal SNARE complex formed by the vesicular SNARE protein VAMP2/synaptobrevin-2 and the plasma membrane SNAREs syntaxin-1 and SNAP-25 provides energy needed for vesicle fusion (*Jahn and Fasshauer, 2012*). Indeed, SNARE proteins are already required for membrane attachment and priming (*de Wit et al., 2009*; *Imig et al., 2014*; *Walter et al., 2010*), and during fusion they continue to influence fusion pore properties through their transmembrane domains

**eLife digest** Cells in our body communicate by releasing compounds called transmitters that carry signals from one cell to the next. Packages called vesicles store transmitters within the signaling cell. When the cell needs to send a signal, the vesicles fuse with the cell's membrane and release their cargo. For many signaling processes, such as those used by neurons, this fusion is regulated, fast, and coupled to the signal that the cell receives to activate release. Specialized molecular machines made up of proteins and fatty acid molecules called signaling lipids enable this to happen.

One signaling lipid called $PI(4,5)P_2$ (short for phosphatidylinositol 4,5-bisphosphate) is essential for vesicle fusion as well as for other processes in cells. It interacts with several proteins that help it control fusion and the release of transmitter. While it is possible to study the role of these proteins using genetic tools to inactivate them, the signaling lipids are more difficult to manipulate. Existing methods result in slow changes in $PI(4,5)P_2$ levels, making it hard to directly attribute later changes to $PI(4,5)P_2$.

Walter, Müller, Tawfik et al. developed a new method to measure how $PI(4,5)P_2$ affects transmitter release in living mammalian cells, which causes a rapid increase in $PI(4,5)P_2$ levels. The method uses a chemical compound called "caged $PI(4,5)P_2$" that can be loaded into cells but remains undetected until ultraviolet light is shone on it. The ultraviolet light uncages the compound, generating active $PI(4,5)P_2$ in less than one second. Walter et al. found that when they uncaged $PI(4,5)P_2$ in this way, the amount of transmitter released by cells increased. Combining this with genetic tools, it was possible to investigate which proteins of the release machinery were required for this effect.

The results suggest that two different types of proteins that interact with $PI(4,5)P_2$ are needed: one must bind $PI(4,5)P_2$ to carry out its role and the other helps $PI(4,5)P_2$ accumulate at the site of vesicle fusion. The new method also allowed Walter et al. to show that a fast increase in $PI(4,5)P_2$ triggers a subset of vesicles to fuse very rapidly. This shows that $PI(4,5)P_2$ rapidly regulates the release of transmitter. Caged $PI(4,5)P_2$ will be useful to study other processes in cells that need $PI(4,5)P_2$, helping scientists understand more about how signaling lipids control many different events at cellular membranes.

DOI: https://doi.org/10.7554/eLife.30203.002

(*Chang et al., 2015*; *Dhara et al., 2016*; *Fang and Lindau, 2014*). The protein families Unc18 (sec-1) and Unc13 interact with the neuronal SNAREs and are required for membrane attachment, priming and fusion (*Rizo and Südhof, 2012*). The $Ca^{2+}$-dependent activator protein for secretion (CAPS) is another priming factor in PC12-cells, chromaffin cells and neurons (*Jockusch et al., 2007*; *Liu et al., 2008*; *Grishanin et al., 2004*), which interacts with SNAREs to stimulate their assembly (*James et al., 2009*). A function of CAPS in membrane attachment of vesicles was also described in synapses of *C.elegans*, hippocampal neurons and PC12 cells (*Imig et al., 2014*; *Kabachinski et al., 2016*; *Zhou et al., 2007*), but not in mouse chromaffin cells (*Liu et al., 2008*). Vesicle fusion is temporally linked to electrical stimulation of the cell by voltage gated $Ca^{2+}$ channels that activate upon depolarization. In mouse chromaffin cells, the resulting increase in intracellular $Ca^{2+}$ concentration is sensed by the vesicular $Ca^{2+}$ binding proteins synaptotagmin-1, and $-7$ (*Schonn et al., 2008*), which interact with the SNAREs (*Zhou et al., 2015*; *Schupp et al., 2016*), and trigger vesicle fusion (*Südhof, 2013*).

The lipid bilayer of the plasma membrane – apart from taking part in the vesicle to plasma membrane merger - contains a variety of signaling lipids that can regulate exocytosis, most notably phosphatidylinositols (PIs) and diacylglycerols (DAGs). Phosphatidylinositols constitute a family of lipids, which can be phosphorylated on one or more positions of the inositol headgroup, giving rise to specific signals on cell membrane compartments (*Di Paolo and De Camilli, 2006*; *Balla, 2013*). $PI(4,5)P_2$ is the major phosphatidylinositol in the inner leaflet of the plasma membrane where it plays multiple essential roles in cell motility, actin cytoskeleton organization, ion channel activity, and vesicle exocytosis (*Di Paolo and De Camilli, 2006*; *Martin, 2012*).

Signaling lipids are recognized by specific protein motifs, particularly C1, C2 and pleckstrin homology (PH) domains (*Martin, 2015*). One of the best characterized signaling lipid interactions with exocytosis proteins is that of the synaptotagmin-1 (syt-1) C2B-domain with $PI(4,5)P_2$ (*Schiavo et al., 1996*; *Honigmann et al., 2013*). The syt-1 C2B domain binds $Ca^{2+}$, and is essential for triggering of vesicle fusion (*Mackler et al., 2002*). In vitro, $PI(4,5)P_2$ binding to the C2B domain markedly increases the affinity for $Ca^{2+}$ (*van den Bogaart et al., 2012*; *Li et al., 2006*). Thus, interactions of syt-1 with synaptic $PI(4,5)P_2$ ensure that the C2B domain interacts with the plasma membrane (*Bai et al., 2004*) and aid the triggering of vesicle fusion by bringing its $Ca^{2+}$-affinity into the physiological range. All known Munc13 isoforms contain C1- and C2 domains that regulate exocytosis. The DAG analog phorbolester binds to the Munc13 C1 domain to strongly enhance exocytosis (*Rhee et al., 2002*; *Basu et al., 2007*). Membrane binding of Munc13 can be further augmented by $PI(4,5)P_2$ binding to the neighboring C2B domain, which also influences the fusion probability of synaptic vesicles (*Shin et al., 2010*). CAPS contains a PH domain which binds $PI(4,5)P_2$ and is essential for vesicle priming (*Nguyen Truong et al., 2014*; *Loyet et al., 1998*; *Kabachinski et al., 2014*). Because different species of signaling lipids may regulate different essential exocytosis proteins, systematic investigation of the relevant interactions for exocytosis is needed.

The most successful approaches to tease apart molecular components relevant for neurosecretion have been to mutate proteins of the release machinery or to regulate their expression. This is not directly possible for signaling lipids; instead, enzymes of lipid metabolism have been targeted. Early experiments in permeabilized bovine adrenal chromaffin cells using bacterial phospholipase C (PLC) showed that secretion depends on $PI(4,5)P_2$ at an upstream ATP-dependent priming step (*Eberhard et al., 1990*). Exocytosis was further found to require the $PI(4,5)P_2$ synthesizing enzyme phosphatidylinositol-4-phosphate 5-kinase (*Hay et al., 1995*; *Gong et al., 2005*). Moreover, experiments using fast capacitance measurements showed that overexpression of $PI(4,5)P_2$ generating or degrading enzymes increased or decreased the RRP, respectively (*Gong et al., 2005*; *Milosevic et al., 2005*). Even though the initial experiments indicated that PLC, which produces DAG at the expense of $PI(4,5)P_2$, inhibits rather than stimulates secretion (*Eberhard et al., 1990*; *Hay et al., 1995*), later experiments showed that phorbolesters – assumed to mimic DAG – strongly stimulate secretion when added to naïve cells (*Smith et al., 1998*). The phorbol ester effect has since been studied in a number of cell types, and has been found to rely on the activation of two priming factors Munc13 (via its C1 domain)(*Rhee et al., 2002*; *Betz et al., 1998*) and Munc18-1 (via protein kinase C phosphorylation)(*Wierda et al., 2007*; *Genc et al., 2014*; *Nili et al., 2006*). In the presence of PLC activity, experiments to increase or decrease the levels of $PI(4,5)P_2$ might cause correlative changes in DAG. The same concern exists for the conversion of $PI(4,5)P_2$ to $PI(3,4,5)P_3$, which might have profound effects on exocytosis in spite of being present in low abundance (*Khuong et al., 2013*). Even the fastest existing techniques to manipulate $PI(4,5)P_2$ levels (by voltage, light, or chemical dimerization)(*Murata et al., 2005*; *Suh et al., 2010*; *Idevall-Hagren et al., 2012*) operate on similar speeds as $PI(4,5)P_2$ metabolism and vesicle priming (i.e. tens of seconds)(*Voets et al., 1999*), making it a general concern how to tease apart the effect of $PI(4,5)P_2$ from that of its metabolites including DAG and other phosphatidylinositols.

To manipulate cellular $PI(4,5)P_2$ levels rapidly and distinguish its function from those of its metabolites in fast cellular reactions we here developed and characterized a new chemical tool: caged, membrane permeant $PI(4,5)P_2$. We show that our compound is taken up into living cells and verify that its UV-uncaging generates physiologically active $PI(4,5)P_2$ with sub-second temporal precision. Uncaging induced the re-distribution of proteins containing $PI(4,5)P_2$ binding motifs and locally increased actin-levels. Capacitance measurements in chromaffin cells showed that following $PI(4,5)P_2$ uncaging exocytosis is enhanced, and the RRP increased, which we demonstrate is specific to $PI(4,5)P_2$ by contrasting the effects of DAG-uncaging. Systematic investigation of the relevant effector proteins revealed a requirement for the potentiation on syt-1 and Munc13-2, but not on CAPS. These results suggest two distinguishable types of $PI(4,5)P_2$ effector proteins: ones that require stoichiometric $PI(4,5)P_2$-binding to exert their function, and ones that function in the local enrichment of $PI(4,5)P_2$ at the vesicle fusion site. Finally, making full use of the rapid uncaging kinetics, we investigate the immediate effects of increasing the levels of signaling lipids on exocytosis and discover that $PI(4,5)P_2$ but not DAG uncaging induces the rapid exocytosis of few vesicles from the RRP. Our data provide an example of how caged lipid compounds can be used to tease apart relevant interactions in fast biological reactions like neurosecretion.

# Results

To achieve fast elevation of PI(4,5)P$_2$ levels on the relevant timescale for exocytosis we developed photoactivatable (caged), membrane-permeant PI(4,5)P$_2$ derivatives (*Figure 1*). Optical uncaging is uniquely suited to increase the levels of signaling molecules non-invasively with high temporal precision (*Höglinger et al., 2014*). Synthesis was based on a commercially available enantiomerically pure precursor (*Figure 1a*). Because the hydroxyl and phosphate groups on the inositol ring make PI(4,5) P$_2$ a highly charged molecule it cannot pass across the cellular plasma membrane. To make the molecule membrane permeant, these groups were equipped with protective groups of acetoxymethyl (AM) esters and butyrates (Bt), respectively (as detailed in *Figure 1a* legend and Methods). Once inside cells, these protective groups are removed by endogenous carboxyesterases (*Schultz, 2003*). Similar approaches were successfully applied to other phosphoinositides previously (*Laketa et al., 2014*; *Mentel et al., 2011*). Photosensitivity was achieved by the addition of a photocleavable cage designed to interfere with biological functions at the phosphate residues in positions 4 or 5 of the inositol ring. The coumarin caging group was chosen for its extraordinarily fast release kinetics as well as its intrinsic fluorescence, which allows verification of cellular uptake. The resulting intermediates **4a** and **4b** (*Figure 1a*) were subsequently coupled to either a dioctanoylglycerol (compound **11**, legend to *Figure 1*) or a stearoyl-arachidonoylglycerol (compound **14**, legend to *Figure 1*) bearing phosphoramidite reagent to form the fully protected caged PI(4,5) P$_2$ intermediates (*Figure 1*). One-pot deprotection and alkylation with AM bromide gave the caged,

**Figure 1.** Synthesis of membrane-permeant and photoactivatable PI(4,5)P$_2$ (cg-PI(4,5)P$_2$). (a) Synthesis of PI(4,5)P$_2$ derivatives **1a,b** and **2a,b**. Reagents and conditions: (a) CH$_2$Cl$_2$:HCO$_2$H 4:1, room temperature (rt), 3 hr, 88%; (b) (FmO)$_2$P-N$i$Pr$_2$**7**, 1*H*-tetrazole, CH$_2$Cl$_2$, rt, 1 hr, then AcO$_2$H, −80°C-rt, 1 hr, 83% over two steps; (c) (Coum)(FmO)P-N$i$Pr$_2$**8**, 1*H*-tetrazole, CH$_2$Cl$_2$, rt, 1 hr, then AcO$_2$H, −80°C-rt, 1 hr; (d) CH$_2$Cl$_2$:HCO$_2$H 1:19, rt, 6 hr; (e) Pr-C(OMe)$_3$, CH$_2$Cl$_2$, JandaJel pyridinium trifluoroacetate, rt, 23 hr, 38% based on **3**. For **1a,b**: (f) (dioctanoylglycerol)(OFm)P-N$i$Pr$_2$**11**, 1*H*-tetrazole, CH$_2$Cl$_2$, rt, 1 hr, then AcO$_2$H, −80°C-rt, 1 hr, 67% over two steps; (g) CH$_2$Cl$_2$, EtNMe$_2$, rt, 30 min; (h) acetoxymethyl bromide, *N,N*-di*iso*propylethylamine, MeCN, rt, 22 hr, 65% over two steps. For **2a,b**: f) (stearoyl-arachidonylglycerol)(OFm)P-N$i$Pr$_2$**14**, 1*H*-tetrazole, CH$_2$Cl$_2$, rt, 1 hr, then AcO$_2$H, −80°C-rt, 1 hr, 89% over two steps; (g) CH$_2$Cl$_2$, EtNMe$_2$, rt, 30 min; (h) acetoxymethyl bromide, *N,N*-di*iso*propylethylamine, MeCN, rt, 22 hr, 43% over two steps. (b) Structure of the caged, membrane-permeant PI(4,5)P$_2$ derivative **2b**. (c) Structure of the de-esterified and uncaged, predominant naturally occurring PI(4,5)P$_2$ variant. (left panel) Ac: acetyl; AM: acetoxymethyl; Bt: butyryl; Coum: 7-diethylamino-4-methyl-2-oxo-2H-chromenyl; Fm: 9-fluorenylmethyl.
DOI: https://doi.org/10.7554/eLife.30203.003

membrane-permeant PI(4,5)P$_2$ derivatives (**1a,b; 2a,b**) in 12% and 10% overall yield, respectively (*Figure 1a–b*).

We first validated that UV uncaging activated lipid-protein interactions with known PI(4,5)P$_2$ binding domains. For this, we directly diluted our compound in an imaging buffer (to a final concentration of 20 μM) and made use of the fact that if the solution was not heavily mixed by vortexing, some of the lipid formed micelles clearly visible on the bottom of the coverslip in the light microscope. The solution furthermore contained a reconstituted fusion protein of the PI(4,5)P$_2$-binding PH domain of PLC-δ$_1$ linked to EGFP. Illumination in the TIRF field was used to limit light excitation to the surface of the glass coverslip. When EGFP was excited, the presence of PH-EGFP was visible as background in the solution of the TIRF field and some of the PH-EGFP was enriched on the micelles. Following UV-uncaging with a 405 nm laser in the TIRF field, the EGFP signal at these positions was greatly increased, indicative of PH-EGFP recruitment to the micelles from the surrounding solution (*Figure 2a*). To confirm that the micelles were indeed composed of cg-PI(4,5)P$_2$, we investigated the images during the irradiation with 405 nm light, which revealed their fluorescence, confirming the presence of the coumarin group (*Figure 2a*). These results demonstrate that UV-cleavage of the coumarin cage activates the compound for interactions with proteins bearing PI(4,5)P$_2$ binding motifs.

To verify that the protective groups synthesized on our compound enabled cellular uptake, we investigated its cellular distribution making use of the intrinsic fluorescence of the coumarin cage. Human Embryonic Kidney (HEK) cells were first loaded with a membrane labelling dye excitable with infrared light (CellMask). Cells were then loaded with caged (cg) PI(4,5)P$_2$ by incubation for 30 min at 37°C with 20 μM of our compound (**2a,b**, diluted from a 20 mM DMSO-stock) in the presence of 0.02% Pluronic (prepared with heavy vortexing) to facilitate membrane passage. Cells receiving identical treatment but without the compound served as controls. To assess the localization of cg-PI(4,5)P$_2$ quantitatively, cells were imaged on a spinning disc confocal microscope and fluorescence line profiles obtained from many cells that bordered open extracellular space (as opposed to ones in contact with other HEK cells). The line profiles were then aligned to local fluorescence maxima of the CellMask signal indicating the position of the plasma membrane. Subsequently, line profiles from all cells were averaged (*Figure 2b*). We saw that average coumarin fluorescence increased inside the plasma membrane, demonstrating cellular uptake. Moreover, coumarin fluorescence was clearly observed at the position of the plasma membrane; however, it was also present inside the cell, possibly on endosomes. This is not surprising, as the coumarin group is expected (and, indeed, intended) to block interactions with PI(4,5)P$_2$-binding proteins. This includes those proteins that usually establish a strict pattern of phosphoinositide composition on distinct cellular organelles. Therefore, it is an unavoidable side effect of using caged lipids that their distribution will be broader than the native lipid. Inevitably, visualization of the coumarin fluorescence by its excitation leads to its uncaging. This could be shown by continuous imaging which significantly reduced the coumarin fluorescence at the location of the plasma membrane (bar graph *Figure 2b*). As expected, neither the fluorescence gradient across the membrane nor the decrease in intensity at the plasma membrane was observed in control cells (*Figure 2b*), indicating that our compound is taken up into cells, present at the plasma membrane and uncaged there.

We next validated that UV uncaging liberated physiologically active PI(4,5)P$_2$ in living cells. For this, we transfected HEK cells with the PLC-δ$_1$-PH-EGFP construct and looked for a possible recruitment of EGFP fluorescence to the plasma membrane by TIRF microscopy upon UV-uncaging. However, we found that even before uncaging, EGFP fluorescence intensities were very high and did not increase further upon UV-uncaging, which we attribute to saturation of the sensor due to relatively high plasma membrane levels of PI(4,5)P$_2$ already at rest (data not shown). To circumvent this problem, we co-transfected COS-7 cells with a plasma membrane targeted 5-phosphatase which degrades PI(4,5)P$_2$ (*Posor et al., 2013*). Upon UV-uncaging in cells loaded with cg-PI(4,5)P$_2$ we found a small, but highly significant increase of EGFP fluorescence at the cell's footprint in line with the liberation of PI(4,5)P$_2$ at the plasma membrane, while no such effect was observed in control cells, which were also subjected to UV-light, but not loaded with cg-PI(4,5)P$_2$ (*Figure 2c*). The relatively small effect size may be caused by the 5-phosphatase activity which likely rapidly degrades uncaged PI(4,5)P$_2$ at the plasma membrane before it can be detected by the PLC-δ$_1$-PH-EGFP sensor which is why this experiment may underestimate the amount of liberated PI(4,5)P$_2$.

We also investigated the behavior of our compound in cells were PI(4,5)P$_2$ was not constitutively depleted, but where degradation was acutely induced pharmacologically. Endogenous PI(4,5)P$_2$

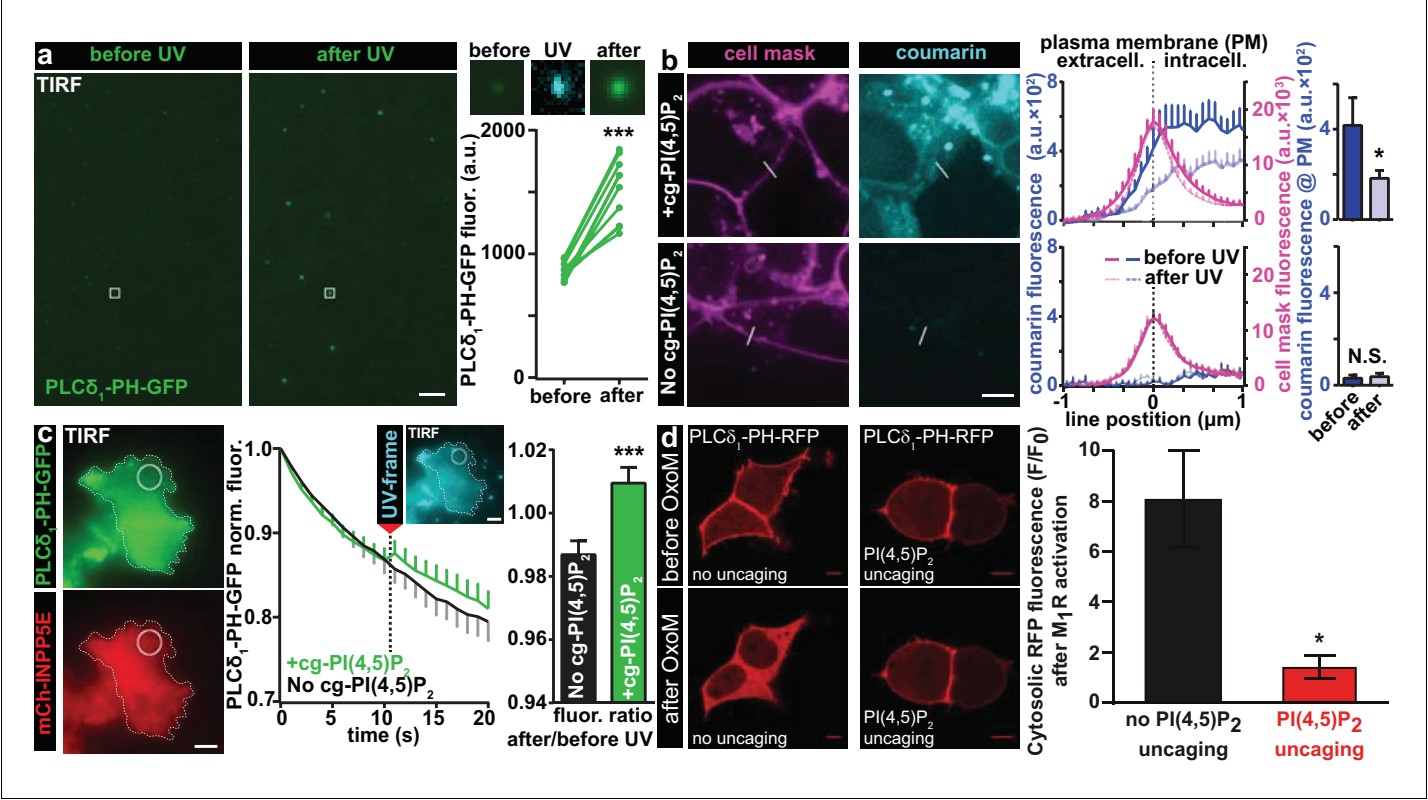

**Figure 2.** Characterization of PI(4,5)P$_2$ UV uncaging in-vitro, loading of cg-PI(4,5)P$_2$ into living cells and visualization of PI(4,5)P$_2$ uncaging in several cell types. (**a**) Uncaging of cg-PI(4,5)P$_2$ micelles on a glass coverslip results in the relocation of a high affinity PI(4,5)P$_2$ sensor, PLCδ$_1$-PH-EGFP, to micelles following UV light exposure, as seen by a local increase in 488 nm excited fluorescence using TIRF microscopy. The two images on the left show the EGFP fluorescence before and after UV uncaging (note the background fluorescence due to soluble PLCδ$_1$-PH-EGFP). The region within the white square is one example of an analyzed micelle. Magnified views are shown on the right before (EGFP fluorescence), during (showing coumarin/cg-PI(4,5)P$_2$-fluorescence) and after (EGFP fluorescence) UV (405 nm) light in the TIRF field. The quantification shows the analysis of the fluorescence of all 10 micelles seen in this image frame. (**b**) HEK cells were either loaded for 30 min at 37°C with 20 µM of cg-PI(4,5)P$_2$ (+cg-PI(4,5)P$_2$, top line), or not loaded (No cg-PI(4,5)P$_2$, bottom line). All cells were treated with the vehicle DMSO (0.2%), Pluronic (0.02%), CellMask Deep Red plasma membrane stain and imaged on a spinning disc confocal microscope. Fluorescence line profiles were collected to investigate cellular uptake of cg-PI(4,5)P$_2$. Profiles were aligned to the local intensity maxima of the CellMask fluorescence indicating the position of the plasma membrane and revealed intracellular coumarin/cg-PI(4,5)P$_2$ (compare dark blue profiles with and without cg-PI(4,5)P$_2$). After cells were exposed to UV (405 nm) illuminations, the intensity distribution of the coumarin fluorescence was altered (light blue profiles) and intensity at the position of the plasma membrane significantly reduced (bar graph in top line), indicating PI(4,5)P$_2$ uncaging. (**c**) COS-7 cells expressing PLCδ$_1$-PH-EGFP (top left panel) and a plasma membrane targeted, m-Cherry tagged inositol polyphosphate 5-phosphatase (mCh-INPP5E) (bottom left panel) were either loaded for 30 min at 37°C with 20 µM cg-PI(4,5)P$_2$ (+cg-PI(4,5)P$_2$) or not loaded (No cg-PI(4,5)P$_2$) and imaged on a TIRF microscope. All cells were treated with the vehicle DMSO (0.2%) and Pluronic (0.02%). Center panel: average EGFP fluorescence of ROIs at the plasma membrane (example shown in the images on the left) imaged at 1 Hz in the TIRF field in both groups (+cg-PI(4,5)P$_2$: green, No cg-PI(4,5)P$_2$: black). Between the 10th and the 11th frame, UV-uncaging was performed. The image acquired during the UV-frame (showing coumarin/cg-PI(4,5)P$_2$-fluorescence) is shown as an insert. Right panel: the fluorescence change following uncaging was calculated by dividing the per-ROI fluorescence values in the 11th frame by those in the 10th frame. In cells loaded with cg-PI(4,5)P$_2$, PLCδ$_1$-PH-EGFP fluorescence increased in the TIRF field after UV-uncaging. (**d**) tsA-201 cells overexpressing M$_1$ muscarinic receptors and PLCδ$_1$-PH-RFP were imaged on a laser scanning confocal microscope. Due to the high affinity of the probe, endogenous PI(4,5)P$_2$ levels are already sufficient to localize the probe to the plasma membrane at the beginning of the experiment (top line). Application of 1 µM of the M$_1$ receptor agonist oxotremorine-M (Oxo-M) resulted in the translocation of the sensor to the cell center indicative of plasmalemmal PI(4,5)P$_2$ breakdown in cells loaded with cg-PI(4,5)P$_2$, but not subjected to UV-uncaging (no uncaging, bottom left image, black bar graph). This response was nearly abolished in cells subjected to UV light (PI(4,5)P$_2$ uncaging, bottom right image and red bar graph). F/F$_0$ signifies the ratio of fluorescence values within the cytosol at the end of the experiment (F) (21–22 s after the uncaging and 20 s after the application of oxotremorine-M) by the fluorescence at the beginning of the experiment (F$_0$). See *Figure 2—figure supplement 1* for further details. Scale bars 5 µm. All values are mean ±SEM. *p<0.05; **p<0.01; ***p<0.001. In panels a and b, paired t-tests were used, in panels c and d, unpaired two-tailed t-tests were performed. Number of cells (n): panel b: n = 14 cells (+cg-PI(4,5)P$_2$), n = 5 cells (No cg-PI(4,5)P$_2$). Panel c: n = 15 cells (+cg-PI(4,5)P$_2$), n = 15 cells (No cg-PI(4,5)P$_2$). Panel d: n = 6 cells (no uncaging), n = 12 cells (PI(4,5)P$_2$ uncaging).

DOI: https://doi.org/10.7554/eLife.30203.004

The following figure supplement is available for figure 2:

**Figure supplement 1.** PI(4,5)P$_2$ uncaging elevates plasmalemmal PI(4,5)P$_2$.
DOI: https://doi.org/10.7554/eLife.30203.005

cause the quantitative plasma membrane binding of the high-affinity PLC-$\delta_1$-PH-RFP sensor in tsA-201 cells (*Figure 2d*). By simultaneously expressing M$_1$ muscarinic receptors in these cells, we could acutely degrade PI(4,5)P$_2$ by the application of oxotremorine-M (Oxo-M) which activates M$_1$ receptors to stimulate PLC. This rapidly decreases PI(4,5)P$_2$ levels selectively at the plasma membrane and reliably induced the relocalization of PLC-$\delta_1$-PH-RFP sensor from the plasma membrane to the cytosol in control cells (*Figure 2d*), indicating near complete plasma membrane PI(4,5)P$_2$ breakdown. Because this assay monitors the sensor's dissociation it may be better suited to monitor PI(4,5)P$_2$ liberation close to the sensor's initial location. We therefore combined OxoM-treatment with PI(4,5)P$_2$-uncaging, which prevented PLC-$\delta_1$-PH-RFP membrane dissociation (*Figure 2d*), in line with substantial PI(4,5)P$_2$ release at the plasma membrane overruling PLC activity. Uncaging itself did not interfere with PI(4,5)P$_2$ breakdown, because DAG was still produced (validated by parallel imaging with a DAG biosensor)(*Figure 2—figure supplement 1b,c*).

To verify that UV-uncaging of our compound activated PI(4,5)P$_2$-dependent cellular responses, we investigated effects of cg-PI(4,5)P$_2$ uncaging on actin bundles, for whose polymerization a pivotal role of PI(4,5)P$_2$ is firmly established (*Di Paolo and De Camilli, 2006*; *Rohatgi et al., 1999*). Actin bundles were visualized by TIRF microscopy in the footprints of HEK cells expressing the actin marker Lifeact-RFP (*Figure 3a*). HEK cells loaded with cg-PI(4,5)P$_2$ were compared to non-loaded cells. Following the measurement of baseline fluorescence (five frames) in the RFP channel, cells were exposed to TIRF illumination with UV light (405 nm laser). This lead to a significant and specific increase in lifeact-RFP in cells loaded with cg-PI(4,5)P$_2$ (*Figure 3a*), in line with PI(4,5)P$_2$ uncaging causing actin accumulation near the plasma membrane.

Because we eventually wanted to investigate the physiological effects of PI(4,5)P$_2$ in adrenal chromaffin cells, we next studied whether PI(4,5)P$_2$ uncaging would also increase PI(4,5)P$_2$ levels at their plasma membrane. As it is a distinct advantage of our compound that cellular PI(4,5)P$_2$ can be rapidly and specifically increased by light without interfering with basic PI(4,5)P$_2$ metabolism, we wanted to verify PI(4,5)P$_2$ uncaging without the prior manipulation of its resting levels. For this we employed a different biosensor harboring the lower affinity PH-domain of PLC$\delta_4$ (*Kabachinski et al., 2014*; *Lee et al., 2004*). Chromaffin cells were infected with a virus expressing PLC$\delta_4$-PH-EGFP, and incubated with caged PI(4,5)P$_2$ (**2a,b**). Experiments were performed on the same microscope setup later used for electrophysiological recordings. Uncaging cg-PI(4,5)P$_2$ by a 1–2 ms light pulse from a Xenon flash bulb caused rapid translocation of the sensor towards the periphery of the cell, in line with PI(4,5)P$_2$ release in the plasma membrane (*Figure 3b*). In cells not incubated with cg-PI(4,5)P$_2$, no translocation was observed. Ongoing imaging of the PLC$\delta_4$-PH-EGFP allowed us to estimate the time constant of the [PI(4,5)P$_2$] relaxation to ~25 s (*Figure 3b*). A second flash caused markedly less translocation, suggesting that most of the cg-PI(4,5)P$_2$ had already been uncaged. In this experiment, the relatively modest recruitment amplitude of to the plasma membrane is probably affected by the widespread intracellular localization of cg-PI(4,5)P$_2$ (*Figure 2b*), because uncaging likely also increases PI(4,5)P$_2$ on intracellular membranes. However, the fact that PH-domains overall relocalize to the plasma membrane (*Figures 2c* and *3b*) indicate that PI(4,5)P$_2$ is uncaged there.

The essential role of PI(4,5)P$_2$ for exocytosis was realized more than 25 years ago, through experiments showing that PI(4,5)P$_2$ depletion inhibits exocytosis from cracked-open adrenal chromaffin cells and PC12 cells (*Eberhard et al., 1990*; *Hay et al., 1995*). Later experiments showed that PI(4,5)P$_2$ delivery to the intracellular compartment via the patch pipette increased the RRP (*Milosevic et al., 2005*). To investigate the physiological effects of PI(4,5)P$_2$ uncaging, cg-PI(4,5)P$_2$ was loaded into the cells (see Materials and methods) and exocytosis was induced with a depolarization protocol to allow Ca$^{2+}$ influx (*Voets et al., 1999*). Exocytosis was monitored using patch-clamp capacitance measurements, which report on increased plasma membrane area upon vesicle fusion. After a pre-pulse, which elicited indistinguishable exocytosis in both groups (*Figure 4a*), cells were either subjected to UV-light flashes (PI(4,5)P$_2$ uncaging group) or not (control group), and a second depolarization protocol was used to assess the effect of uncaging. PI(4,5)P$_2$ uncaging significantly augmented exocytosis in wildtype cells (*Figure 4b,ci*). The overall level of exocytosis found in these

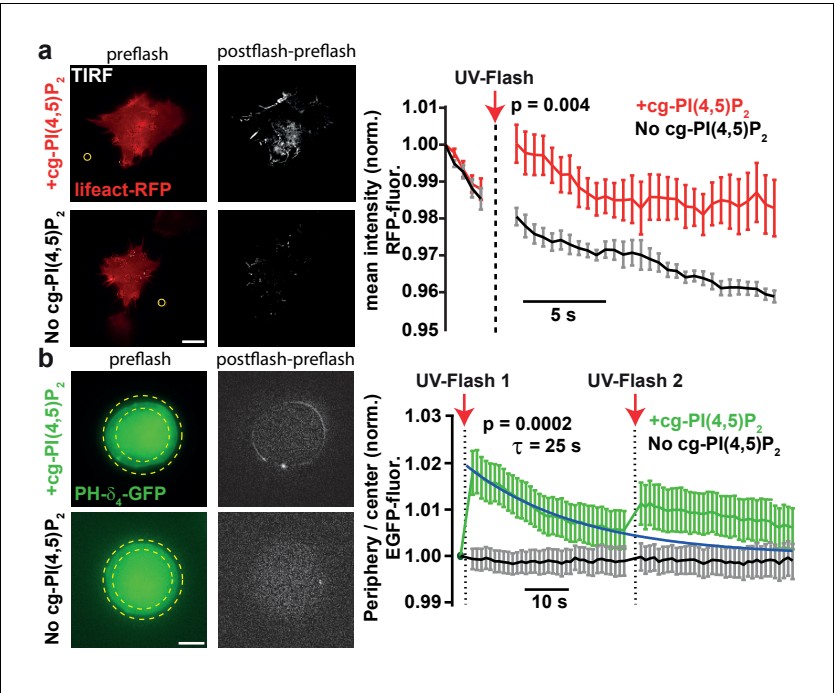

**Figure 3.** PI(4,5)P$_2$ uncaging increases actin levels near the plasma membrane and recruits the low affinity PI(4,5)P$_2$ sensor PLCδ$_4$-PH-EGFP to plasma membranes of adrenal chromaffin cells. (**a**) TIRF imaging of HEK cell footprints transfected with lifeact-RFP to label actin. Cells were either loaded for 30 min at 37°C with 20 µM cg-PI(4,5)P$_2$ (+cg-PI(4,5)P$_2$, top) or not loaded (No cg-PI(4,5)P$_2$, bottom). All cells were treated with the vehicle DMSO (0.2%) and Pluronic (0.02%). Five baseline images were acquired at 2 Hz in the RFP channel, before a 405 nm UV laser was used to uncage PI(4,5)P$_2$ in the TIRF field. Imaging in the RFP channel was then resumed at 2 Hz. The second column depicts difference images of the frames immediately after and before UV exposure (only fluorescence increase is shown). To quantify fluorescence, regions of interests (ROIs) were placed on fluorescence-rich regions that appeared to be actin bundles (white circles in the left images). A background subtraction was performed in each frame (yellow ROI). Fluorescence values were averaged per cell, normalized to the values of the first frame and then averaged across cells. The right panel depicts the average normalized fluorescence per frame in both groups (+cg-PI(4,5)P$_2$: red, No cg-PI(4,5)P$_2$: black). The RFP fluorescence in the TIRF field increased in cells loaded with cg-PI(4,5)P$_2$ after uncaging. (**b**) To verify PI(4,5)P$_2$ uncaging in chromaffin cells, the low-affinity PI(4,5)P$_2$-sensor PLCδ$_4$-PH-EGFP was expressed and cells were imaged on a bright-field fluorescence microscope. Cells were either loaded for 30–45 min at 37°C with 25 µM cg-PI(4,5)P$_2$ (+cg-PI(4,5)P$_2$, top) or not loaded (No cg-PI(4,5)P$_2$, bottom). After a single EGPF frame, a strong UV-flash was applied. Imaging was then resumed in the EGFP channel at 1 Hz. The second column depicts difference images of the frames immediately after and before UV-flash exposure (only fluorescence increase is shown). To quantify translocation of the PLCδ$_4$-PH-EGFP probe, the ratio of EGFP fluorescence in the periphery (between the two yellow dotted circles) and the center of the cell (inner yellow dotted circle) was measured and normalized to pre-flash values. The right panel shows the frame-wise quantification of the average (cell wise) ratio in both groups (+cg-PI(4,5)P$_2$: green, No cg-PI(4,5)P$_2$: black). The fluorescence ratio increased in cells loaded with cg-PI(4,5)P$_2$ after UV-uncaging, indicating release of PI(4,5)P$_2$ in the plasma membrane. The fluorescence ratio relaxed to baseline with a mono-exponential time course (blue line). A second UV-flash applied 38.5 s after the first one also increased the ratio, but to a lesser degree. Scale bars 5 µm. All values are mean ±SEM. Mann-Whitney U-tests were used to calculate p-values. Number of cells (n): panel a: n = 6 cells (+cg-PI(4,5)P$_2$), n = 5 cells (No cg-PI(4,5)P$_2$). Panel b: n = 15 cells (+cg-PI(4,5)P$_2$), n = 20 cells (No cg-PI(4,5)P$_2$).

DOI: https://doi.org/10.7554/eLife.30203.006

experiments is similar to previous findings from wild type Bl6 mice (*Voets et al., 2001a*; *Voets et al., 2001b*). Note that in these experiments, to protect against any effect of the loading protocol or the cg-PI(4,5)P$_2$ compound itself, both control and uncaging groups were loaded with the cg-PI(4,5)P$_2$, but only the uncaging group was exposed to UV-light. Nevertheless, in separate experiments we compared cells loaded with cg-PI(4,5)P$_2$ to cells exposed to the same loading

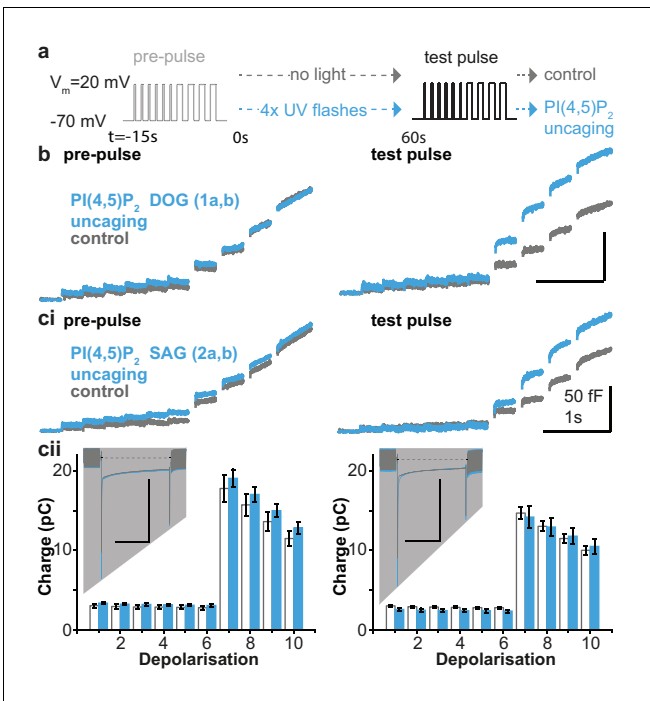

**Figure 4.** PI(4,5)P$_2$ uncaging potentiates exocytosis in adrenal chromaffin cells, which depends on the lipid head group but does not alter depolarization-induced currents. (a) Physiological stimulation paradigm to investigate the effect of PI(4,5)P$_2$ uncaging on exocytosis. Cells were loaded with compounds **1a,b** or **2a,b** prior to experiments. After a pre-pulse of depolarizing voltage steps, cells were either subjected to UV uncaging (PI(4,5)P$_2$ uncaging group) or not (control group). The effect of PI(4,5)P$_2$ uncaging was investigated in a subsequent test pulse. The pre-pulse and the test pulse consisted of six brief (10 ms) and four longer (100 ms) depolarizations to allow Ca$^{2+}$ influx and induce exocytosis (**Voets et al., 1999**). (b,ci) Whole-cell membrane capacitance measurements during the pre- and the test pulse were performed to quantify exocytosis (average traces are shown). (b) Uncaging a PI (4,5)P$_2$ variant featuring a non-natural short-chain fatty acid composition (**1a,b** in **Figure 1a**) increased exocytosis during the test pulse. (ci) Uncaging of PI(4,5)P$_2$ with the natural fatty acid composition (SAG, compound **2a,b**, **Figure 1a,b**) had similar effects. (cii) Depolarization-induced cumulative currents (charges, Q, which mostly originate from Ca$^{2+}$-currents) were similar between both groups for all 10 depolarization steps of pre- and test pulse. Insert: average currents during the first 100 ms depolarization, dashed line indicates baseline. Scale bar in the insert: 0.5 nA and 50 ms. See **Figure 4—figure supplement 2** for corresponding analysis of compound **1a,b**. Number of cells (n): n = 27 (wild type control, loaded with **1a,b**), n = 26 (wild type PI(4,5)P$_2$ uncaging, loaded with **1a,b**); n = 23 (wild type control, loaded with **2a,b**), n = 23 (wild type PI(4,5)P$_2$ uncaging, loaded with **2a,b**).
DOI: https://doi.org/10.7554/eLife.30203.007

The following figure supplements are available for figure 4:

**Figure supplement 1.** Incubation with cg-PI(4,5)P$_2$ does not affect exocytosis.
DOI: https://doi.org/10.7554/eLife.30203.008

**Figure supplement 2.** Uncaging of PI(4,5)P$_2$ DOG (compound 1a,b in **Figure 1a**) does not alter depolarization-induced currents.
DOI: https://doi.org/10.7554/eLife.30203.009

**Figure supplement 3.** Uncaging of PI(4,5)P$_2$ does not cause an increase of intracellular [Ca$^{2+}$] but enhances the rate of single vesicle fusion events.
DOI: https://doi.org/10.7554/eLife.30203.010

protocol, but without cg-PI(4,5)P$_2$, and found that without UV light cg-PI(4,5)P$_2$ had no effect on depolarization induced exocytosis (**Figure 4—figure supplement 1**).

We specifically investigated the role of the fatty acid tail in exocytosis using two different compounds, one containing a short DOG-analog (**1a,b**) and one containing the natural SAG-chain (**2a,b**). However, we found similar potentiation in both cases, showing that the inositol headgroup is responsible for the enhanced exocytosis (**Figure 4b,ci**). Importantly, the augmentation of exocytosis was not due to changes in Ca$^{2+}$ influx during membrane depolarization, as revealed by similar Ca$^{2+}$

currents during the depolarizations (*Figure 4cii*, *Figure 4—figure supplement 2*), and unchanged intracellular $Ca^{2+}$ concentration immediately after PI(4,5)P$_2$ uncaging (*Figure 4—figure supplement 3a*).

Because the compounds **1a,b** and **2a,b** elicited similar effects, we pooled both datasets for further analysis. The depolarization protocol included steps of varying duration which can be used to quantify the release of different populations of vesicles undergoing fusion. The first six 10 ms depolarizations release vesicles positioned close to $Ca^{2+}$ channels in the so-called Immediately Releasable Pool (IRP)(*Voets et al., 1999*), while the following two longer (100 ms) depolarizations (a total of four were given) are assumed to deplete the full Readily Releasable Pool (RRP; the IRP is part of the RRP [*Voets et al., 1999*]). We found that PI(4,5)P$_2$ uncaging did not influence the release of the IRP. However, the RRP was approximately doubled by PI(4,5)P$_2$ uncaging (*Figure 5a,b*), confirming that increasing PI(4,5)P$_2$ enhances priming of vesicles into the RRP (*Milosevic et al., 2005*). In addition, secretion elicited by residual $Ca^{2+}$ in-between depolarizations was enhanced (as seen by steeper

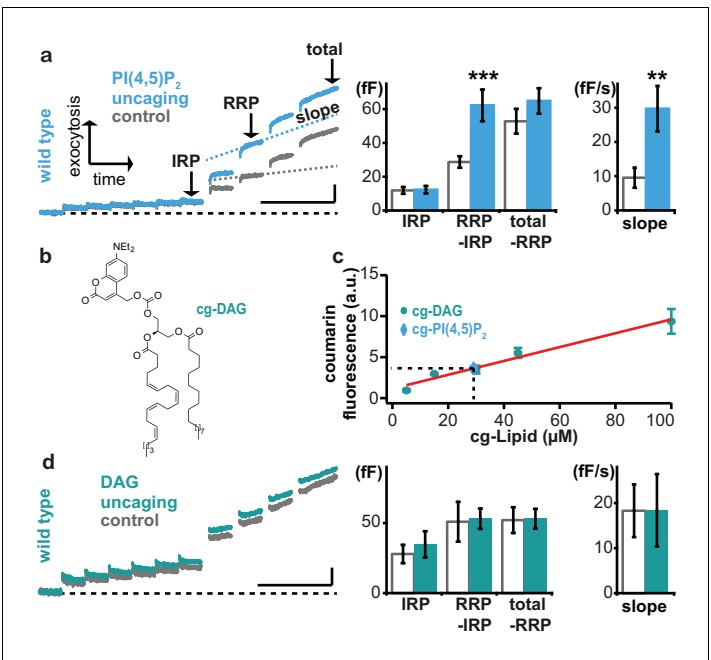

**Figure 5.** Uncaging of PI(4,5)P$_2$, but not DAG augments exocytosis. (**a**) Left panel: mean whole-cell capacitance responses during the test pulse of chromaffin cells loaded with cg-PI(4,5)P$_2$ (data from compounds **1a,b** and **2a,b** pooled, uncaging group: blue, control group: grey). Different secretion phases are indicated. Right panel: cell-wise quantification. IRP = Immediately Releasable Pool; RRP = Readily Releasable Pool, slope = slope determined by linear fit of sliding capacitance. (**b**) Structure of caged DAG (cg-DAG). (**c**) Titration to determine the intracellular cg-PI(4,5)P$_2$ concentration by comparison of coumarin fluorescence in cells loaded with known cg-DAG concentrations via the patch pipette, yielding [cg-PI(4,5)P$_2$]=29 µM. (**d**) Left panel: depolarization-induced capacitance (average trace) elicited by the test pulse (same stimulation as in *Figure 4a*) in cells exposed to DAG uncaging (green), or not (grey, control). No augmentation was seen. Middle and right panel: quantification of IRP, RRP, total secretion and slope revealed no significant changes. Scale bars 20 fF/1 s. Statistical testing by unpaired Student's t-test; **p<0.01; ***p<0.001. Number of cells (**n**): panel a: n = 50 (wild type control), n = 49 (wild type PI (4,5)P$_2$ uncaging); panel c: n = 16 (cg-DAG 5 µM), n = 3 (cg-DAG 15 µM), n = 6 (cg-DAG 30 µM), n = 4 (cg-DAG 45 µM), n = 5 (cg-DAG 100 µM), n = 14 (cg-PI(4,5)P$_2$), panel d: we used two different cg-DAG concentrations (cg-DAG, 45 µM = 6 cells and cg-DAG, 30 µM = 15 cells); pooled results are shown; n = 20 (wild type control), n = 21 (wild type DAG-uncaging).

DOI: https://doi.org/10.7554/eLife.30203.011

The following figure supplement is available for figure 5:

**Figure supplement 1.** Blocking PI(4,5)P$_2$-degradation to DAG augments recovery of the RRP.

DOI: https://doi.org/10.7554/eLife.30203.012

slopes of the capacitance increase, *Figure 5a*, right-hand panels); this could be due to faster priming followed by fusion or increased fusion probability of the remaining vesicles.

Augmentation of the RRP was noted before in adrenal chromaffin cells following longer-term elevation of PI(4,5)P$_2$ (*Milosevic et al., 2005*), but those manipulations might also have elevated the levels of the downstream metabolite DAG. Indeed, Phorbol esters, which are assumed to act as DAG analogues, augment the RRP size in chromaffin cells (*Smith et al., 1998*). Therefore, we wanted to distinguish between PI(4,5)P$_2$ vs. DAG requirements for rapid exocytosis augmentation in chromaffin cells. To this end we performed DAG uncaging. Coumarin-caged DAG (*Nadler et al., 2013*) (cg-DAG, *Figure 5b*) was infused into cells via the patch pipette. Because both cg-DAG and cg-PI(4,5)P$_2$ bear the same fluorescent coumarin cage, titration of coumarin fluorescence at different (known) DAG concentrations in the patch pipette allowed us to estimate the cellular concentration of cg-PI(4,5)P$_2$ in the experiments above. This concentration was not known, since AM-ester loading allows progressive accumulation in the cell with time. Based on comparable fluorescence values in the titration, we estimated that the final intracellular concentration of the caged PI(4,5)P$_2$ corresponded to ~29 µM (*Figure 5c*) and therefore performed DAG uncaging using 30 or 45 µM DAG. However, UV-induced DAG uncaging (unlike PI(4,5)P$_2$ uncaging) failed to potentiate secretion in adrenal chromaffin cells (*Figure 5d*).

Next, we sought an independent method to confirm that refilling of the primed vesicle pool (RRP) depends on PI(4,5)P$_2$, and not on DAG in a similar concentration regime. To this end, we performed double Ca$^{2+}$-uncaging experiments (*Figure 5—figure supplement 1*). In these experiments no caged lipids were present, but UV uncaging of a photolysable Ca$^{2+}$ chelator allowed the direct triggering of vesicles without voltage depolarization of the cell. Ca$^{2+}$ uncaging increases intracellular Ca$^{2+}$ concentrations to the tens-of micromolar range, which is sufficient to deplete the entire RRP and expected to activate endogenous PLC, leading to PI(4,5)P$_2$ degradation. Two sequential Ca$^{2+}$-uncaging stimuli were used to assess RRP sizes and thus refilling in cells incubated with a PLC inhibitor, or an inactive control compound. Recovery of the RRP was significantly enhanced by inhibiting PLC (*Figure 5—figure supplement 1*), indicating that preventing PI(4,5)P$_2$ degradation enhances vesicle priming. Thus, conversion of endogenous PI(4,5)P$_2$ to DAG is overall negative for refilling of the RRP, which confirms our findings using caged lipid compounds.

We next sought to identify relevant PI(4,5)P$_2$ effectors among the molecular release machinery. Syt-1, the Ca$^{2+}$ sensor for rapid exocytosis in chromaffin cells (*Voets et al., 2001a*), was among the first PI(4,5)P$_2$-binding presynaptic proteins to be identified (*Schiavo et al., 1996*; *Honigmann et al., 2013*; *van den Bogaart et al., 2012*; *Bai et al., 2004*). The relevance of PI(4,5)P$_2$-binding was indicated by mutation, which increased the Ca$^{2+}$ requirements for exocytosis. However, mutations can have other effects than those intended in the experiment. For instance, the same residues in the syt-1 C2B domain interacting with PI(4,5)P$_2$ were also shown to interact with the neuronal SNARE complex (*Zhou et al., 2015*). Therefore, and to complement those experiments, we here uncaged PI(4,5)P$_2$ in syt-1 knockout mice, and found that uncaging did not potentiate exocytosis (*Figure 6a*). Proper loading of the compound was ensured after the experiment by the intrinsic fluorescence of the coumarin group (*Figure 6—figure supplement 1a*). Exocytosis from syt-1 KO cells is reduced compared to wild type cells (*Voets et al., 2001a*), although sizable release – including a small RRP (*Mohrmann et al., 2013*) – remains. We asked whether the lack of PI(4,5)P$_2$ augmentation was due to the smaller exocytosis amplitude, rather than the lack of syt-1. To this end, we reanalyzed data, identifying wild type cells with intrinsically low exocytosis amplitude, and syt-1 KO cells with high exocytosis amplitude. However, we still found significant potentiation in WT cells, but not in syt-1 KO cells (*Figure 6—figure supplement 1d*), suggesting a molecular requirement for syt-1. Neurosecretion is known to depend on the key vesicle priming factor Munc13, and the relevant isoform in chromaffin cells, Munc13-2, harbors a C2-domain (C2B), which displays a strong PI(4,5)P$_2$-dependence (*Shin et al., 2010*; *Kabachinski et al., 2014*). Adrenal chromaffin cells isolated from Munc13-2 knockout mice lacked the capacity of PI(4,5)P$_2$ uncaging to potentiate exocytosis (*Figure 6b*). Thus, PI(4,5)P$_2$ potentiation in chromaffin cells occurs via specific activation of the vesicular release machinery and requires syt-1 and Munc13-2. To identify additional molecular targets, we repeated experiments in knockout mouse cells for the major PI(4,5)P$_2$ binding proteins CAPS1 and −2. CAPS interacts with PI(4,5)P$_2$ via a pleckstrin homology domain and loss of this interaction impedes vesicle exocytosis by reducing the number of releasable vesicles (*Nguyen Truong et al., 2014*). However, uncaging PI(4,5)P$_2$ in CAPS1 and −2 double knockout mice revealed a similar enhancement of

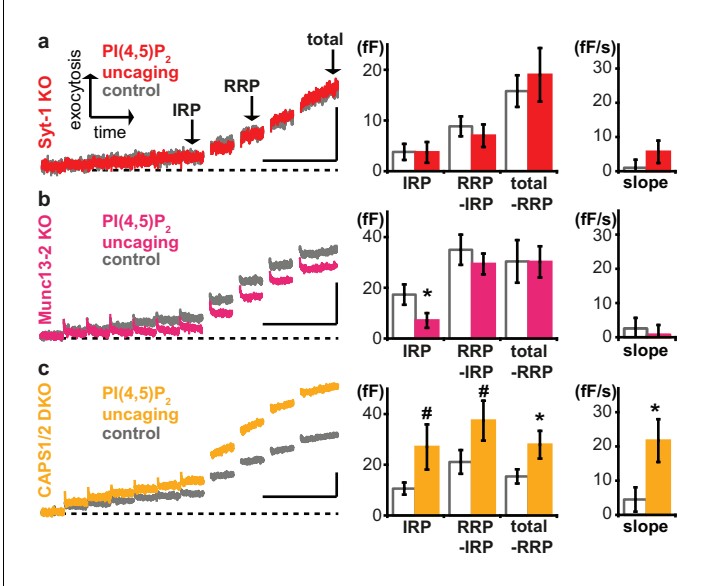

**Figure 6.** Exocytosis potentiation by PI(4,5)P$_2$ uncaging requires synaptotagmin-1 and Munc13-2, but not CAPS. (a–c) All cells were loaded with cg-PI(4,5)P$_2$ prior to experiments and subjected to the stimulation paradigm shown in *Figure 4a*. Average whole-cell capacitance responses during the test pulse are shown and the secretion phases analyzed by cell-wise statistics (for secretion during the pre-pulse and further quantification see *Figure 6—figure supplement 1*). Cells were either subjected to UV uncaging (PI(4,5)P$_2$ uncaging group) or not (control group). In Syt1-KO (a, red) and Munc13-2 KO (b, magenta) cells, exocytosis was not enhanced by PI(4,5)P$_2$ uncaging. (c) In contrast, average exocytosis in CAPS 1/2 double knockout (DKO) cells (yellow) was increased. Scale bar 20 fF/1 s. Statistical testing by unpaired Student's t-test; #p<0.08; *p<0.05. Number of cells (n): n = 33 (syt-1 KO control), n = 36 (syt-1 KO PI(4,5)P$_2$ uncaging), n = 32 (Munc13-2 KO control), n = 37 (Munc13-2 KO PI(4,5)P$_2$ uncaging), n = 21 (CAPS1/−2 DKO control), n = 20 (CAPS1/−2 DKO PI(4,5)P$_2$ uncaging).

DOI: https://doi.org/10.7554/eLife.30203.013

The following figure supplement is available for figure 6:

**Figure supplement 1.** Uncaging cg-PI(4,5)P$_2$ in-between the pre-pulse and the test pulse enhances exocytosis during the test-pulse.

DOI: https://doi.org/10.7554/eLife.30203.014

release upon PI(4,5)P$_2$ uncaging as in wild type cells (*Figure 6c*), arguing that augmentation of exocytosis observed here occurs independently of CAPS, or bypasses CAPS (see Discussion). Surprisingly, the IRP size was actually reduced by PI(4,5)P$_2$-uncaging in the Munc13-1 KO, whereas it was (nonsignificantly, p<0.08) increased in the CAPS-1/2 DKO (*Figure 6*). The implication of this finding is unclear, but Munc13 and CAPS-proteins play distinct roles during priming (*Kabachinski et al., 2014*; *Liu et al., 2010*), and if they are both required for the formation of the IRP-vesicles, then the elimination of one or the other might create IRPs with distinct properties, including PI(4,5)P$_2$-dependence. We conclude that PI(4,5)P$_2$-dependent activation of exocytosis operates via Munc13-2 and syt-1 to potentiate RRP size.

Use of cg-PI(4,5)P$_2$ for the first time allowed investigating the consequences of an abrupt increase in PI(4,5)P$_2$ abundance on a subsecond timescale. When inspecting the capacitance trace around the first uncaging flash (see *Figure 4a* for stimulation protocol), we found an abrupt jump in the capacitance, indicating fast fusion of a few (5-10) vesicles (*Figure 7a*). This jump was observed only with the first uncaging flash, indicating that it is unlikely to be a photo-artifact (*Figure 7—figure supplement 1*). The release appears specific, because the size of the response strongly correlated with the RRP sizes in these cells (*Figure 7b*). Furthermore, uncaging of the PI(4,5)P$_2$ downstream metabolite DAG, bearing the same photolysable coumarin group as PI(4,5)P$_2$ did not induce any capacitance increase (*Figure 7a–c*). Finally, the jump was reduced in size – or absent – in cells from Munc13-2, CAPS-1/2 and syt-1 knockout mice, which all have smaller RRP sizes (*Liu et al., 2008*; *Mohrmann et al., 2013*; *Man et al., 2015*) (*Figures 5*, *6* and *7b*). Thus, rapidly increasing PI(4,5)P$_2$

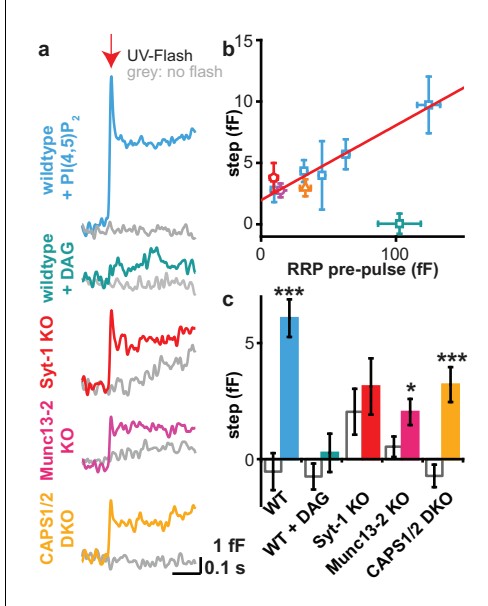

**Figure 7.** Uncaging PI(4,5)P$_2$ induces rapid exocytosis. (a) PI(4,5)P$_2$ uncaging rapidly increased membrane capacitance measured during the first uncaging flash (stimulation protocol: see *Figure 4a*), indicative of fast vesicle fusion. Averaged capacitance traces during the first uncaging flash are shown for wild type (WT, light blue), Syt-1 KO (red), Munc13-2 KO (magenta) and CAPS1/−2 DKO (yellow) and together with their respective controls (no UV light, grey). Note that the uncaging event follows the first depolarization train, and in the syt-1 KO there is still some ongoing, delayed secretion, as indicated by the upward 'sloping' control trace. (b) In wild type cells, the size of the capacitance step was highly correlated to the size of the readily releasable pool (RRP; assayed during the pre-pulse – see *Figure 4a*). Data are median values ± SEM of cells sorted by their RRP size and binned. The correlation (corr. R$^2$-value: 0.97) indicates that the capacitance step is likely caused by rapid fusion of RRP vesicles. (c) Quantification of traces depicted in (a). Shown is the average capacitance increase (from the first to the last value shown in (a)) in control (no UV light, grey) and uncaging groups. Statistical testing by unpaired Student's t-test; *p<0.05; **p<0.01; ***p<0.001. Number of cells (n): n = 50 (wild type, control, data of the compounds **1a,b** and **2a,b** were pooled), n = 49 (wild type, PI(4,5)P$_2$ uncaging, data of the compounds **1a,b** and **2a,b** were pooled), n = 21 (CAPS1/−2 DKO control), n = 20 (CAPS1/−2 DKO PI(4,5)P$_2$ uncaging), n = 32 (Munc13-2 KO control), n = 37 (Munc13-2 KO PI (4,5)P$_2$ uncaging), n = 33 (Syt-1 KO control), n = 36 (syt-1 KO PI(4,5)P$_2$ uncaging).

DOI: https://doi.org/10.7554/eLife.30203.015

The following figure supplement is available for figure 7:

levels can fuse vesicles. The release of a fraction of the RRP is consistent with the stimulation of some vesicles close to fusion threshold (*Yang et al., 2002*) whose Ca$^{2+}$-sensitivity may increase further due to the interaction of syt-1 with PI(4,5)P$_2$ (*van den Bogaart et al., 2012*; *Li et al., 2006*), leading to increased release probability.

## Discussion

Here, we developed a photocaged membrane-permeant PI(4,5)P$_2$ and combined this compound with high time-resolution electrophysiology and genetic manipulations to identify relevant PI(4,5) P$_2$ effectors in neuroendocrine chromaffin cells. The main PI(4,5)P$_2$-binding proteins in the secretory pathway are syt-1 (*Schiavo et al., 1996*; *Honigmann et al., 2013*; *van den Bogaart et al., 2012*; *Bai et al., 2004*), the Ca$^{2+}$-sensor for exocytosis, and Munc13-2 (*Shin et al., 2010*) and CAPS (*Loyet et al., 1998*; *Kabachinski et al., 2014*), two priming proteins, which are responsible for establishing and replenishing the RRP (*Man et al., 2015*; *Liu et al., 2010*). Possible effects of PI(4,5)P$_2$-binding to these proteins in living cells were previously not investigated by altering PI(4,5)P$_2$ levels, but by using correlative analyses following protein mutation (*Li et al., 2006*; *Shin et al., 2010*). However, exactly which of these proteins are acutely activated by PI(4,5) P$_2$ to facilitate secretion was not clear. By uncaging our compound we could now verify that PI (4,5)P$_2$ enhances exocytosis and by studying mouse knockouts we could provide mechanistic insight into distinct PI(4,5)P$_2$-dependent processes in exocytosis:

- Synaptotagmin-1 and Munc13-2 are required for the potentiating effect of PI (4,5)P$_2$ on exocytosis.
- The priming- and PI(4,5)P$_2$-binding proteins CAPS-1/2 are not involved in (or are bypassed by, see below) the potentiating effect of PI(4,5)P$_2$ uncaging.
- Increasing PI(4,5)P$_2$ triggers the rapid release of a part of the Readily Releasable Pool of vesicles.

PI(4,5)P$_2$ uncaging specifically potentiated RRP size, but not the size of the IRP, which forms a subpool of the RRP, consisting of vesicles co-localized with Ca$^{2+}$-channels (*Voets et al., 1999*). One interpretation is that IRP-vesicles are already saturated with PI(4,5)P$_2$; another possible explanation is that the IRP is limited in size by additional factors (for instance the availability of Ca$^{2+}$-channels), and therefore cannot be further

*Figure 7 continued*

**Figure supplement 1.** Fast release of vesicles upon first PI(4,5)P$_2$ uncaging event in wild type chromaffin cells.

DOI: https://doi.org/10.7554/eLife.30203.016

augmented. Based on the different types of effects we observe in our experiments (loss of PI(4,5)P$_2$-dependent augmentation in syt-1 KO and Munc13-2 KO vs. no effect in CAPS DKO), two different mechanisms of PI(4,5)P$_2$-binding proteins can be envisioned (*Figure 8*): the first is specific, and probably stoichiometric, PI(4,5)P$_2$-binding to support protein function, for instance the ability of Munc13 to stimulate SNARE-complex formation, or to increase the Ca$^{2+}$ binding affinity of syt-1 during secretion triggering. The other potential function is to co-localize the fusion machinery with PI(4,5)P$_2$-patches in the plasma membrane. The latter, but not the former, function might be bypassed by PI(4,5)P$_2$-uncaging, which uncovers PI(4,5)P$_2$ under the vesicle (*Figure 8*). Thus, lipid uncaging can serve as an exquisite tool to distinguish between these two different functions of PI(4,5)P$_2$-binding proteins, just as Ca$^{2+}$ uncaging has been instrumental in distinguishing between effects on Ca$^{2+}$-binding to the release machinery itself, and effects on colocalizing vesicles with Ca$^{2+}$ channels (*Voets et al., 1999*; *Wadel et al., 2007*). Therefore, the established essential requirement

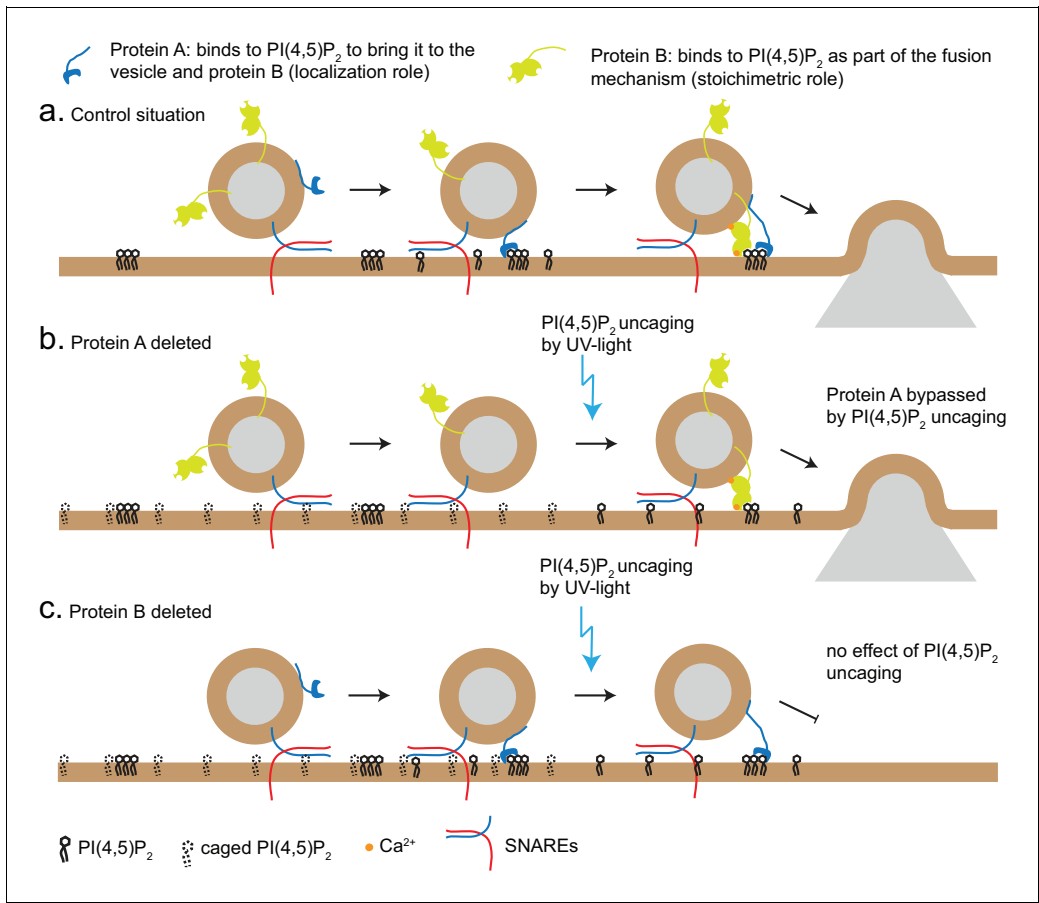

**Figure 8.** Uncaging PI(4,5)P$_2$ distinguishes mechanism of lipid-binding. Two different roles of lipid-binding proteins can be distinguished by lipid uncaging: protein A (e.g. CAPS) binds to PI(4,5)P$_2$ in order to bring it to the vesicle and fusion machinery (localization role); protein B (e.g. synaptotagmin-1) binds to PI(4,5)P$_2$ as an obligatory part of its mechanism (stoichiometric role). (a) In the control situation, protein A colocalizes PI(4,5)P$_2$ with protein B, leading to fusion. (b). Upon deletion of protein A, protein B is unable to interact with PI(4,5)P$_2$. Uncaging uncovers PI(4,5)P$_2$ underneath the vesicle, allowing protein B to interact and support secretion. Thus, the defect (lack of protein A) is bypassed by PI(4,5)P$_2$ uncaging. (c) Upon deletion of protein B, protein A still co-localizes PI(4,5)P$_2$ with the vesicle, but protein B is missing and PI(4,5)P$_2$ uncaging does not overcome the defect.
DOI: https://doi.org/10.7554/eLife.30203.017

of the CAPS PH-domain for its function in vesicle priming (*Nguyen Truong et al., 2014*; *Kabachinski et al., 2014*) can be reconciled with our findings here, if the function of CAPS is to cause local enrichment of PI(4,5)$P_2$ at the sites of vesicle priming, where it will interact with other exocytotic proteins.

The use of uncaging made it possible for the first time to investigate the consequences of an acute, millisecond, increase in PI(4,5)$P_2$ and DAG abundance. We found that PI(4,5)$P_2$, but not DAG, uncaging caused the rapid fusion of vesicles (*Figure 7a,c*). The amount of fusion correlated with the RRP size, and this correlation extended to the knockouts tested (*Figure 7b*). Thus, the acutely fusing vesicles probably constitute a fraction of the RRP. The fact that rapid fusion was only seen when uncaging PI(4,5)$P_2$, but not DAG, argue that PI(4,5)$P_2$ is more directly linked to exocytosis triggering in adrenal chromaffin cells, possibly because PI(4,5)$P_2$ binding to the C2-domains in Munc13-2 and syt-1 directly change the $Ca^{2+}$ affinities of those domains. The fusing vesicles might be members of the 'Highly Calcium Sensitive Pool' (HCSP), which fuse at lower $Ca^{2+}$ concentrations than the rest of the RRP vesicles (*Yang et al., 2002*). Since these vesicles are close to fusion threshold, rapid binding of PI(4,5)$P_2$ might increase the $Ca^{2+}$-affinity of syt-1 enough that the vesicles fuse due to a rapid increase in $Ca^{2+}$-affinity rather than a rapid increase in $Ca^{2+}$ concentration as would normally be the case.

A definite advantage of the photocaged approach is that it allows inducing sub-second increases in the phospholipid composition of membranes, which can be used to identify direct effects of a phospholipid before its metabolism takes place. A possible complication is that by shielding the head group, the lipid will no longer be recognized by proteins (enzymes, lipid-shuttling proteins) that establish the cellular pattern of lipid composition between different organelles. Thus, the localization of the caged lipid will likely be broader than for the native lipid. Indeed, investigating the sub-cellular distribution of our cg-PI(4,5)$P_2$ revealed the uptake in compartments other than the plasma membrane (*Figure 2b*). However, our data also clearly show its specific uncaging at the plasma membrane, making it a suitable tool to address reactions at the latter (*Figures 2* and *3*). Moreover, uncaging PI(4,5)$P_2$ in chromaffin cells led to a specific potentiation of vesicle priming (*Figures 4–6*), which is consistent with previous findings using enzymatic over/underexpression (*Gong et al., 2005*; *Milosevic et al., 2005*), and with the use of a PLC-inhibitor to prevent PI(4,5)$P_2$ breakdown (*Figure 5—figure supplement 1*). Furthermore, the effect depended on known PI(4,5)$P_2$-binding proteins. Thus, although we cannot rule out that PI(4,5)$P_2$ is liberated elsewhere in the cell, PI(4,5)$P_2$ uncaging results in valid and specific effects on exocytosis. The wide-spread distribution of our caged compound may also be considered a distinct advantage, because this allows to study the consequences of its focal liberation in regions where PI(4,5)$P_2$ is sparse.

Collectively, our data demonstrate the power of caged phospholipids to dissect physiological functions of different, but interconvertible, phospholipids. The main power of the approach is that it outpaces the rate of metabolism/interconversion of one lipid into another. Using this method we have dissected the molecular requirement for the potentiating effect of PI(4,5)$P_2$ on exocytosis, and we have demonstrated a novel, acute effect of uncaged PI(4,5)$P_2$: to trigger rapid exocytosis. We anticipate that caged lipid second messengers will serve as valuable experimental tools to uncover mechanistic details of fast cellular processes.

## Materials and methods

### Key resources table

| Reagent type (species) or resource | Designation | Source or reference | Identifiers | Additional information |
|---|---|---|---|---|
| strain, strain background (Mus.musculus) | CD1 | Department of Experimental Medicine, Faculty of Health and Medical Sciences, Unviersity of Copenhagen. | | |
| genetic reagent (M.musculus) | Syt-1 nul allele (gene symbol: syt1) | *Geppert et al., 1994* | PMID: 18308932 | |
| genetic reagent (M.musculus) | Munc13-2 null allele (gene symbol: Unc13b) | *Varoqueaux et al., 2002* | PMID: 12070347 | |

*Continued on next page*

*Continued*

| Reagent type (species) or resource | Designation | Source or reference | Identifiers | Additional information |
|---|---|---|---|---|
| genetic reagent (M.musculus) | CAPS1 null allele (gene symbol: Cadps) | *Speidel et al., 2005* | PMID: 15820695 | |
| genetic reagent (M.musculus) | CAPS2 null allele (gene symbol: Cadps2) | *Jockusch et al., 2007* | PMID: 18022372 | |
| cell line (HEK 293T) | HEK 293T | ATCC | CRL-1573 | Experiments in *Figure 2b* |
| cell line (HEK 293T) | HEK 293T | A gift from Dr. Theres Schaub and Prof Victor Tarabykin, Institute of Cell Biology and Cell Biology, Charité Berlin | | Experiments *Figure 3a* |
| cell line (COS-7) | COS-7 | ATCC | CRL-1651 | |
| cell line (tsA201) | tsA201 | Sigma-aldrich | Sigma-aldrich: 96121229 | |
| transfected lentiviral construct (p156rrl-pCMV-PLCδ4PH-EGFP) | PLCδ4-PH-GFP | This paper | Local reference: 131 | plasmid with PLCδ4 received from Thomas F. J. Martin (Department of Biochemistry, University of Wisconsin) |
| transfected construct (pCMV-PLCδ1-PH-EGFP) | PLCδ1-PH-GFP | Michael Krauss (Leibniz-Forschungsinstitut für Molekulare Pharmakologie, Berlin, Germany). | | |
| transfected construct (pCMV-PLCδ1-PH-RFP) | PLCδ1-PH-RFP | Ken Mackie (The Gill Center for Biomolecular Science, Bloomington, Indiana) | | |
| transfected construct (pCMV-mcherry-INPP5E) | mCh-INPP5E | *Posor et al., 2013* | | |
| transfected construct (pCMV-mRFPruby-N1*Lifeact) | lifeact-RFP | Geerd van den Bogaart (Radboud University Medical Center, Nijmegen, The Netherlands) | PMID: 18536722 | |
| commercial assay or kit | QIAprep Spin Miniprep Kit | Qiagen | | |
| commercial assay or kit | QIAquick Gel Extraction Kit | Qiagen | | |
| commercial assay or kit | QIAquick PCR Purification Kit | Qiagen | | |
| chemical compound, drug | DMSO | Sigma-aldrich | Sigma-aldrich: D8418 | |
| chemical compound, drug | Ascorbic aci | Sigma-aldrich | Sigma-aldrich: A5960 | |
| chemical compound, drug | CaCl2 | Sigma-aldrich | Sigma-aldrich: 499609 | |
| chemical compound, drug | CellMask | Invitrogen | Invitrogen: C10046 | |
| chemical compound, drug | CsOH | Sigma-aldrich | Sigma-aldrich 516988 | |
| chemical compound, drug | DMEM | Gibco/Thermo Fisher | Gibco/Thermo Fisher: 31966047 | Experiments in *Figure 3a* |
| chemical compound, drug | DMEM | Lonza | Lonza: BE12-741F | Experiments in *Figure 2b,c* |
| Chemical compound, drug | HBSS | Gibco/Thermo Fisher | 14025–050 | |
| chemical compound, drug | caged DOG-PI(4,5)P2 | This paper | | European Molecular Biology Laboratory (EMBL), Cell Biology and Biophysics Unit, Meyerhofstr. 1, 69117 Heidelberg, Germany. Att: Carsten Schultz (schultz@embl.de) |
| chemical compound, drug | EDTA | Sigma-aldrich | Sigma-aldrich: E5134 | |
| chemical compound, drug | Fetal Bovine Serum (FBS) | Gibco/Thermo Fisher | Thermo Fisher/Gibco: 16140063 | Experiments in *Figure 3a* |

*Continued on next page*

*Continued*

| Reagent type (species) or resource | Designation | Source or reference | Identifiers | Additional information |
|---|---|---|---|---|
| chemical compound, drug | Fetal Bovine Serum (FBS) | Gibco/Thermo Fisher | Thermo Fisher/Gibco: 10270–106 | Experiments in *Figure 2b,c* |
| chemical compound, drug | Fura-4F | Invitrogen | Invitrogen: F14174 | |
| chemical compound, drug | Furaptra | Invitrogen | Invitrogen: M1290 | |
| chemical compound, drug | Glucose | Sigma-aldrich | Sigma-aldrich: G8270 | |
| chemical compound, drug | HEPES | Sigma-aldrich | Sigma-aldrich: H3375 | |
| chemical compound, drug | Insulin-transferrin-selenium-X | Invitrogen | Invitrogen: 51500056 | |
| chemical compound, drug | KCl | Sigma-aldrich | Sigma-aldrich: P5405 | |
| chemical compound, drug | L-Cysteine | Sigma-aldrich | Sigma-aldrich: C7352 | |
| chemical compound, drug | L-Glutamic acid | Sigma-aldrich | Sigma-aldrich: G1251 | |
| chemical compound, drug | Lipofectamin 2000 | Thermo Fisher | Thermo Fisher: 11668027 | |
| chemical compound, drug | Lipofectamin LTX | Thermo Fisher | Thermo Fisher: 15338100 | |
| chemical compound, drug | Opti-MEM I Reduced Serum Medium | Thermo Fisher | Thermo Fisher: 31985070 | |
| chemical compound, drug | Dulbecco's Modified Eagle Medium | Thermo Fisher | ThermoFirsher: 31966021 | |
| chemical compound, drug | Mg-ATP | Sigma-aldrich | Sigma-aldrich: A9187 | |
| chemical compound, drug | MgCl2 | Sigma-aldrich | Sigma-aldrich: 449172 | |
| chemical compound, drug | NaCl | Sigma-aldrich | Sigma-aldrich: S9888 | |
| chemical compound, drug | Na-GTP | Sigma-aldrich | Sigma-aldrich: G8877 | |
| chemical compound, drug | NaH2PO4 | Sigma-aldrich | Sigma-aldrich: S8282 | |
| chemical compound, drug | NPE | Synaptic Systems | SySy: 510 006 | |
| chemical compound, drug | Papain | Worthington Biochemical | Worthington Biochemical: LS003126 | |
| chemical compound, drug | Penicillin/streptomycin | Invitrogen | Invitrogen: 15140122 | |
| chemical compound, drug | Pluronic F-127 | Thermo Fisher | Thermo Fisher: P3000MP | |
| chemical compound, drug | cg-DAG | *Nadler et al., 2013* | PMID: 23720390 | |
| chemical compound, drug | caged SAG-PI(4,5)P2 | This paper | | European Molecular Biology Laboratory (EMBL), Cell Biology and Biophysics Unit, Meyerhofstr. 1, 69117 Heidelberg, Germany. Att: Carsten Schultz (schultz@embl.de) |
| chemical compound, drug | trypsin-inhibitor | Sigma-aldrich | Sigma-aldrich: T9253 | |
| chemical compound, drug | U73122 | Sigma-aldrich | Sigma-aldrich: U6756 | |
| chemical compound, drug | U73343 | Sigma-aldrich | Sigma-aldrich: U6881 | |
| software, algorithm | Igor Pro | Wavemetrics | | |
| | ImageJ version 1.50b | Waybe Rasband, National Institute of Health, USA | | |
| | SigmaPlot v. 12.3 | Systat Software Inc. | | |
| | Matlab | MathWorks | | |

## Chemical synthesis

### Synthesis of caged PI(4,5)P$_2$/AM 1a,b and 2a,b

#### General procedures

All chemicals from commercial sources (Acros, Sigma, Aldrich, VWR) were used as received without further purification. Dried solvents were also used as delivered. 3,6-Di-O-butyryl-1:2,5:6-di-O-isopropylidene-myo-inositol was obtained from SiChem GmbH, Bremen, Germany.

TLC was performed on precoated plates of silica gel (Merck, 60 F$_{254}$) using UV-light (254 or 366 nm) or a solution of phosphomolybdic acid in sulfuric acid (2.5 g phosphomolybdic acid, 1 g cerium (IV)sulfate and 6 mL concentrated sulfuric acid in 94 mL water). Preparative column chromatography was performed using silcal gel 60 from Macherey-Nagel, Germany (grain size 0.04–0.063 mm) with a pressure of 1–2 bar. Phosphoramidites were purified on silica deactivated with the eluent containing 10% triethylamine prior to use. Reverse phase column chromatography was performed using either Polygoprep 60–80 C18 from Macherey-Nagel, Germany or LiChroprep RP-18 (0.040–0.63 mm) from Merck.

HPLC was performed on a Knauer HPLC Smartline Pump 1000 using a Knauer Smartling UV Detector 2500 instrument. Unless stated otherwise, LiChrospher 100 RP-18, 10 µm partical size, 250 × 4 mm with LiChrospher 100 RP-18 precolumn, 5 µm particle size, 4 × 4 mm were employed for analytical HPLC at a flow rate of 1.5 mL/min. For semi-preparative HPLC a 250 × 10 mm LiChrospher 100 RP-18 column was used. Preparative HPLC was performed using a Knauer preparative pump K-1800 with K-2501 UV detector and a Merck Prebbar steel column, 250 × 50 mm, filled with LiChrospher 100 RP18, 12 µm material.

$^1$H-, $^{13}$C- and $^{31}$P-NMR-spectra were obtained on a 400 MHz Bruker UltraShield instrument. Chemical shifts were referenced indirectly to tetramethylsilane and $^{31}$P chemical shifts were referenced to 85% H$_3$PO$_4$. J values are given in Hz and chemical shifts were measured in ppm. Deuterated solvents were obtained from Deutero GmbH, Karlsruhe, Germany. Splitting patterns are designated as follows: s, singlet; d, doublet; t, triplet; q, quartet; m, multiplet; b, broad. $^{13}$C- and $^{31}$P-spectra were broadband proton decoupled.

Mass spectra (ESI) were recorded using a Waters Micromass ZQ mass spectrometer. High-resolution mass spectra were recorded at the University of Heidelberg on a HP ICR Apex-Qe mass spectrometer. Masses are given as m/z.

Melting points were determined on a Buechi B-540 and are uncorrected.

## Synthesis of head group 10a,b

**Chemical structure 1.** Synthesis of head group 10a,b.
DOI: https://doi.org/10.7554/eLife.30203.018

Reagents and conditions: (a) $CH_2Cl_2$:$HCO_2H$ 4:1, rt, 3 hr, 88%; (b) $(FmO)_2P$-$NiPr_2$ **7** (*Mentel et al., 2011*), 1*H*-tetrazole, $CH_2Cl_2$, rt, 1 hr, then $AcO_2H$, −80°C-rt, 1 hr, 83% over two steps; (c) (Coum)(FmO)P-$NiPr_2$ **8** (*Subramanian et al., 2010*), 1*H*-tetrazole, $CH_2Cl_2$, rt, 1 hr, then $AcO_2H$, −80°C-rt, 1 hr, 79%; (d) $CH_2Cl_2$:$HCO_2H$ 1:19, rt, 6 hr; (e) $Pr$-$C(OMe)_3$, $CH_2Cl_2$, JandaJel pyridinium trifluoroacetate, rt, 23 hr, 37.5% over five steps based on **3**.

## 3,6-Di-O-butyryl-1,2-O-isopropylidene-myo-inositol 5

3,6-Di-*O*-butyryl-1,2:4,5-di-*O*-*iso*propylidene-*myo*-inositol **3** (801 mg, 2 mmol) was dissolved in dichloromethane:formic acid (4:1, 16 mL) at 25°C with stirring. After 4 hr, the solution was diluted with dichloromethane (100 mL) and washed with phosphate buffer (pH 7, 150 mL). The pH of the aqueous phase was adjusted to 6–7 by the careful addition of saturated sodium bicarbonate solution (~95 mL). The aqueous layer was extracted twice with dichloromethane (2 × 100 mL), the pooled organic phases were dried ($Na_2SO_4$), filtrated and evaporated under reduced pressure. The solid residue obtained was dried at 0.2 mbar to give the title compound (633 mg, 87.8%) as a white solid.

$^{1}$H NMR (400 MHz, $CDCl_3$) δ = 5.10 (dd, *J* = 10.3, 7.7, 1H, ins H-6), 5.02 (dd, *J* = 10.1, 4.0, 1H, ins H-3), 4.47 (t, *J* = 4.4 Hz, 1H, ins H-2), 4.14 (dd, *J* = 7.6, 4.9 Hz, 1H, ins H-1), 4.01 (t, *J* = 9.7 Hz, 1H, ins H-4), 3.42 (t, *J* = 9.8 Hz, 1H, ins H-5), 2.76 (s, 1H, OH), 2.73 (s, 1H, OH), 2.43 (t, *J* = 7.4, 2 H, α-$CH_2$), 2.39 (t, *J* = 7.5 Hz, 2H, α-$CH_2$), 1.79–1.64 (m, 4H, 2 x β-$CH_2$), 1.56 (s, 3H, $CH_3$ ketal), 1.32 (s, 3 H, $CH_3$ ketal), 0.97 (t, *J* = 7.4, 3H, γ-$CH_3$), 0.96 (t, *J* = 7.4, 3 hr, γ-$CH_3$).

$^{13}$C NMR (101 MHz, CDCl$_3$) δ = 173.98, 173.66, 110.63, 76.47, 75.14, 73.82, 72.47, 70.99, 70.92, 36.16, 36.01, 27.79, 26.03, 18.46, 18.36, 13.52, 13.48.

$T_R$80% methanol = 2.2 min.

Mp108–110°C.

HR-MS (ESI positive) calculated C$_{17}$H$_{29}$O$_8$ m/z 361.18569, found 361.18588 [M + H]$^+$.Rosahl

## 3,6-Di-O-butyryl-4(5)-O-bis(9H-fluoren-9-ylmethyl)phosphoryl-1,2-O-isopropylidene-myo-inositol (mixture of 4-O- and 5-O- isomers with respect to the position of the caged phosphate) 6a,b

3,6-Di-O-butyryl-1,2-O-iso propylidene-myo-inositol 5 (900 mg, 2.5 mmol) is subsequently evaporated with acetonitrile (5 mL) and 1H-tetrazole solution in acetonitrile (11 mL, 5 mmol,~0.45 M). The remaining solids were suspended in anhydrous dichloromethane (15 mL) and a solution of bis-(9H-fluoren-9-ylmethyl)-N,N-diisopropylphosphoramidite 7 (1.25 g, 2.4 mmol) in dichloromethane (5 mL) was added. The mixture was stirred for 1 hr at 24°C. After cooling to −80°C (acetone/liquid nitrogen), peracetic acid solution (610 μL, 3.6 mmol, 39% in 45% acetic acid) was added. The cooling bath was removed and stirring continued for 1 hr. The solution was diluted with dichloromethane (50 mL) and poured into stirring phosphate buffer (pH 7, 200 mL). The pH was adjusted to neutral by the careful addition of saturated sodium bicarbonate solution. The organic layer was separated, washed with phosphate buffer (pH 7, 100 mL), dried (Na$_2$SO$_4$), filtrated and concentrated under reduced pressure to give 1.84 g of a white foam. The crude product was purified by chromatography on a column of silica gel 60 (20 × 3 cm) with 1. dichloromethane:cyclohexane 1:5 (300 mL), 2. 1:3 (100 mL), 3. 1:1, four ethyl acetate:methanol 9:1 (400 mL). A second chromatography with 1. dichloromethane:methanol 1:0 (1 L), 2. 98:2 (100 mL), 3. 96:4 (100 mL), 94:6 (100 mL), 92:8 (100 mL) afforded the title compound as white foam (1.58 g, 82.7%).

$T_R$100% methanol = 3.7 min.

$^1$H NMR (400 MHz, CDCl$_3$) δ = 7.82–7.12 (m, 16H), 5.21–5.13 (m, 0.5H, ins H), 5.13–5.02 (m, 1H, ins H), 4.92 (dd, J = 9.7, 3.9, 0.5H, ins H), 4.67–4.52 (m, 1H, ins H), 4.43 (t, J = 4.4, 0.5H), 4.37–3.92 (m, 7.5H), 3.85 (dd, J = 7.6, 2.4, 0.5H), 3.60 (t, J = 9.0, 0.5H), 3.18 (q, J = 9.2, 0.6H), 2.43–2.40 (m, 1H), 2.37 (t, J = 7.4, 1H), 2.23–2.02 (m, 2H, β-CH$_2$), 1.90–1.63 (m, 5H, β-CH$_2$), 1.63–1.39 (m, 5H, ketal CH$_3$), 1.36–1.17 (m, 4H, ketal CH$_3$), 1.02–0.92 (m, 3H, CH$_3$), 0.85–0.74 (m, 1.5H), 0.68 (t, J = 7.4, 1.5H).

$^{31}$P NMR (162 MHz, CDCl$_3$) δ = −0.49,−0.51.

## 3,6-Di-O-butyryl-4(5)-O-(7-diethylamino-2-oxo-2H-chromen-4-ylmethyl)-(9H-fluoren-4-ylmethyl)phosporyl-5(4)-O-bis(9H-fluoren-9-ylmethyl)phosphoryl-1,2-O-isopropylidene-myo-inositol (mixture of 4-O- and 5-O- isomers with respect to the position of the caged phosphate) 9a,b

Inositol 6a,b (1.52 g, 1.91 mmol) is evaporated with 1H-tetrazole solution in acetonitrile (12.54 mL, 5.73 mmol,~0.45 M). The solids were then evaporated two times with acetonitrile (2 × 2 mL). Anhydrous dichloromethane (8 mL) and a solution of (7-diethylamino-2-oxo-2H-chromen-4-ylmethyl)-(9H-fluoren-9-ylmethyl)-N,N-diisopropylaminophosphoramidite 8 (1.51 g, 2.6 mmol) in dichloromethane (15 mL) were subsequently added. The mixture was stirred for 2.5 hr, diluted with dichloromethane (20 mL) and cooled to −80°C. With stirring peracetic acid solution (727 μL, 4.29 mmol, 39% in 45% acetic acid) was added. The cooling bath is removed and the slightly yellow mixture was stirred for 1 hr. The reaction was then poured into phosphate buffer (200 mL, pH 7) and the pH was adjusted to neutral by the careful addition of saturated sodium bicarbonate solution. The organic layer was separated and the aqueous phase extracted two times with dichloromethane (2 × 50 mL). The combined extracts were dried (Na$_2$SO$_4$), filtrated and evaporated under reduced pressure to afford a slightly colored foam (2.8 g) that was purified by chromatography on column of silica gel 60 (120 mL, 18 × 4 cm) with 1. dichloromethane (200 mL), 2. ethyl acetate:cyclohexane 3:1 (200 mL), 3. ethyl acetate:cyclohexane 4:1 (300 mL) to afford the title compound as a light yellow foam (2.5 g).

$T_R$100% methanol = 3.7 min.

Part of the compound (100 mg) was further purified by semi-preparative HPLC (97.5% methanol) to give isomers P1 (12 mg) and P2 (60 mg, $t_R$ = 4.9 min, 97.5% methanol). A small amount of yellow oil (20 mg) that did not dissolve in 10 mL methanol for injection remained.

$^1$H NMR (400 MHz, CDCl$_3$) δ = 7.80–7.07 (m, 24H), 7.05–6.92 (m, 1H, coum H-5), 6.50–6.30 (m, 2H, coum H-8, H-6), 6.08–5.91 (m, 1H, coum H-3), 5.39–5.12 (m, 2H, ins H), 4.96–4.73 (m, 3H), 4.59–3.79 (m, 12H), 3.44–3.28 (m, 4H, 2xNCH$_2$), 2.36–2.07 (m, 4H, 2xα-CH$_2$), 1.62 (s, 3H, CH$_3$-ketal), 1.62–1.43 (m, 4H, 2xβ-CH$_2$), 1.31 (s, 3H, CH$_3$-ketal), 1.22–1.10 (m, 6H, 2xNCH$_2$C$\underline{H}_3$), 0.92–0.76 (m, 6H, 2xCH$_3$).

$^{31}$P NMR (162 MHz, CDCl$_3$) δ = −1.66 − −1.87 (0.6P), −1.87 − −2.15 (1.4P).

HR-MS (ESI positive) calculated C$_{73}$H$_{75}$KNO$_{16}$P$_2$ m/z 1322.41927, found 1322.42088 [M + K]$^+$.

## 3,6-Di-O-butyryl-4(5)-O-(7-diethylamino-2-oxo-2H-chromen-4-ylmethyl)-(9H-fluoren-4-ylmethyl)phosphoryl-5(4)-O-bis(9H-fluoren-9-ylmethyl)phosphoryl-myo-inositol (mixture of 4-O- and 5-O-isomers) 4a,b

Ketal **9a,b** (2.5 g, 1.95 mmol) is treated with dichloromethane (750 µL) and formic acid (16 mL, 424 mmol). The yellow solution is stirred at 25°C for 6 hr. The reaction was then poured into phosphate buffer (from 300 mL buffer pH 7 and 200 mL 1 M K$_2$HPO$_4$ solution) and extracted with ethyl acetate (3 × 100 mL). The combined extracts were dried (Na$_2$SO$_4$), filtrated and evaporated under reduced pressure to afford a yellow foam (2.2 g). The crude compound is dried overnight at 0.2 mbar.

$^1$H NMR (400 MHz, CDCl$_3$) δ = 7.79–7.02 (m, 24H), 7.03–6.95 (m, 0.37H, coum H-5, 5-cage), 6.95–6.88 (m, 0.59H, coum H-5, 4-cage, dia-1/2), 6.55–6.28 (m, 2H, coum H-8, H-6), 5.96–5.86 (m, 1H, coum H-3), 5.43–5.25 (m, 1H, ins H-6), 5.05–4.57 (m, 5H, ins H-3, H-4), 4.56–3.85 (m, 10H, ins H-5, H-2), 3.63–3.55 (m, 0.53H, ins H-1, 5-cage), 3.55–3.47 (m, 0.47H, ins H-1, 4-cage), 3.45–3.26 (m, 4H, 2xNCH$_2$), 2.35–2.05 (m, 4H, 2xα-CH$_2$), 1.65–1.38 (m, 4H, 4xβ-CH$_2$), 1.24–1.09 (m, 6H, 2xNCH$_2$C$\underline{H}_3$), 0.89–0.73 (m, 6H, 2xCH$_3$).

$^{31}$P NMR (162 MHz, CDCl$_3$) δ = −2.08 − −2.27 (m, 0.6P), −2.27 − −2.47 (m, 0.8P), −2.47 − −2.63 (m, 0.6P).

HR-MS (ESI positive) calculated C$_{70}$H$_{71}$NNaO$_{16}$P$_2$ m/z 1266.41403, found 1266.41571 [M + Na]$^+$.

## 2,3,6-Tri-O-butyryl-4(5)-O-(7-diethylamino-2-oxo-2H-chromen-4-ylmethyl)-(9H-fluoren-4-ylmethyl)phosphoryl-5(4)-O-bis(9H-fluoren-9-ylmethyl)phosphoryl-myo-inositol (mixture of 4-O- and 5-O- isomers with respect to the position of the caged phosphate) 10a,b

1. Preparation of catalyst: JandaJel-poly(pyridine) resin (3.1 g, 24.8 mmol) is swollen in dichloromethane (50 mL) and treated with trifluoroacetic acid (20 mL, 261 mmol). The slurry was allowed to stand overnight. The liquid was removed by filtration, the resin washed with dichloromethane (100 mL), dichloromethane:methanol 1:1 (100 mL), dichloromethane (100 mL) and dried at 0.2 mbar for 4 hr.

2. Reaction: to the crude diol 4a,b (2.2 g, 1.95 mmol), the above resin, activated molecular sieve 4 Å (10 g), anhydrous dichloromethane (40 mL) and trimethyl orthobutyrate (11 mL, 68.7 mmol) were added in succession. The flask was sealed and shaken on an orbital shaker for 23 hr. Analytical HPLC indicated complete conversion of the diol (97.5% methanol, $t_R$P1 = 2.7 min, $t_R$P2 = 3.5 min) into the cyclic intermediate (97.5% methanol, $t_R$P1 = 3.8 min, $t_R$P2 = 5.4 min). The mixture was filtered, the solids washed with dichloromethane and the yellow filtrate was concentrated under reduced pressure. Acetonitrile:water 1:1 (100 mL) and DOWEX 50W × 8 ion exchange resin (6 g, H$^+$) were added and stirred for 4 hr at 25°C. HPLC indicated complete conversion of the intermediate to the mono butyrate ($t_R$P1 = 3 min, $t_R$P2 = 4 min). The resin was removed by filtration, washed with acetonitrile (50 mL) and the filtrate was evaporated under reduced pressure to afford a yellow foam (2 g) that was purified in three runs by preparative HPLC using isocratic 90% acetonitrile to give the title compound as a yellow solid (1.3 g, 37.5% over five steps).

$^1$H NMR (400 MHz, CDCl$_3$) δ = 7.80–7.03 (m, 24H), 7.0–6.85 (m, 1H, coum H-5), 6.53–6.24 (m, 2H, coum H-8, H-6), 5.99–5.93 (0.6H, 4-cage, dia-1/2), 5.97 (s, coum H-3, dia-2, 4-cage), 5,96 (s, coum H-3, dia-1, 4-cage), 5.86 (s, 0.4H, coum H-3, 5-cage), 5.63–5.50 (m, 1H, ins H-2), 5.46–5.23 (m, 1H, ins H-6,), 5.09–4.93 (m, 1H, ins H-3), 4.92–4.66 (m, 4.8H, ins H-4), 4.65–4.22 (m, 6.2H, ins H-5), 4.22–

3.88 (m, 6H, ins H-5, dia-1), 3.87–3.78 (m, 0.4H, ins H-1, dia-2), 3.78–3.69 (m, 0.6H, ins H-1, dia-1), 3.43–3.26 (m, 4H, 2xNCH$_2$), 2.49–2.02 (m, 6H, 3xα-CH$_2$), 1.73–1.33 (m, 6H, 3xβ-CH$_2$), 1.21–1.07 (m, 6H, 2xNCH$_2$C<u>H$_3$</u>), 0.97 (t, J = 7.6, 1H, CH$_3$, 5-cage), 0.94–0.84 (m, 2H, CH$_3$, 4-cage, dia-1/2), 0.84–0.74 (m, 6H, 2xCH$_3$).

$^{31}$P NMR (162 MHz, CDCl$_3$) δ = −2.02 − −2.10 (m, 0.33P), −2.10 − −2.29 (m, 1.22P), −2.29 − −2.43 (m, 0.45 P).

$^{13}$C NMR (101 MHz, CDCl$_3$) δ = 173.89, 173.87, 173.82, 172.99, 172.92, 172.89, 172.75, 172.69, 172.62, 161.58, 161.52, 161.50, 156.14, 156.10, 156.05, 150.53, 150.49, 150.41, 150.37, 149.21, 149.18, 149.12, 149.10, 149.03, 148.99, 148.94, 148.91, 143.19, 143.17, 143.06, 143.01, 142.97, 142.92, 142.89, 142.87, 142.84, 142.76, 142.73, 142.70, 142.61, 142.55, 142.51, 141.52, 141.43, 141.40, 141.33, 141.28, 141.24, 141.21, 141.19, 141.13, 128.13, 128.07, 128.02, 127.93, 127.91, 127.88, 127.84, 127.79, 127.36, 127.22, 127.13, 127.04, 127.00, 125.40, 125.33, 125.28, 125.23, 125.14, 125.05, 125.02, 124.95, 124.89, 124.54, 124.43, 120.23, 120.14, 120.11, 120.02, 119.98, 119.92, 108.65, 108.64, 108.61, 108.58, 108.55, 106.40, 106.34, 106.33, 106.17, 97.80, 97.76, 76.65, 76.61, 76.56, 76.44, 76.39, 76.34, 76.13, 76.07, 71.85, 71.78, 71.74, 70.16, 70.11, 70.06, 70.01, 69.92, 69.90, 69.83, 69.81, 69.75, 69.71, 69.65, 69.56, 69.48, 69.43, 69.24, 69.19, 69.05, 68.21, 68.19, 64.93, 64.88, 64.77, 64.73, 64.62, 64.58, 64.48, 64.44, 47.99, 47.92, 47.88, 47.84, 47.79, 47.76, 47.70, 44.80, 44.70, 35.95, 35.91, 35.89, 35.85, 35.81, 35.68, 35.61, 35.60, 35.55, 29.73, 18.53, 18.49, 18.46, 18.07, 18.03, 17.79, 17.75, 13.58, 13.53, 13.49, 12.41.

HR-MS (ESI positive) calculated C$_{74}$H$_{77}$NNaO$_{17}$P$_2$ m/z 1336.45589, found 1336.45569 [M + Na]$^+$.

## Synthesis of 1a,b

**Chemical structure 2.** Synthesis of 1a,b.
DOI: https://doi.org/10.7554/eLife.30203.019

Reagents and conditions: (f) (dioctanoylglycerol)(OFm)P-N*i*Pr$_2$ **11**(*Subramanian et al., 2010*), 1*H*-tetrazole, CH$_2$Cl$_2$, rt, 1 hr, then AcO$_2$H, −80°C-rt, 1 hr, 67% over two steps; (g) CH$_2$Cl$_2$, EtNMe$_2$, rt, 30

min; (h) acetoxymethyl bromide, *N,N*-di*iso*propylethylamine, acetonitrile, rt, 22 hr, 65% over two steps;

## 2,3,6-Tri-O-butyryl-4(5)-O-(7-diethylamino-2-oxo-2H-chromen-4-ylmethyl)-(9H-fluoren-4-ylmethyl)phosphoryl-5(4)-O-bis(9H-fluoren-9-ylmethyl)phosphoryl-1-O-(9H-fluoren-9-ylmethyl)-(1',2'-di-O-octanoyl-sn-glycero)phosphoryl-myo-inositol (mixture of 4-O- and 5-O- isomers with respect to the position of the caged phosphate) 12a,b

Mono alcohol **10a,b** (445 mg, 0.34 mmol) in 1*H*-tetrazole solution in acetonitrile (~0.45 M, 3 mL, 1.37 mmol) was evaporated under reduced pressure. The solid obtained was again evaporated with acetonitrile (2 mL) to dryness. The residue was suspended in anhydrous dichloromethane (5 mL) and neat 3-*O*-(9H-fluoren-4-ylmethyl)−1,2-di-*O*-octanoyl-*sn*-glycero-*N,N*-di*iso*propylphosphoramidite **11** (450 μL,~0.67 mmol) was added. After stirring for 1 hr at 20°C the mixture was cooled in a liquid N$_2$/acetone bath. Peracetic acid solution (39% in 45% acetic acid, 240 μL, 1.4 mmol,) was added and the cooling bath was removed. After 1 hr the solution was concentrated under reduced pressure. The residue obtained was twice evaporated with toluene (2 × 20 mL), washed two times with water (2 × 20 mL) and dried in high vacuum for 2 hr.

The residue was purified by chromatography on a column of silica gel 60 with a stepwise gradient of cyclohexane:ethyl acetate 4:1 (400 mL), 3:1 (400 mL), 2:1 (200 mL), 3:2 (400 mL), 1:1 (200 mL) to give the compound as a yellow foam (428 mg, 66.6%).

T$_R$100% methanol = 13 min.

R$_f$ cyclohexane:ethyl acetate 1:1 = 0.57.

$^1$H NMR (400 MHz, CDCl$_3$) δ = 7.94–7.03 (m, 32H), 7.02–6.86 (m, 1H, coum H-5), 6.55–6.40 (m, 1H, coum H-8), 6.40–6.27 (m, 1H, coum H-6), 6.01–5.93 (m, 0.66H, coum H-3), 5.90–5.84 (m, 0.34H, coum H-3), 5.64–5.54 (m, 1H, ins H-4), 5.50–5.26 (m, 1H, ins H-6), 5.22–5.09 (m, 1H, H-sn2), 5.06–4.60 (m, 4H), 4.60–3.80 (m, 18H), 3.75–3.50 (m, 0.6H), 3.45–3.24 (m, 4H, 2xNCH$_2$), 2.45–1.90 (m, 10H, 5xα-CH$_2$), 1.70–1.40 (m, 10H, 5xβ-CH$_2$), 1.35–1.08 (m, 22H, 10xCH$_2$, 2xNCH$_2$C<u>H</u>$_3$), 0.98–0.71 (m, 14H), 0.68–0.50 (m, 1H).

$^{31}$P NMR (162 MHz, CDCl$_3$) δ = −1.35 − −1.60 (m, 1P), −1.65 − −1.80 (m, 2P), −1.94 (s, 0.6P), −1.95 − −2.14 (m, 2P), −2.14 − −3.39 (m, 2P).

$^{13}$C NMR (101 MHz, CDCl$_3$) δ = 173.18, 173.15, 172.83, 172.80, 172.75, 172.72, 172.70, 172.68, 172.37, 172.35, 172.27, 172.23, 172.18, 172.11, 172.10, 161.38, 161.34, 156.16, 156.11, 156.05, 149.04, 148.98, 148.95, 143.26, 143.16, 143.14, 143.07, 143.05, 143.01, 142.98, 142.96, 142.91, 142.89, 142.87, 142.86, 142.85, 142.80, 142.77, 142.65, 142.61, 142.58, 142.52, 142.50, 142.47, 141.63, 141.59, 141.56, 141.50, 141.47, 141.45, 141.43, 141.41, 141.35, 141.31, 141.26, 141.24, 141.22, 141.21, 141.10, 128.25, 128.19, 128.14, 128.07, 128.01, 127.93, 127.90, 127.86, 127.83, 127.80, 127.78, 127.38, 127.37, 127.32, 127.24, 127.20, 127.17, 127.13, 127.03, 126.97, 126.96, 125.46, 125.40, 125.35, 125.30, 125.24, 125.22, 125.17, 125.08, 125.05, 125.00, 124.96, 124.92, 124.90, 124.79, 124.76, 124.48, 124.42, 120.50, 120.40, 120.31, 120.29, 120.21, 120.20, 120.17, 120.06, 120.03, 119.98, 119.94, 108.70, 108.68, 108.66, 108.64, 106.35, 106.34, 106.33, 97.92, 97.90, 97.88, 77.46, 77.14, 76.82, 75.92, 75.87, 75.83, 75.78, 75.73, 75.58, 72.84, 72.79, 70.20, 69.86, 69.82, 69.66, 69.60, 69.56, 69.53, 69.51, 69.47, 69.41, 69.35, 69.30, 69.26, 69.19, 69.12, 68.25, 68.20, 68.15, 68.10, 66.01, 65.99, 65.93, 65.80, 65.76, 64.79, 64.78, 64.62, 64.60, 64.43, 64.39, 61.53, 61.51, 48.02, 47.97, 47.90, 47.85, 47.80, 47.70, 47.61, 44.88, 35.87, 35.85, 35.80, 35.73, 35.69, 35.68, 35.61, 35.59, 35.56, 35.51, 35.39, 35.27, 34.03, 33.96, 31.67, 29.07, 29.01, 28.93, 26.93, 24.82, 24.77, 22.61, 18.59, 18.56, 18.53, 18.49, 18.45, 18.43, 18.41, 17.87, 17.84, 17.81, 17.62, 17.58, 17.44, 14.09, 13.68, 13.65, 13.64, 13.57, 13.53, 13.51, 13.46, 13.44, 12.39.

HR-MS (ESI positive) calculated C$_{107}$H$_{122}$NNaO$_{24}$P$_3$ m/z 1920.74618, found 1920.74599 [M + Na]$^+$.

## 2,3,6-Tri-O-butyryl-4(5)-O-(7-diethylamino-2-oxo-2H-chromen-4-ylmethyl) phosphoryl-5(4)-O-phosphoryl-1-O-(1',2'-di-O-octanoyl-sn-glycero) phosphoryl-myo-inositol tetrakis(acetoxymethyl) ester (mixture of 4-O- and 5-O- isomers with respect to the position of the caged phosphate) 1a,b

Purified **12a,b** (66 mg, 35 µmol) was dissolved in acetonitrile (2 mL) and dimethylethylamine (1 mL, 9.2 mmol) under argon. After 30 min of stirring at 20°C all volatiles were removed at 0.3 mbar for 1 hr. Under argon atmosphere, anhydrous acetonitrile (1 mL), *N,N*-di*iso*propylethylamine (145 µL, 834 µmol) and acetoxymethyl bromide (54.5 µL, 556 µmol) were subsequently added. The flask was tightly sealed and the mixture was stirred overnight at 20°C protected from light. After 22 hr all volatiles were removed at 0.3 mbar. The residue was twice evaporated with toluene (2 × 3 mL) and subjected to semi-preparative HPLC using 92% methanol as eluent. The last peak ($t_R$ = 12 min) was collected and the solvent was evaporated under reduced pressure. The residue (39.1 mg) was extracted three times with water (3 × 1 mL). The remaining oil was dried at 0.3 mbar to yield caged PI(4,5)P$_2$/AM **1a,b** as a yellow oil (33.2 mg, 64.8%).

$T_R$90% methanol = 13 min.

$^1$H NMR (400 MHz, CDCl$_3$) δ = 7.36–7.20 (m, 1H, coum H-5), 6.62–6.53 (m, 1H, coum H-6), 6.51–6.43 (m, 1H, coum H-8), 6.24–6.18 (m, 0.6H, coum H-3), 6.18–6.11 (m, 0.4H, coum H-3), 5.77–5.42 (m, 10H), 5.38–4.90 (m, 4H), 4.90–4.72 (m, 1H), 4.65–4.47 (m, 2H), 4.40–3.99 (m, 4H), 3.47–3.30 (m, 4H), 3.48–3.31 (m, 4H, 2xNCH$_2$), 2.49–2.16 (m, 10H, 5xα-CH$_2$), 2.16–1.90 (m, 12H, 4xCOCH$_3$), 1.77–1.47 (m, 10H, 5xβ-CH$_2$), 1.33–1.20 (m, 16H, 8xCH$_2$), 1.18 (t, *J* = 7.0, 3H, 2xNCH$_2$C$\underline{H}_3$), 1.01–0.75 (m, 15H, 5xCH$_3$).

HR-MS (ESI positive) calculated C$_{63}$H$_{98}$NNaO$_{32}$P$_3$ m/z 1496.51770, found 1496.51780 [M + Na]$^+$.

## Synthesis of 2a,b

**Chemical structure 3.** Synthesis of 2a,b.

DOI: https://doi.org/10.7554/eLife.30203.020

Reagents and conditions, A: (f) (2-O-arachidonyl-1-O-stearoylglycerol)(OFm)P-N*i*Pr$_2$**14**, 1*H*-tetrazole, CH$_2$Cl$_2$, rt, 1 hr, then AcO$_2$H, −80°C-rt, 1 hr, 89%; (g) CH$_2$Cl$_2$, EtNMe$_2$, rt, 30 min; (h) acetoxymethyl bromide, *N,N*-di*iso*propylethylamine, MeCN, rt, 22 hr, 43% over two steps; for **19** 30% over two steps; B: (i) 2-arachidonyl-3-stearoylglycerol **16** (*Nadler et al., 2013*), 1*H*-tetrazole, CH$_2$Cl$_2$, 2 hr, 0–21°C, 94%.

## 3-O-(9H-Fluoren-9-ylmethyl)-1-O-stearoyl-2-O-arachidonyl-sn-glycero-N,N-diisopropylphosphoramidite 14

2-*O*-Arachidonyl-1-*O*-stearoyl-*sn*-glycerol **16** (415 mg, 0.64 mmol) and 1*H*-tetrazole solution in acetonitrile (~0.45 M, 1.43 mL, 0.64 mmol) were evaporated at 0.4 mbar for 1 hr. Under argon atmosphere a solution of phosphoramidite **13** (275 mg, 0.64 mmol) in anhydrous dichloromethane (5 mL) was added. The suspension was cooled in an ice bath. After 30 min the cooling bath was removed and stirring was continued at 20°C. After 2 hr the mixture was diluted with cyclohexane (10 mL) and concentrated under reduced pressure. The residue was purified by chromatography on column of silica gel 60 (100 mL, 9 × 4 cm) with cyclohexane:ethyl acetate:triethylamine 92:7:1. Individual fractions we analyzed by TLC (deactivated silica).

Yield: 584 mg (93.5%) colorless oil, R$_f$ cyclohexane:ethyl acetate:triethylamine 92:7:1 = 0.88.

Purity: ~96% (NMR).

$^1$H NMR (400 MHz, CDCl$_3$) δ = 7.81–7.59 (m, 4H), 7.45–7.26 (m, 4H), 5.49–5.26 (m, 8H), 5.24–5.15 (m, 1H, CH-*sn2*), 4.41–4.31 (m, 1H), 4.22–4.13 (m, 2H), 4.05–3.96 (m, 1H), 3.85–3.53 (m, 5H), 2.87–2.74 (m, 6H), 2.33–2.26 (m, 4H), 2.13–2.01 (m, 4H), 1.73–1.53 (m, 4H), 1.41–1.21 (m, 34H), 1.15 (dd, *J* = 14.3, 6.9, 12H, 4xNCHC$\underline{H}_3$), 0.89 (t, *J* = 7.2, 3H, CH$_3$), 0.88 (t, *J* = 6.8, 3H, CH$_3$).

$^{31}$P NMR (162 MHz, CDCl$_3$) δ = 148.25 (s, 0.5P, P$_{dia1/2}$), 148.17 (s, 0.5P, P$_{dia1/2}$).

$^{13}$C NMR (101 MHz, CDCl$_3$) δ = 173.40, 172.75, 144.88, 144.87, 144.51, 144.50, 141.37, 141.27, 130.50, 128.89, 128.87, 128.74, 128.60, 128.26, 128.15, 127.87, 127.73, 127.55, 127.46, 127.42, 127.36, 127.05, 126.97, 126.90, 126.89, 126.84, 125.44, 125.20, 125.16, 119.98, 119.85, 119.79, 70.96, 70.90, 70.88, 66.24, 66.18, 66.07, 66.01, 62.48, 62.44, 61.68, 61.51, 61.48, 61.31, 49.22, 49.16, 49.09, 43.12, 43.00, 34.14, 33.73, 31.95, 31.54, 29.73, 29.69, 29.67, 29.51, 29.39, 29.35, 29.32, 29.17, 27.24, 26.93, 26.53, 25.65, 25.63, 25.62, 24.92, 24.88, 24.79, 24.70, 24.63, 24.59, 24.52, 22.72, 22.60, 14.15, 14.10.

HR-MS (ESI positive) calculated C$_{61}$H$_{96}$NNaO$_7$P m/z 1008.68224, found 1008.68592 [M + Na + O]$^+$.

## 2,3,6-Tri-O-butyryl-4(5)-O-(7-diethylamino-2-oxo-2H-chromen-4-ylmethyl)-(9H-fluoren-4-ylmethyl)phosphoryl-5(4)-O-bis(9H-fluoren-9-ylmethyl)phosphoryl-1-O-(9H-fluoren-9-ylmethyl)-(1'-O-stearoyl-2'-O-arachidonyl-sn-glycero)phosphoryl-myo-inositol (mixture of 4-O- and 5-O- isomers with respect to the position of the caged phosphate) 15a,b

Head group **10a,b** (198 mg, 151 µmol) was evaporated with 1*H*-tetrazole in acetonitrile solution (1.5 mL, 675 µmol) to dryness at <0.3 mbar. Under argon atmosphere, anhydrous dichloromethane (5 mL) and neat phosphoramidite **14** (152 mg, 157 µmol) were added with stirring at 21°C. After 2 hr the mixture was diluted with dichloromethane (15 mL), cooled in a dry ice/acetone bath and peracetic acid solution (39% in 45% acetic acid, 35 µL, 206 µmol) was added. The cooling bath was removed and the mixture was allowed to come to room temperature. After 45 min the oxidation was quenched by stirring with aqueous ascorbic acid solution (0.2 M, 0.5 mL) and phosphate buffer (pH 7, 50 mL). The organic phase was separated, washed with water (50 mL), dried (Na$_2$SO$_4$), filtered and evaporated under reduced pressure. The remaining oil (350 mg) was purified by chromatography on a column of LiChroprep RP18 (14 × 3 cm) with methanol (1.500 mL). The yellow band on top of the column was eluted with dichloromethane (200 mL). The solvent was removed under reduced pressure to afford the compound.

Yield: 295 mg (89.1%) yellow oil.

$^1$H NMR (400 MHz, CDCl$_3$) δ = 7.95–6.85 (m, 32H), 7.01–6.85 (m, 1H, H-5 coumarin), 6.52–6.26 (m, 2H), 6.02–5.93 (m, 0.64H), 5.90–5.82 (m, 0.36H), 5.66–5.51 (m, 1H), 5.50–5.25 (m, 9H), 5.23–5.09

(m, 1H), 5.06–3.82 (m, 22H), 3.43–3.25 (m, 4H, 2xNCH$_2$), 2.90–2.74 (m, 6H), 2.34–1.97 (m, 14H), 1.74–1.18 (m, 44H), 1.15 (t, J = 7.6, 6 hr, 2xNCH$_2$C$\underline{H}_3$), 0.99–0.57 (m, 15H).

$^{31}$P NMR (162 MHz, CDCl$_3$) δ = −1.17,–1.42, -1.48,–1.81, −1.83,–1.92, −2.06,–2.18, −2.32,–2.37, −2.52,–2.79.

$^{13}$C NMR (101 MHz, CDCl$_3$) δ = 173.17, 173.14, 172.83, 172.81, 172.76, 172.47, 172.43, 172.37, 172.35, 172.27, 172.23, 172.18, 172.11, 172.10, 161.40, 161.36, 156.18, 156.13, 156.07, 149.05, 148.96, 143.27, 143.17, 143.15, 143.14, 143.07, 143.06, 143.02, 142.99, 142.97, 142.91, 142.87, 142.86, 142.80, 142.77, 142.64, 142.59, 142.52, 142.50, 142.47, 141.64, 141.60, 141.56, 141.54, 141.51, 141.46, 141.42, 141.40, 141.35, 141.32, 141.26, 141.24, 141.21, 141.10, 130.49, 129.03, 129.00, 128.95, 128.88, 128.82, 128.78, 128.77, 128.72, 128.61, 128.29, 128.20, 128.12, 128.10, 128.07, 128.00, 127.93, 127.89, 127.85, 127.79, 127.78, 127.76, 127.55, 127.39, 127.33, 127.29, 127.24, 127.19, 127.17, 127.12, 127.03, 126.97, 126.95, 125.46, 125.40, 125.36, 125.30, 125.21, 125.17, 125.07, 125.05, 125.01, 124.95, 124.90, 124.83, 124.80, 124.78, 124.47, 124.41, 120.51, 120.41, 120.30, 120.20, 120.16, 120.15, 120.05, 120.03, 119.98, 119.94, 108.58, 108.53, 108.43, 106.25, 97.81, 97.77, 97.75, 75.90, 75.87, 75.82, 75.77, 75.65, 75.57, 72.85, 72.79, 70.20, 69.85, 69.81, 69.66, 69.61, 69.56, 69.51, 69.47, 69.31, 69.25, 68.24, 68.18, 68.15, 68.10, 66.02, 65.99, 65.95, 65.89, 65.75, 65.71, 64.42, 61.52, 61.48, 48.03, 47.97, 47.90, 47.79, 47.71, 44.79, 35.87, 35.85, 35.80, 35.78, 35.73, 35.71, 35.68, 35.61, 35.60, 35.56, 35.50, 35.40, 35.28, 33.95, 33.43, 31.95, 31.53, 29.73, 29.70, 29.69, 29.67, 29.53, 29.39, 29.34, 29.32, 29.16, 27.23, 26.46, 25.66, 25.63, 25.61, 24.83, 24.65, 22.72, 22.60, 18.60, 18.57, 18.54, 18.49, 18.46, 18.43, 18.42, 17.88, 17.85, 17.81, 17.63, 17.59, 17.44, 17.43, 14.17, 14.11, 13.69, 13.66, 13.57, 13.54, 13.52, 13.49, 13.46, 13.44, 12.41.

HR-MS (ESI positive) calculated C$_{129}$H$_{159}$NNO$_{24}$P$_3$ m/z 2199.04594, found 2199.04847 [M + H]$^+$.

## 2,3,6-Tri-O-butyryl-4(5)-O-(7-diethylamino-2-oxo-2H-chromen-4-ylmethyl) phosphoryl-5(4)-O-phosphoryl-1-O-(1'-O-stearoyl-2'-O-arachidonyl-sn-glycero)phosphoryl-myo-inositol tetrakis(acetoxymethyl) ester (mixture of 4-O- and 5-O- isomers with respect to the position of the caged phosphate) 2a,b

In a 50 mL pear shaped flask **15a,b** (160 mg, 72.7 µmol) was treated with acetonitrile (3 mL) and dimethylethylamine (3 mL, 27.7 mmol). After 30 min volatiles were removed under reduced pressure. The slightly colored oil obtained was dried at 0.3 mbar for 1 hr. Anhydorous acetonitrile (2 mL), di*i-so*propylethylamine (442 µL, 2.52 mmol) and acetoxymethyl bromide (165 µL, 1.68 mmol) were subsequently added under argon atmosphere. The flask was sealed and the mixture was stirred in the dark at 21°C for 22 hr. The mixture was diluted with acetonitrile (10 mL) and evaporated under reduced pressure. The yellow residue obtained was suspended in acetonitrile, filtrated, concentrated under reduced pressure and purified by preparative HPLC (100% methanol).

Yield: 55 mg (42.6%) colorless oil, t$_R$ (100% methanol)=12.8 min.

$^1$H NMR (400 MHz, CDCl$_3$) δ = 7.35–7.22 (m, 1H), 6.65–6.53 (m, 1H), 6.49 (s, 1H), 6.22 (s, 0.56H), 6.17 (d, J = 6.4, 0.44H), 5.80–5.45 (m, 10H), 5.44–5.12 (m, 11H), 5.12–5.00 (m, 1H), 4.90–4.74 (m, 1H), 5.67–4.46 (m, 2H), 4.34–4.00 (m, 4H), 3.40 (q, J = 7.0, 4H, 2xNCH$_2$), 2.88–2.73 (m, 6H), 2.54–2.19 (m, 11H), 2.18–1.99 (m, 16H), 1.76–1.48 (m, 12H), 1.39–1.20 (m, 32H), 1.19 (t, J = 7.0, 6 hr, 2xNCH$_2$C$\underline{H}_3$), 1.03–0.79 (m, 14H).

$^{31}$P NMR (162 MHz, CDCl$_3$) δ = −2.74–3.28 (m, 1P), −3.28 − −3.78 (m, 0,44P), −3.88 − −4.30 (s, 0.47P), −5.549 − −5.23 (m, 1P).

$^{13}$C NMR (101 MHz, CDCl$_3$) δ = 173.24, 173.18, 172.69, 172.68, 172.53, 172.51, 172.11, 172.06, 172.04, 169.19, 169.14, 169.05, 161.70, 156.25, 156.20, 130.47, 128.95, 128.91, 128.81, 128.76, 128.59, 128.27, 128.08, 127.82, 127.51, 124.48, 124.46, 124.45, 124.38, 124.36, 124.32, 108.77, 106.29, 106.07, 97.79, 97.77, 83.19, 83.10, 83.05, 82.99, 82.94, 82.91, 82.87, 82.83, 82.76, 82.73, 82.68, 82.62, 77.38, 76.56, 76.53, 76.15, 76.13, 76.09, 76.05, 73.10, 73.06, 69.20, 69.13, 68.18, 68.11, 67.99, 67.94, 67.91, 67.82, 66.23, 66.18, 66.10, 66.05, 65.41, 65.36, 65.32, 61.41, 53.75, 44.77, 35.83, 35.60, 35.57, 35.51, 35.46, 33.95, 33.93, 33.45, 33.43, 31.91, 31.50, 29.69, 29.64, 29.49, 29.35, 29.30, 29.29, 29.13, 27.19, 26.46, 25.61, 25.59, 24.80, 24.64, 22.68, 22.56, 20.58, 20.55, 18.62, 18.55, 17.86, 17.83, 17.80, 17.70, 17.66, 17.65, 17.35, 14.11, 14.06, 13.63, 13.57, 13.53, 13.51, 13.49, 12.40.

HR-MS (ESI positive) calculated $C_{85}H_{135}NNaO_{32}P_3$ m/z 1774.81746, found 1774.82789 $[M + H]^+$.
**19** was also isolated.

Yield: 35 mg (29.7%) colorless oil, $t_R$ (100% methanol)=11 min

$^1H$ NMR (400 MHz, $CDCl_3$) δ = 5.78–5.43 (m, 12H), 5.42–5.25 (m, 8H), 5.24–5.12 (m, 1H), 5.10–4.98 (m, 1H), 8.84–4.71 (m, 1H), 4.64–4.43 (m, 2H), 4.32–4.00 (m, 4H), 2.90–2.70 (m, 6H), 2.55–2.18 (m, 11H), 2.19–1.98 (m, 18H), 1.77–1.48 (m, 10H), 1.41–1.15 (m, 34H), 1.02–0.78 (m, 15H).

$^{31}P$ NMR (162 MHz, $CDCl_3$) δ = −3.02 (s, 0.5P, dia-1),−3.58 (s, 0.5P, dia-2),−4.85 – 5.15 (m, 2P).

$^{13}C$ NMR (101 MHz, $CDCl_3$) δ = 173.23, 173.17, 172.68, 172.67, 172.53, 172.51, 172.09, 172.04, 172.00, 169.35, 169.25, 169.17, 169.15, 169.05, 130.46, 128.94, 128.90, 128.81, 128.76, 128.57, 128.26, 128.08, 127.81, 127.50, 83.05, 83.00, 82.95, 82.82, 82.77, 82.68, 82.63, 76.50, 76.48, 76.03, 75.99, 75.94, 73.09, 73.04, 69.18, 69.11, 68.93, 68.89, 68.15, 67.98, 67.96, 67.78, 66.23, 66.17, 66.09, 66.03, 61.40, 35.84, 35.56, 35.43, 35.40, 33.93, 33.92, 33.44, 33.42, 31.90, 31.49, 29.68, 29.64, 29.48, 29.34, 29.29, 29.28, 29.12, 27.19, 26.45, 25.60, 25.57, 24.79, 24.63, 22.67, 22.55, 20.61, 20.59, 20.57, 18.54, 17.82, 17.79, 17.69, 17.63, 14.10, 14.05, 13.62, 13.56, 13.52, 13.48.

HR-MS (ESI positive) calculated $C_{74}H_{123}NaO_{32}P_3$ m/z 1639.71025, found 1639.71082 $[M + Na]^+$.

## Structure determination of 4- and 5-isomers

**Chemical structure 4.** Structure determination of 4- and 5-isomers.
DOI: https://doi.org/10.7554/eLife.30203.021

Reagents and conditions: (a) $(FmO)_2P\text{-}NiPr_2$ **7**, 1H-tetrazole, $CH_2Cl_2$, rt, 1 hr; then $AcO_2H$, −80°C–rt, 1 hr, separation of isomers; (b) $(CoumO)(FmO)P\text{-}NiPr_2$ **8**, 1H-tetrazole, $CH_2Cl_2$, rt, 1 hr, then $AcO_2H$, −80°C-rt, 1 hr, 75.2% over two steps; (c) $CH_2Cl_2$:$HCO_2H$ 1:16, rt, 0.5 hr, 85.5%; (d) $CH_2Cl_2$:$HCO_2H$ 1:19, rt, 2 hr, 89.6%; (e) poly(4-vinylpyridine)/TFA, molecular sieve 4 Å, $CH_2CH_2$, $n$-PrC(OMe)$_3$, rt, 23 hr.

## 3,6-Di-O-butyryl-4-O-bis(9H-fluoren-9-ylmethyl)phosphoryl-1,2-O-isopropylidene-myo-inositol 6a

Crude **6a,b** (1.3 g, mixture of 4- and 5-phosphorylated isomers, prepared as above from 668 mg, 1.85 mmol diol **5**) was purified by four runs on a column of silica gel 60 (4 × 26 cm) using ethyl acetate:cyclohexane (32:68) as eluent. Fraction containing the pure 4-isomer ($R_f$ = 0.23, eluent) were pooled and evaporated under reduced pressure to give the title compound (200 mg, 13.5%).

$^1H$ NMR (400 MHz, $CDCl_3$) δ = 7.84–7.18 (m, 16H), 5.08 (dd, $J$ = 10.0, 7.8, 1H), 4.94 (dd, $J$ = 9.7, 3.9, 1H), 4.60–4.48 (m, 1H), 4.35 (t, $J$ = 4.4, 1H), 4.33–4.20 (m, 2H), 4.15–4.04 (m, 2H), 4.03–3.93 (m, 2H), 3.86 (dd, $J$ = 7.7, 4.9, 1H), 3.22 (dd, $J$ = 17.5, 9.0, 1H), 2.44 (dt, $J$ = 7.6, 2.8, 2H), 1.91–1.56 (m, 6H), 1.52 (s, 3H, $CH_3$-ketal), 1.31 (s, 3H, $CH_3$-ketal), 0.98 (t, $J$ = 7.4, 3H, γ-$CH_3$), 0.69 (t, $J$ = 7.4, 3H, γ-$CH_3$).

$^{31}P$ NMR (162 MHz, $CDCl_3$) δ = −0.48 (s, 1P).

$^{13}$C NMR (101 MHz, CDCl$_3$) δ = 173.30, 172.43, 143.17, 142.98, 142.76, 142.55, 141.81, 141.51, 141.34, 127.95, 127.89, 127.53, 127.24, 127.18, 127.12, 125.32, 125.24, 125.19, 125.16, 125.03, 124.46, 120.09, 120.04, 119.99, 110.83, 79.04, 78.98, 76.08, 73.44, 71.85, 71.79, 70.46, 70.02, 69.96, 69.62, 68.79, 68.74, 48.04, 47.96, 47.72, 47.64, 36.03, 35.28, 27.57, 25.91, 22.71, 18.50, 17.87, 14.15, 13.53, 13.48.

$T_R$ (Nucleodur 100–5 C18ec, 90% methanol) = 8.4 min.

### 3,6-Di-O-butyryl-5-O-bis(9H-fluoren-9-ylmethyl)phosphoryl-1,2-O-isopropylidene-myo-inositol 6b

From above purification, the 5-phosphorylated isomer (420 mg, 28.4.5%, $R_f$ = 0.29) was also isolated.

$T_R$ (Nucleodur 100–5 C18ec, 90% methanol)=7.9 min.

$^1$H NMR (400 MHz, CDCl$_3$) δ = 7.82–7.03 (m, 16H), 5.18 (dd, J = 9.3, 7.4, 1H), 5.12 (dd, J = 9.9, 4.0, 1H), 4.64 (dd, J = 17.5, 8.5, 1H), 4.44 (t, J = 4.5, 1H), 4.37–4.08 (m, 7H), 3.79 (bs, 1H, OH-5), 3.61 (t, J = 9.0, 1H), 2.38 (t, J = 7.4, 2H), 2.25–2.00 (m, 2H), 1.78–1.63 (m, 2H), 1.58 (s, 3H, CH$_3$-ketal), 1.56–1.45 (m, 2H), 1.32 (s, 3H, CH$_3$-ketal), 0.97 (t, J = 7.6, 3H), 0.79 (t, J = 7.6, 3H).

$^{31}$P NMR (162 MHz, CDCl$_3$) δ = −0.45 (s, 1P).

$^{13}$C NMR (101 MHz, CDCl$_3$) δ = 172.98, 172.88, 142.94, 142.93, 142.91, 142.84, 141.35, 141.34, 141.32, 128.01, 127.96, 127.92, 127.21, 127.19, 127.15, 127.12, 127.09, 125.23, 125.20, 125.18, 125.10, 120.07, 120.03, 110.79, 78.52, 78.46, 75.94, 74.01, 73.29, 72.03, 69.96, 69.90, 69.85, 69.20, 69.13, 47.82, 47.74, 36.13, 35.67, 27.55, 25.74, 18.44, 18.19, 13.59, 13.42.

Mp. 161–162°C.

HR-MS (ESI positive) calculated C$_{45}$H$_{49}$NaO$_{11}$P m/z 819.29047, found 819.29095 [M + Na]$^+$.

### 3,6-Di-O-butyryl-4,5-di-O,O-bis(9H-fluoren-9-ylmethyl)phosphoryl-1,2-O-isopropylidene-myo-inositol 15

From above purification, the 4,5-diphosphorylated derivative 15 (60 mg) was also isolated.

$^1$H NMR (400 MHz, CDCl$_3$) δ = 7.74–7.04 (m, 32H), 5.42–5.31 (m, 1H), 5.23 (dd, J = 9.5, 3.8, 1H), 4.92 (dd, J = 16.8, 9.1, 1H), 4.50–4.41 (m, 1H), 4.39–4.07 (m, 10H), 4.07–3.94 (m, 4H), 2.28 (td, J = 7.5, 4.6, 2H), 2.16 (td, J = 8.0, 4.0, 2H), 1.62–1.47 (m, 4H), 1.53 (s, 3H), 1.32 (s, 3H), 0.85 (t, J = 7.6, 3H), 0.83 (t, J = 7.6, 3H).

$^{31}$P NMR (162 MHz, CDCl$_3$) δ = −1.97 (s, 1P), −2.03 (s, 1P).

$^{13}$C NMR (101 MHz, CDCl$_3$) δ = 172.96, 172.33, 143.18, 143.03, 142.98, 142.88, 141.38, 141.29, 141.25, 127.87, 127.83, 127.76, 127.72, 127.14, 127.09, 127.05, 127.04, 125.46, 125.36, 125.29, 125.27, 125.23, 125.18, 125.05, 119.98, 119.97, 119.91, 119.87, 119.85, 110.89, 75.65, 75.59, 75.49, 72.98, 71.58, 69.74, 69.68, 69.62, 69.41, 69.34, 69.28, 68.79, 47.88, 47.82, 47.75, 35.79, 35.75, 27.13, 25.39, 18.12, 18.06, 13.56, 13.46.

$T_R$100% methanol = 5.0 min.

HR-MS (ESI positive) calculated C$_{73}$H$_{70}$NNaO$_{14}$P$_2$ m/z 1255.41330, found 1255.41416 [M + Na]$^+$.

### 3,6-Di-O-butyryl-4-O-(7-diethylamino-2-oxo-2H-chromen-4-ylmethyl)-(9H-fluoren-4-ylmethyl)phosporyl-5-O-bis(9H-fluoren-9-ylmethyl)phosphoryl-1,2-O-isopropylidene-myo-inositol 9b

Monophosphate 6b (350 mg, 0.44 mmol) and 1H-tetrazole solution in acetonitrile (0.45 M, 3.6 mL) was evaporated at 0.3 mbar. To the solid obtained was added a solution of phosphoramidite 8 in dichloromethane (12 mL) under an argon atmosphere with stirring at 21°C. After 1 hr the reaction was cooled in a dry ice/acetone bath and peracetic acid solution (39%, 170 µL, 1.0 mmol) was added. The cooling bath was removed and stirring continued at room temperature for 1 hr. The mixture was washed twice with phosphate buffer (pH7, 2 × 200 mL), and water (100 mL), dried (Na$_2$SO$_4$), filtrated and evaporated under reduced pressure to afford a yellow foam (641 mg). The crude compound was subjected to chromatography on a LiChroprep RP18 column (98 g) with a stepwise gradient of methanol:water 9:1 (2 L), 94:6 (0.5 L) and 96:4 (0.5 L).

Yield: 424 mg (75.2%).

$^1$H NMR (400 MHz, CDCl$_3$) δ = 7.80–7.07 (m, 24H), 7.03 (d, $J$ = 9.0, 0.6H, coum H-5, dia-1), 6.98 (d, $J$ = 8.9, 0.4H, dia-2), 6.44 (s, 1H, coum H-8), 6.42–6.30 (m, 1H, coum H-6), 6.00 (s, 1H, coum H-3), 5.35 (dd, $J$ = 15.0, 8.3, 1H, ins H-6), 5.27–5.18 (m, 1H, ins H-3), 4.97–4.75 (m, 3H, ins H-4), 4.61–3.92 (m, 12H, ins H-2, H-5, H-1), 3.35–3.25 (m, 4H, 2xNCH$_2$), 2.38–2.09 (m, 4H, 2xα-CH$_2$), 1.90–1.68 (m, 0H), 1.67–1.45 (m, 7H, 2xβ-CH$_2$, CH$_3$-ketal), 1.37–1.23 (m, 3H, ketal-CH$_3$), 1.23–1.08 (m, 6H, 2xNCH$_2$CH$_3$), 0.95–0.74 (m, 6H, 2xCH$_3$).

$^{31}$P NMR (162 MHz, CDCl$_3$) δ = −1.78 (s, 0.6P), −1.97 (s, 1P), −2.08 (s, 0.4P).

$^{13}$C NMR (101 MHz, CDCl$_3$) δ = 172.95, 172.92, 172.38, 172.33, 161.48, 156.12, 156.06, 149.21, 149.12, 148.91, 148.83, 143.13, 143.08, 143.03, 142.99, 142.94, 142.86, 142.78, 142.67, 141.52, 141.48, 141.40, 141.34, 141.28, 141.25, 141.21, 127.97, 127.89, 127.87, 127.84, 127.78, 127.27, 127.25, 127.19, 127.14, 127.06, 125.37, 125.35, 125.30, 125.25, 125.20, 125.17, 125.09, 125.03, 125.01, 124.60, 124.55, 120.06, 120.02, 119.98, 119.93, 119.90, 119.86, 110.93, 108.74, 108.72, 106.76, 106.41, 106.39, 77.58, 77.53, 77.50, 77.49, 75.70, 75.64, 75.59, 75.54, 75.41, 72.98, 71.55, 71.51, 69.77, 69.73, 69.68, 69.63, 69.51, 69.45, 68.71, 64.95, 64.90, 64.84, 64.80, 48.01, 47.97, 47.93, 47.90, 47.82, 47.75, 47.70, 44.89, 35.91, 35.73, 27.11, 27.08, 25.34, 25.30, 18.10, 13.55, 13.44, 12.39.

T$_R$ (Nucleodur 100–5 C18ec, 100% MeOH)=3.4 min.

HR-MS (ESI positive) calculated C$_{73}$H$_{75}$NNaO$_{16}$P$_2$ m/z 1306.44533, found 1306.44675 [M + Na]$^+$.

### 3,6-Di-O-butyryl-4-O-(7-diethylamino-2-oxo-2H-chromen-4-ylmethyl)-(9H-fluoren-4-ylmethyl)phosphoryl-5-O-bis(9H-fluoren-9-ylmethyl)phosphoryl-myo-inositol 4b

A solution of **9b** (340 mg, 0.26 mmol) in dichloromethane (1 mL) and formic acid (16 mL, 424 mmol) was stirred at 21°C. After 2 hr the solution is poured into a stirring mixture of phosphate buffer (pH 7, 200 mL) and ethyl acetate (50 mL). The yellow organic phase was separated, washed twice with phosphate buffer (2 × 200 mL), dried (Na$_2$SO$_4$), filtrated and evaporated under reduced pressure.

Yield: 295 mg, (89.6%) yellow film.

$^1$H NMR (400 MHz, CDCl$_3$) δ = 7.84–7.02 (m, 24H), 6.99–6.84 (m, 1H, coum H-5), 6.47–6.40 (m, 1H, coum H-8), 6.40–6.27 (m, 1H, coum H-6), 5.96 (s, 0.43H, coum H-3, dia-1), 5.91 (s, 0.57H, coum H-3, dia-2), 5.44 (t, $J$ = 9.6, 1H, ins H-6), 4.99–4.63 (m, 4H, ins H-4, H-3), 4.58–3.88 (m, 11H, ins H-5, H-2), 3.80–3.15 (bs, 2H, 2xOH), 3.64–3.56 (m, 1H, ins H-1), 3.39–3.26 (m, 4H, 2xNCH$_2$), 2.34–2.10 (m, 4H, 2xα-CH$_2$), 1.64–1.37 (m, 4H, 2xβ-CH$_2$), 1.19–1.08 (m, 6H, 2xNCH$_2$CH$_3$), 0.89–0.71 (m, 6H, 2xCH$_3$).data-p-fig-width

$^{31}$P NMR (162 MHz, CDCl$_3$) δ = −2.02,–2.18, -2.31,–2.42.

$^{13}$C NMR (101 MHz, CDCl$_3$) δ = 165.13, 164.88, 142.90, 142.75, 141.27, 128.05, 127.98, 127.87, 127.84, 127.77, 127.15, 127.04, 127.01, 125.33, 125.28, 125.20, 125.08, 125.02, 125.00, 124.95, 124.69, 120.75, 120.72, 120.09, 120.03, 119.99, 119.91, 119.87, 72.23, 72.20, 72.18, 70.28, 70.27, 70.24, 70.21, 69.56, 69.55, 69.51, 69.50, 48.03, 47.98, 47.90, 47.81, 47.73, 35.99, 35.91, 35.68, 35.64, 18.50, 18.12, 18.07, 18.02, 17.99, 13.52, 12.16, 12.09.

T$_R$ (Nucleodur 100–5 C18ec, 100% methanol)=2.6 min.

HR-MS (ESI positive) calculated C$_{70}$H$_{71}$NNaO$_{16}$P$_2$ m/z 1266.41403, found 1266.41622 [M + Na]$^+$.

### 3,6-Di-O-butyryl-5-O-bis(9H-fluoren-9-ylmethyl)phosphoryl-myo-inositol 17b

To a solution of **6b** (16.9 mg, 26.2 μmol) in dichloromethane (0.75 mL) formic acid (12 mL, 318 mmol) was added with stirring. After 10 min analytical HPLC indicated almost complete reaction. After 0.5 hr the solution is poured into a stirring mixture of phosphate buffer (pH 7, 200 mL) and ethyl acetate (50 mL). The organic phase was separated, washed with phosphate buffer (200 mL), dried (Na$_2$SO$_4$), filtrated and evaporated under reduced pressure. The crude compound was purified by semi-preparative HPLC in four runs.

Yield 13.6 mg (84.5%).

$^1$H NMR (400 MHz, CDCl$_3$) δ = 7.82–7.04 (m, 16H), 5.32 (t, $J$ = 9.8, 1H, ins H-6), 4.97–4.79 (m, 2H, ins H-3, H-5), 4.45–4.42 (m, 2H), 4.21–3.96 (m, 5H, ins H-2), 3.69–3.61 (m, 1H), 3.61–3.51 (m, 1H, ins H-4), 2.42 (td, $J$ = 7.3, 2.0, 2H), 2.27–1.95 (m, 2H), 1.75–1.61 (m, 2H), 1.58–1.36 (m, 2H), 0.96 (t, $J$ = 7.4, 3H), 0.72 (t, $J$ = 7.4, 3H).

$^{31}$P NMR (162 MHz, CDCl$_3$) δ = −1.18 (s, 1P).

$^{13}$C NMR (101 MHz, CDCl$_3$) δ = 175.05, 172.81, 143.09, 142.99, 142.92, 142.71, 141.33, 141.32, 141.26, 141.20, 128.01, 127.97, 127.82, 127.74, 127.72, 127.18, 127.10, 127.08, 126.95, 125.20, 125.09, 125.05, 120.01, 119.98, 119.94, 119.88, 119.83, 78.79, 78.73, 74.62, 72.33, 71.36, 71.31, 70.71, 70.28, 70.01, 69.95, 69.88, 69.82, 47.87, 47.80, 47.67, 47.60, 36.21, 35.59, 18.42, 18.04, 13.53, 13.50.

$T_R$ (100% methanol, Nucleodur 100–5 C18ec) = 1.9 min.

Mp 165–169°C.

HR-MS (ESI positive) calculated C$_{42}$H$_{45}$NaO$_{11}$P m/z 779.25917, found 779.25958 [M + Na]$^+$.

## 2,3,6-Tri-O-butyryl-4-O-(7-diethylamino-2-oxo-2H-chromen-4-ylmethyl)-(9H-fluoren-4-ylmethyl)phosphoryl-5-O-bis(9H-fluoren-9-ylmethyl)phosphoryl-myo-inositol 10b

1. Preparation of catalyst: poly(4-vinylpyridine) resin (3.2 g, 25.6 mmol) is swollen in dichloromethane (50 mL) and treated with trifluoroacetic acid (20 mL, 261 mmol). The slurry was shaken for 1 hr. The liquid was removed by filtration, the resin washed with dichloromethane (100 mL), dichloromethane:methanol 1:1 (100 mL), dichloromethane (100 mL) and dried in high vacuum for 4 hr.
2. Reaction: to diol 4b (250 mg, 0.2 mmol) in a 250 mL RBF, were subsequently added PVP/trifluoroacetic acid resin (1 g, 8 mmol), activated molecular sieve 4 Å (3 g), anhydrous dichloromethane (20 mL) and trimethyl orthobutyrate (3 mL, 18.7 mmol). The flask was tightly sealed and shaken on an orbital shaker for 23 hr. Analytical HPLC indicated complete conversion of the diol (100% methanol, $t_R$ = 2.6 min) into the cyclic intermediate ($t_R$ = 3.6 min). The mixture was diluted to a volume of 100 mL with dichloromethane, filtered and evaporated under reduced pressure to afford a greenish oil (264 mg). This oil was dissolved in acetonitrile:water 9:1 (100 mL) and DOWEX 50W × 8 ion exchange resin (5 g, H$^+$) was added and stirred for 3 hr at 20°C. The resin was removed by filtration, washed with acetonitrile (50 mL) and the filtrate was evaporated under reduced pressure to afford the crude compound as a yellow foam (254 mg). Analytical HPLC (95% methanol) indicated two peaks ($t_R$ = 6.9, 10b, dia-1 and 7.6 min, 10b, dia-2). TLC with ethylacetate:cylclohexane mixtures gave no separation.

The residue was subjected to chromatography on a column of LiChrospher 100 RP18 (98 g) with 1. 84% (1 L), 86% (2 L), and 90% methanol (2 L).

Yield: **10b, dia-1** 30 mg, **10b, dia-2** 86.7 mg.

**10b, dia-1**

$^1$H NMR (400 MHz, CDCl$_3$) δ = 7.80–7.02 (m, 24H), 6.92 (d, J = 9.0, 1H, coum H-5), 6.42 (d, J = 2.4, 1H, coum H-8), 6.38 (dd, J = 9.0, 2.4, 1H, coum H-6), 5.94 (s, 1H, coum H-3), 5.54 (t, J = 2.8, 1H, ins H-2), 5.31 (t, J = 10.0, 1H, ins H-6), 4.97 (dd, J = 10.4, 2.8, 1H, ins H-3), 4.84–4.76 (m, 1H, ins H-4), 4.84–4.76 (m, 2H), 4.61–4.46 (m, 2H), 4.35–4.26 (m, 1H), 4.20 (q, J = 8.8, 1H, ins H-5), 4.18–3.88 (m, 6H), 3.75 (dd, J = 10.0, 2.8, 1H, ins H-1), 3.34 (q, J = 7.1, 4H, 2xNCH$_2$), 2.42–2.29 (m, 2H, α-CH$_2$), 2.27 (t, J = 7.6, 2H, α-CH$_2$), 2.11 (t, J = 7.6, 2H, α-CH$_2$), 1.69–1.36 (m, 6H, β-CH$_2$), 1.16 (t, J = 7.1, 6H, 2xNCH$_2$CH$_3$), 0.91 (t, J = 7.4, 3H), 0.85 (t, J = 7.4, 3H), 0.76 (t, J = 7.4, 3H).

$^{31}$P NMR (162 MHz, CDCl$_3$) δ = −2.32 (s, 1P), −2.35 (s, 1P).

$^{13}$C NMR (101 MHz, CDCl$_3$) δ = 174.16, 172.76, 172.60, 143.17, 142.93, 142.88, 142.73, 142.66, 142.58, 141.56, 141.34, 141.29, 141.24, 141.21, 141.12, 128.04, 127.97, 127.92, 127.86, 127.84, 127.21, 127.20, 127.16, 127.12, 127.03, 126.98, 125.38, 125.28, 125.15, 125.00, 124.97, 124.90, 124.41, 120.10, 120.07, 120.06, 120.03, 119.97, 119.92, 106.50, 71.93, 70.00, 69.78, 69.72, 69.02, 68.63, 47.91, 47.76, 47.72, 47.68, 44.76, 35.89, 35.58, 18.49, 18.03, 17.74, 13.52, 13.47, 12.41.

Mp. 96–98°C.

HR-MS (ESI positive) calculated C$_{74}$H$_{77}$NNaO$_{17}$P$_2$ m/z 1336.45589, found 1336.45664 [M + Na]$^+$.

**10b, dia-2**

$^1$H NMR (400 MHz, CDCl$_3$) δ = 7.83–7.04 (m, 24H), 6.97 (d, J = 9.0, 1H, coum H-5), 6.43 (d, J = 2.0, 1H, coum H-8), 6.34 (d, J = 8.9, 1H, coum H-6), 5.95 (s, 1H, coum H-3), 5.57 (t, J = 2.8, 1H, ins H-2), 5.38 (t, J = 9.9, 1H, ins H-6), 5.02 (dd, J = 10.2, 2.8, 1H, ins H-3), 4.90–4.76 (m, 3H, ins H-4), 4.53–4.30 (m, 4H, ins H-5), 4.25–4.05 (m, 4H), 3.97 (t, J = 6.8, 2H), 3.81 (dd, J = 10.2, 2.8, 1H, ins H-1), 3.34 (q, J = 7.0, 4H, 2xN-CH$_2$), 2.73 (bs, 1H, OH), 2.45–2.31 (m, 2H, α-CH$_2$), 2.25–2.16 (m, 2H, α -CH$_2$), 2.09 (t, J = 7.6, 2H, α-CH$_2$), 1.70–1.34 (m, 6H, 3xβ-CH$_2$), 1.15 (t, J = 7.1, 6H, 2xNCH$_2$CH$_3$), 0.93 (t, J = 7.4, 3H), 0.84 (t, J = 7.4, 3H), 0.75 (t, J = 7.4, 3H).

$^{31}$P NMR (162 MHz, CDCl$_3$) δ = −2.02,−2.19.

$^{13}$C NMR (101 MHz, CDCl$_3$) δ = 173.98, 172.84, 172.64, 161.45, 156.08, 150.44, 149.10, 149.02, 143.20, 142.91, 142.84, 142.76, 142.74, 142.53, 141.42, 141.40, 141.33, 141.24, 141.22, 128.06, 128.02, 127.92, 127.90, 127.84, 127.77, 127.34, 127.28, 127.26, 127.22, 127.19, 127.13, 127.10, 127.01, 126.97, 125.36, 125.32, 125.23, 125.15, 125.10, 125.06, 124.99, 124.50, 120.12, 120.06, 120.01, 119.97, 119.94, 119.89, 108.50, 106.21, 97.73, 77.34, 77.29, 77.24, 76.07, 76.01, 75.96, 71.94, 70.09, 69.94, 69.88, 69.74, 69.69, 69.64, 69.13, 68.44, 64.78, 64.74, 47.92, 47.85, 47.78, 47.75, 47.70, 47.67, 44.77, 35.94, 35.85, 35.58, 18.51, 18.07, 17.74, 13.60, 13.55, 13.48, 12.42.

Mp. 103–105°C.

## Synthesis of 7-diethylamino-4-hydroxymethyl-2-oxo-2H-chromen 20

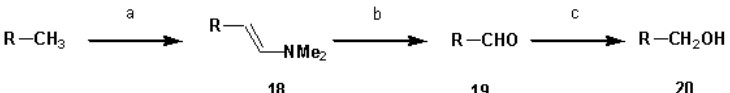

**Chemical structure 5.** Synthesis of 7-diethylamino-4-hydroxymethyl-2-oxo-2H-chromen 20.
DOI: https://doi.org/10.7554/eLife.30203.022

Conditions: (a) DMF-DMA, 140°C, 22 hr, 70%; (b) NaIO$_4$, THF:H$_2$O 1:1, 25°C, 2 hr, 97%; (c) NaBH$_4$, MeOH, 25°C, 60%. Abbreviations: DMF *N,N*-dimethylformamide, DMF-DMA dimethylformamide dimethylacetal, THF tetrahydrofuran, MeOH methanol, R 7-diethylamino-2-oxo-2*H*-chromen-4-yl.

### 7-Diethylamino-4-(2-dimethylamino-1-E-ethen-1-yl)−2-oxo-2H-chromen 18

7-Diethylamino-4-methyl-2-oxo-2*H*-chromen (11.56 g, 50 mmol) and *N,N*-dimethylformamide dimethylacetal (10 mL, 75 mmol) in anhydrous *N,N*-dimethylformamide (20 mL) were stirred at 140°C for 22 hr under argon atmosphere. After allowing to cool to room temperature, volatiles were removed under reduced pressure. The residue was triturated with cyclohexane (20 mL) and filtrated. The filter cake was suspended in acetone (50 mL), filtrated and washed with two portions of acetone (2 × 25 mL). The compound **18** was obtained as a yellow solid (10 g, 34.9 mmol, 69.8%).

$^1$H NMR (400 MHz, CDCl$_3$) δ = 7.52 (d, *J* = 9.1, 1H, H-5), 7.23 (d, *J* = 13, 1H, CH=), 6.54 (dd, *J* = 9.0, 2.6, 1H, H-6), 6.48 (d, *J* = 2.6, 1H, H-8), 5.85 (s, 1H, H-3), 5.21 (d, *J* = 13.0, 1H, CH=), 3.39 (q, *J* = 7.1, 4H, 2xNCH$_2$), 2.99 (s, 6H, 2xNCH$_2$C*H*$_3$), 1.19 (t, *J* = 7.1, 6H, 2xCH$_3$).

$^{13}$C APT NMR (100 MHz, CDCl$_3$) δ = 163.40, 156.33, 152.39, 150.08, 108.06, 44.61, 150.08, 124.89, 107.90, 97.95, 93.06, 87.31, 40.81 (br), 12.50.

R$_f$ dichloromethane:methanol 99:1 = 0.16
Mp 175–177.5°C
HR-MS (ESI positive) calculated C$_{17}$H$_{23}$N$_2$O$_2$ m/z 287.17540, found 287.17557 [M + H]$^+$.

### 7-Diethylamino-4-formyl-2-oxo-2H-chromen 19

Sodium (meta)periodate (22.4 g, 105 mmol) was added to a stirring suspension of **18** (10 g, 35 mmol) in tetrahydrofuran:water (1:1, 500 mL). After 2 hr, solids were removed by filtration and washed with ethylacetate (300 mL). The organic layer was separated, washed twice with saturated sodium bicarbonate solution (2 × 200 mL), dried (Na$_2$SO$_4$), filtrated and evaporated under reduced pressure. The black residue obtained was dissolved in dichloromethane and dried under high vacuum to afford compound **19** (8.3 g, 33.8 mmol, 96.9%).

$^1$H NMR (400 MHz, CDCl$_3$) δ = 10.02 (s, 1H, CHO), 8.29 (d, *J* = 9.2, 1H, H-5), 6.62 (dd, *J* = 9.2, 2.6, 1H, H-6), 6.51 (d, *J* = 2.6, 1H, H-8), 6.44 (s, 1H, H-3), 3.42 (q, *J* = 7.1 Hz, 4H, 2xCH$_2$), 1.21 (t, *J* = 7.1, 6H, 2xCH$_3$).

$^{13}$C NMR (101 MHz, CDCl$_3$) δ = 192.56, 161.94, 157.38, 151.01, 143.89, 127.04, 117.32, 109.56, 103.72, 97.62, 44.82, 12.44.

T$_R$70% methanol = 3.7 min.

### 7-Diethylamino-4-hydroxymethyl-2-oxo-2H-chromen 20

Sodium borohydride (5.55 g, 146.7 mmol) was added portion wise to a stirring solution of **19** in methanol (400 mL) over the course of 45 min. After 1.5 hr, 1 N HCl (150 mL) was added with stirring and the solution was concentrated under reduced pressure. The mixture was made alkaline by

addition of saturated sodium bicarbonate solution (~50 mL) and extracted three times with ethyl acetate (3 × 100 mL). The combined organic layers were dried ($Na_2SO_4$), filtrated and evaporated under reduced pressure. The dark tar obtained was purified by chromatography on a column of Poligoprep 60–80 RP18 (100 g) to give the pure compound as a tan solid (5 g, 20.3 mmol, 60%).

[1]H NMR (400 MHz, $CDCl_3$) δ = 7.31 (d, J = 9.0, 1H, H-5), 6.55 (dd, J = 9.0, 2.6, 1H, H-6), 6.48 (d, J = 2.6, 1H, H-8), 6.27 (s, 1H, H-3), 4.83 (s, 2H, $CH_2O$), 3.39 (q, J = 7.1, 4H), 2.66 (bs, 1H, OH), 1.19 (t, J = 7.1, 6H, $2xCH_3$).

[13]C NMR (101 MHz, $CDCl_3$) δ = 163.22, 155.97, 155.75, 150.48, 124.42, 108.70, 106.36, 105.01, 97.54, 77.42, 77.10, 76.79, 60.65, 44.68, 12.43.

$t_R$70% methanol = 3.3 min.

$R_f$ cyclohexane:acetone 3:1 = 0.22.

Mp 143–144°C (literature mp (*Eckardt et al., 2002*) 175–185°C).

## Plasmids

The following plasmids were generously given to us: Human mRFP-PH($PLC\delta_1$) from Ken Mackie (The Gill Center for Biomolecular Science, Bloomington, Indiana); GFP-PKD-C1ab from Tamas Balla (National Institutes of Health, Bethesda, MD), $M_1R$ from Neil Nathanson (University of Washington, Seattle, WA), Lifeact-RFP (pmRFPruby-N1*Lifeact (GB lab plasmid nr 28) in pmRFPruby-N1) from Geerd van den Gogaard (Radboud University Medical Center, Nijmegen, the Netherlands), and the $PLC\delta_4$-PH-mKate2 plasmid from Thomas F. J. Martin (Department of Biochemistry, University of Wisconsin) and the EGFP-PH-PH ($PLC\delta_1$) plasmid from Michael Krauss (Leibniz-Forschungsinstitut für Molekulare Pharmakologie, Berlin, Germany). The $PLC\delta_4$-PH was fused to an EGFP lentiviral plasmid under the control of CMV promotor and lentiviral particles were produced following standard protocols. HEK and COS-7 cells were transfected using Lipofectamine 2000 (Life Technologies) or Lipofectamine LTX (in the case of Lifeact-RFP, Thermo Fisher) according to manufacturer's protocol using a total amount of 1.8 μg of DNA per well (6-well plate) for HEK cells and 2 μg EGFP-PH-PH together with 1 μg mCherry-INPP5E-CAAX for COS-7 cells.

## PI(4,5)P$_2$ uncaging in vitro

For imaging of PI(4,5)P$_2$ on glass, cg-PI(4,5)P$_2$ was added to imaging buffer (HBSS with 5% FCS) to a final concentration of 20 μM. The high-affinity PI(4,5)P$_2$-sensor $PLC\delta_1$-PH-GFP was stored in a 1.8 mg/ml PBS/20% Glycerol stock. This was added 1:20 to the cg-PI(4,5)P$_2$ solution (e.g. 5 μl in 100 μl). The solution was pipetted onto a glass coverslip and imaged using a TIRF microscope (Nikon Ti Eclipse), equipped with an incubation chamber (37°C), a x60 TIRF objective (Apo TIRF 1.49NA, Nikon), a sCMOS camera (Neo, Andor), four excitation laser lines: (405,488 nm, 568 nm, 647 nm) an appropriate dicroic mirror (Di01-R405/488/561/635),filter (FF01-446/523/600/677). The TIRF microscope was operated by open-source ImageJ-based micromanager software (https://micro-manager.org/). Images were captured at 1 s intervals using a 488 nm laser (200 ms exposure) at 50% power (30 MW). Image analysis was performed with Fiji (ImageJ). Each 488 nm excitation frame was immediately followed by an uncaging frame, performed using a 405 nm laser (200 ms exposure) at 100% power (60 MW). ROIs of cg-PI(4,5)P$_2$ on glass were selected in the 405 nm channel and the fluorescence intensities of the $PLC\delta_1$-PH-EGFP sensor in the same ROIs in response to uncaging over time measured in the 488 nm channel

## Imaging of cellular cg-PI(4,5)P$_2$ uptake and uncaging (*Figure 2b and c*)

HEK 293T and COS-7 cells used for experiments depicted in *Figure 2b and c* were purchased from ATCC (https://www.lgcstandards-atcc.org); the identity of the cells has been confirmed by STR profiling performed by ATCC. Cell lines were tested for mycoplasma contaminations on a monthly basis. Cells were cultured in DMEM medium (Lonza) supplied with 10% fetal bovine serum (FBS, Gibco 10270–106) and 1% penicillin/streptomycin. Cells were not used beyond passage 30 from original. Preparation of cg-PI(4,5)P$_2$ was performed in the dark under red light. Loading solution was prepared by adding cg-PI(4,5)P$_2$ to imaging buffer (HBSS with 5% FCS) to a final concentration of 20 μM (from a 20 mM DMSO-stock). An equal volume of Pluronic F-127 (Thermo Fisher Scientific, 20% in DMSO) was added (final concentration: 0.02% Pluronic F-127). The final DMSO concentration was 0.2%. The loading solution was thoroughly vortexed for 3 min. Cell medium was removed from the

cells and cg-PI(4,5)P$_2$ loading solution was pipetted gently in at the edge of the well. Cells were incubated with the loading solution for 30 min in a CO$_2$-incubator at 37°C. Loading solution was removed and cells were gently washed twice with imaging buffer. Control loading solutions contained DMSO in place of cg-PI(4,5)P$_2$. CellMask Deep Red plasma membrane stain was stored in the dark at room temperature in a 5 mg/ml stock in DMSO (Thermo Fisher Scientific) was applied to HEK 293 T cells after loading with cg-PI(4,5)P$_2$. Cells were incubated in CellMask (1:1000 dilution of stock in imaging buffer) for 5 min. Cells were washed twice in imaging buffer and imaged immediately.

The experiments depicted in *Figure 2b* were performed on a Spinning Disk Confocal Microscope (Nikon TI-Eclipse) equipped with an incubation chamber (37°C), a x60 objective (P-Apo NA 1.40, Nikon), Yokogawa spinning disk (CSU-X1), an EMCCD camera (AU-888 Andor), four excitation laser lines: (405, 488 nm, 561 nm, 638 nm), an Borealis unit (Andor), an appropriate dicroic mirror (Di01-R405/488/561/635) and specific filter (BP450/50 and BP700/75 for coumarin and CellMask, respectively). The microscope was operated by NIS Elements (Nikon). Images were captured at 0.5 s intervals (200 ms exposure) using a 638 nm laser at 20% power (100 mW) and a 405 nm laser at 30% power (100 mW). Images were analysed with Fiji (ImageJ 1.50 g). Line profile ROIs used to investigate fluorescence intensities across the cell membrane were placed in the CellMask imaging channel (excitation 638 nm). ROIs were selected such that they crossed the plasma membrane from the extracellular space into nucleus-free cytosol at a 90° angle in relation to the visible cell membrane. In each frame, a 3 µm long sub-region of each line profile ROI was selected and aligned such that the mid-point of the line coincided with the position of the plasma membrane (recognized as a local maximum in the intensity value of the CellMask staining). This position was found using the second output parameter of the built-in MatLab function 'max' (MatLab vers. 7.12.0 R2011a). The intensity values along the line profile at 15 positions preceding and 15 positions succeeding the mid-point were read out. The exact same line positions were considered for the images containing the coumarin fluorescence (405 nm excitation). In both channels, the pixel intensity value of the 1 st position on each line (i.e. 1.5 µm extracellular to the plasma membrane) was subtracted from values at all other positions to obtain background subtracted line profiles. Line profiles were then averaged across cells.

For experiments depicted in *Figure 2c*, COS-7 cells were transfected with EGFP-PH-PH (PLCδ$_1$) and mChINPP5E-CAAX (*Posor et al., 2013*). Cells were loaded with cg-PI(4,5)P$_2$ and Pluronic F-127 as described above and imaged on the TIRF setup as described for the in vitro imaging. Images were captured at 1 s intervals with 200 ms exposure using a 488 nm laser at 50% power (30 MW), immediately followed by a 561 nm laser at 100% power (50 MW). Between the 10[th] and 11[th] loop (10–11 s), UV uncaging was performed with a single 400 ms exposure frame using a 405 nm laser at 100% power (60 MW). COS-7 cells expressing the constitutive phosphatase and lipid sensor were analysed by selecting circular ROIs of plasma membrane in the 488 nm channel only and measuring mean intensities over time. A ratio of fluorescence intensity in these ROIs was calculating by dividing intensities after the UV uncaging frame by the corresponding intensities prior to the UV uncaging frame.

## tsA201 cell culture and microscopy

tsA201 cells were purchased from Sigma-Adrich (St. Louis, MO, USA), and the identity of the cells has been confirmed by STR profiling performed by Sigma-Aldrich. The cells have been eradicated from mycoplasma at the European Collection of Cell Cultures (ECACC). Cells were cultured at 37°C and 5% CO$_2$ in DMEM-medium (Invitrogen Inc, Carlsbad, USA) supplemented with 10% FBS (PAA, Pasching, Austria) and 0.2% penicillin/streptomycin (Invitrogen Inc., USA). Transfection was performed with Lipofectamine 2000 (Invitrogen Inc., USA) according to the manufacturer's specifications. Cells were plated onto poly-D-lysine coated glass chips 16–20 hr before experiments. The tsA201 cell experiments were carried out at room temperature on a Zeiss LSM710 laser confocal microscope (Zeiss LLC, Thornwood, NY). Cells were superfused with Ringer's solution (160 mM NaCl, 2.5 mM KCl, 2 mM CaCl$_2$, 1 mM MgCl$_2$, 8 mM glucose, and 10 mM HEPES at pH 7.4) throughout the experiments. Uncaging of PI(4,5)P$_2$ was achieved on the microscope by a combined 5 s light pulse of both a 405 nm diode and a 451 nm laser line at 50% intensity of the light sources.

## Lifeact-RFP imaging (*Figure 3a*)

The Lifeact-RFP experiments depicted in *Figure 3a* were performed in HEK 293T cells provided by Dr Therese Schaub and Victor Tarabykin (Institute of Cell Biology and Neurobiology, Charité Berlin). These were cultured in DMEM GlutaMAX (Thermo Fisher/Gibco) supplied with 10% fetal bovine serum (FBS, Gibco 16140063) and 1% penicillin/streptomycin at 37°C in a humidified atmosphere (5% $CO_2$). Cells were not used beyond passage 40 from original. This cell line was not tested for mycoplasm contaminations. The cells density was between 0,25–1 $\times$ $10^6$ plated on 24 mm glass coverslips. 18–24 hr following transfection with Lifeact-RFP, loading solution was prepared by adding cg-PI(4,5)$P_2$ to culture medium removed from cells, to a final concentration of 20 μM (from a 20 mM DMSO-stock). An equal volume of Pluronic F-127 (Thermo Fisher Scientific, 20% in DMSO) was added (final concentration: 0.02% Pluronic F-127). The final DMSO concentration was 0.2%. The loading solution was thoroughly vortexed for 3 min. Loading was performed as described above (37°C, 30 min). Cells were washed twice and imaged in a solution containing (in mM) 145 NaCl, 3 KCl, 10 HEPES, 1 $CaCl_2$, 1 $MgCl_2$ and 6 Glucose at pH 7.4 and osmolarity, 290 mOsm/l.

Imaging was performed on a Nikon Ti eclipse TIRF microscope equipped with an incubation chamber (37°C), a x100 objective (Apo TIRF 1.49NA, Nikon), an EMCCD camera (iXon 888 Andor, EM gain set to 300), and suitable filtersets. Image acquisition was controlled by the Nikon NIS-Elements AR Software (vers. 4.51.01). Frames were collected at 2 Hz, images in the RFP channel were acquired by excitation with a 561 nm laser (2% intensity) and an exposure time of 100–200 ms. Following the acquisition of five frames in the RFP-channel, three consecutive UV frames were acquired at 2 Hz by excitation with a 405 nm laser at 25% laser intensity. Images during UV light were captured on the same camera with an exposure time of 100 ms. Imaging was then immediately resumed at 2 Hz in the RFP channel with the laser and camera settings mentioned above. Image analysis was performed offline in Fiji (ImageJ 1.50 g). Several equally sized circular ROIs were placed in the RFP images on filamentous structures presumed to be actin bundles (white circles in the left-hand images depicted in *Figure 3a*). The mean intensity value per ROI was calculated and corrected for background signal by subtraction of the mean intensity within one equally sized ROI placed in a background region outside the cell (yellow circle in the left-hand images depicted in *Figure 3b*). Background subtraction was performed in each frame. The intensity values of all ROIs within one cell were then averaged frame-wise and normalized by dividing the mean intensity values of all frames by that of the first frame. These normalized intensities were then averaged frame-wise across all investigated cells.

## Chromaffin cell culture and electrophysiology

Wildtype chromaffin cells were prepared as described previously (*Sørensen et al., 2003*) and used for experiments after 3–5 days. Cells were loaded with AM-ester coupled caged lipid compounds for varying durations. All lipid compounds were kept in 20–25 mM stock solutions in DMSO and stored at −20°C. Stock solutions were diluted in the cellular medium and Pluronic was added to facilitate uptake of the compound. The solution was heavily vortexed to avoid the generation of micelles before placing it onto the cells at a final lipid concentration of 20 μM with 0.02% Pluronic. Cells were kept in a $CO_2$-incubator at 37°C for 30 to 45 min. In order to document successful loading of the caged compounds, cells were checked after recordings for fluorescence levels.

For recordings, cells were transferred to a recording chamber and superfused with external recording solution containing (in mM): 145 NaCl, 2.8 KCl, 2 $CaCl_2$, 1 $MgCl_2$, 10 HEPES, 11.1 glucose, adjusted to pH 7.2 with NaOH. The solution had an osmolarity of approximately 305 mOsm. The patch pipette solution contained (in mM): 100 Cs-glutamate, 8 NaCl, 32 Cs-HEPES, 2 Mg-ATP, 0.3 NaGTP, one ascorbic acid, 0.4 Fura-4f (Invitrogen), 0.4 furaptra (Invitrogen), adjusted to pH 7.2 with CsOH. For DAG-uncaging experiment, the coumarin-caged DAG (cg-DAG) was loaded into the cells through the patch pipette for 60–100 s prior to stimulation. The patch pipette solution contained (in mM): 125 Cs-glutamate, 40 Cs-HEPES, 2 Mg-ATP, 0.3 NaGTP, 0.5 EGTA, 0.030 or 0.045 cg-DAG, adjusted to pH 7.2 with CsOH. The setup used for patching and uncaging of lipids consisted of a Zeiss inverted microscope (Axiovert 10) equipped with a specialized flash lamp (Rapp Optoelectronic, JML-C2). The light passed through a 395 nm low-pass filter, a light guide and a TILL Photonics dual port condensor before being focused on the sample through a Fluar 40X/N.A. 1.30 oil objective for maximal UV transmittance. For the composition of the filter cube, see below.

Cells were voltage clamped to −70 mV (liquid junction potential was not corrected for). After 1 min at rest, the cellular membrane was depolarized by stepping the voltage six times to +20 mV for 10 ms at 300 ms intervals followed by four 100 ms depolarizations at 400 ms intervals. The cell membrane capacitance was measured before, after, and in-between depolarizations(Voets et al., 1999). 8.5 s after the final depolarization, a strong flash of UV-light (1–2 ms duration, JML-C2, setting around 300V on the third capacitor bank) was triggered to uncage the lipid while the cellular capacitance was measured. Uncaging was repeated four times at the same power at 15 s intervals, after which another round of voltage depolarizations was initiated. All capacitance measurements were performed using the Lindau-Neher technique(Lindau and Neher, 1988). Amperometry measurements were performed with 5-µm-diameter polyethylene-insulated carbon fibers (Thornel P-650/42, Cytec [Bruns, 2004]). The voltage was clamped at 700 mV via an EPC-7 using an external power supply. Currents were filtered at 3 kHz and sampled at 12 kHz. For analysis, amperometric traces were filtered off-line at 1 kHz.

To quantify the IRP size, the cellular capacitance 2 s into the recording (after the last 10 ms depolarization pulse) was subtracted by the cellular capacitance at the beginning of the recording. The RRP size was quantified as the cellular capacitance at 2.8 s (after the second 100 ms depolarization) subtracted by the cellular capacitance at the beginning of the recording. The IRP size was subtracted from this value in each cell to isolate the release elicited by the first two 100 ms depolarizations (RRP-IRP). The total capacitance increase was measured 4 s after the beginning of the recording. From this the RRP size was subtracted in each cell to quantify the exocytosis increase caused by the last two 100 ms depolarizations (total-RRP). The step size elicited by the lipid-uncaging was measured in each cell by calculating the difference between the cellular capacitance 100 ms before and 300 ms after the UV-flash, which was elicited 500 ms after the beginning of the recording. Traces in Figure 4a were filtered with a binomial Gaussian filter using the 'smooth' function (window of 23 points) in IGOR Pro (vers. 6.22A) for clarity.

In experiments where $Ca^{2+}$ was uncaged the patch pipette solution contained the following (in mM) 100 Cs-glutamate, 8 NaCl, 4 $CaCl_2$, 32 HEPES, 2 Mg-ATP, 0.3 NaGTP, five nitrophenyl-EGTA, one ascorbic acid (to prevent photo damage to the $Ca^{2+}$-dyes), 0.4 fura-4f (Invitrogen), 0.4 furaptra (Invitrogen), adjusted to pH 7.2 with CsOH. $Ca^{2+}$ uncaging experiments and Ca2 +microfluorimetry were performed as described previously(Walter et al., 2014).

## Fluorescence quantification and live imaging in chromaffin cells

Loading of the caged compound was evaluated semi quantitatively, using the fluorescence of the compound. To quantify the fluorescence, cells were imaged on a Zeiss Axiovert 200 equipped with a TILL Monochromator V and a 25X/N.A. 0.8 LD LCI Plan-Apo oil/water/air objective with 405 nm excitation light and an EM-CCD camera (Andor 885, gain 1). Images were exposed for 500 ms. Fluorescence intensities were quantified using Image J software (version 1.46 r) by integrating the fluorescence in a square region (61 × 61 pixel) containing the cell, subtracted by the integrated intensity of the same size of background.

For live-cell imaging using lentivirally encoded low-affinity PI(4,5)$P_2$-sensor EGFP-PLCδ$_4$-PH, chromaffin cells were transduced with lentivirus 24 hr after seeding and allowed to express for 24–48 hr. Imaging was carried on a Zeiss Axiovert 200 equipped with a TILL Monochromator V and a flash lamp from Rapp Optoelectronic (JML-C2); both were coupled through the epifluorescence port using a 2-way splitter from TILL Photonics. An F-Fluar 40X/1.30 oil objective, and a CCD camera (PCO sensicam). To enable UV flashing and imaging of EGFP we used a dichroic mirror with efficient reflection from the near-UV range up to 488 nm (Chroma 495dcxru), together with a long-pass emission filter (Chroma et500 LP). Imaging was carried out using 488 nm excitation light and 80 ms exposure times. A single image was acquired prior to the first UV-flash(pre-flash) followed by subsequent images at 1 Hz. The second UV-flash was applied 38.5 s after the first flash.

To quantify the redistribution of PLCδ$_4$PH-EGFP the fluorescence background was removed from images by subtracting the mean intensity of the background. Unusually bright spots on the PM, visible on some of the images, were cut out and excluded from analysis. Integrated fluorescence density of a circular region of interest (ROI1) containing the entire cell was calculated. For each cell another, smaller, ROI – ROI2 – was defined as the interior of the cell, excluding the periphery. The content of the inner ROI was subtracted from the other ROI to isolate the intensity of the periphery of the cell (which includes the plasma membrane), and the ratio of periphery to inner ROI was calculated, i.e.

(ROI1-ROI2)/ROI2. This ratio was calculated as a function of time and normalized to the pre-flash ratio. These normalized values were then plotted as a function of time.

## Statistics

Results are shown as average ±s.e.m. unless otherwise indicated, with n referring to the number of cells for each group. Two-tailed paired or unpaired t-tests or Mann-Whitney U-test (if data were heteroscedastic) were used to compare between two groups, as indicated in figure legends. Significance was assumed when $p < 0.05$. Statistical testing was performed using SigmaPlot 12.3 (Systat Software Inc). In figures, the significance levels are indicated by asterisks; *$p < 0.05$; **$p < 0.01$; ***$p < 0.001$.

## Acknowledgements

The authors thank Anne Marie Nordvig Petersen for excellent technical assistance. We thank Geert van den Bogaart for the gift of the lifeact-RFP construct. This work was supported by the European Union Seventh Framework Programme under grant agreement HEALTH-F2-2009-242167 ('SynSys' project), the Danish Medical Research Council, the Novo Nordisk Foundation, Vera and Carl Johan Michaelsens grant (all JBS), the Lundbeckfoundation (JBS and PSP), the EMBL, Transregio83 of the DFG (to CS), Transregio186 (to CS, VH and AMW), the Emmy Noether Programme of the DFG (to AMW), the National Institutes of Health grant R37NS008174 and the Wayne E Crill Endowed Professorship (to BH), and an Alexander von Humboldt-Foundation fellowship (MK).

## Additional information

### Competing interests

Iwona Ziomkiewicz: Performed experiments as an employee of University of Copenhagen and is now an employee of AstraZeneca; IZ has no financial investments in AstraZeneca. The other authors declare that no competing interests exist.

### Funding

| Funder | Grant reference number | Author |
|---|---|---|
| Deutsche Forschungsgemeinschaft | Emmy Noether Programme | Alexander M Walter |
| Deutsche Forschungsgemeinschaft | Transregio186 | Alexander M Walter<br>Volker Haucke<br>Carsten Schultz |
| Lundbeckfonden | | Paulo S Pinheiro<br>Jakob Balslev Sørensen |
| Alexander von Humboldt-Stiftung | | Martin Kruse |
| National Institutes of Health | R37NS008174 | Bertil Hille |
| Wayne E Crill Endowed Professorship | | Bertil Hille |
| European Molecular Biology Laboratory | | Carsten Schultz |
| Deutsche Forschungsgemeinschaft | Transregio83 | Carsten Schultz |
| Seventh Framework Programme | HEALTH-F2-2009-242167 | Jakob Balslev Sørensen |
| Independent Research Fund Denmark | | Jakob Balslev Sørensen |
| Novo Nordisk Foundation | | Jakob Balslev Sørensen |

The funders had no role in study design, data collection and interpretation, or the decision to submit the work for publication.

## Author contributions
Alexander M Walter, Conceptualization, Data curation, Formal analysis, Supervision, Funding acquisition, Validation, Writing—original draft, Writing—review and editing,; Alexander M Walter, Rainer Müller, Bassam Tawfik, Investigation, Methodology; Keimpe DB Wierda, Resources, Validation, Investigation, Writing—review and editing; Paulo S Pinheiro, Anthony W McCarthy, Formal analysis, Validation, Investigation, Writing—review and editing; André Nadler, Gregor Reither, Formal analysis, Investigation, Writing—review and editing; Iwona Ziomkiewicz, Investigation, Methodology, Writing—review and editing; Martin Kruse, Validation, Investigation, Writing—review and editing; Jens Rettig, Resources, Supplied CAPS double knockout mice; Martin Lehmann, Resources, Investigation, Methodology; Volker Haucke, Resources, Funding acquisition; Bertil Hille, Conceptualization, Supervision, Funding acquisition, Writing—review and editing; Carsten Schultz, Conceptualization, Resources, Supervision, Funding acquisition, Validation, Methodology, Writing—review and editing; Jakob Balslev Sørensen, Conceptualization, Supervision, Funding acquisition, Writing—original draft, Project administration, Writing—review and editing

## Author ORCIDs
Alexander M Walter http://orcid.org/0000-0001-5646-4750
Rainer Müller http://orcid.org/0000-0003-3464-494X
Bassam Tawfik http://orcid.org/0000-0003-1193-8494
Keimpe DB Wierda http://orcid.org/0000-0002-8784-9490
Anthony W McCarthy http://orcid.org/0000-0002-3771-351X
Jakob Balslev Sørensen http://orcid.org/0000-0001-5465-3769

## Ethics
Animal experimentation: Permission to keep and breed knockout mice for this study was obtained from The Danish Animal Experiments Inspectorate (2006/562−43, 2012−15−2935−00001). The animals were maintained in an AAALAC-accredited stable in accordance with institutional guidelines as overseen by the Institutional Animal Care and Use Committee (IACUC).

## Decision letter and Author response
Decision letter https://doi.org/10.7554/eLife.30203.025
Author response https://doi.org/10.7554/eLife.30203.026

## Additional files
### Supplementary files
• Transparent reporting form
DOI: https://doi.org/10.7554/eLife.30203.023

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
