## [Decision Letter]

[Editors’ note: this article was originally rejected after discussions between the reviewers, but the authors were invited to resubmit after an appeal against the decision.]

Thank you for submitting your work entitled "Phosphatidylinositol 4,5-bisphosphate optical uncaging potentiates exocytosis" for consideration by *eLife*. Your article has been reviewed by three peer reviewers, one of whom Suzanne Pfeffer is a member of our Board of Reviewing Editors, and the evaluation has been overseen a Senior Editor.

Our decision has been reached after consultation between the reviewers. Based on these discussions and the individual reviews below, we regret to inform you that your work will not be considered further for publication in *eLife*.

This study uses impressive chemistry to create a caged, membrane-permeant PI(4,5)P_2_, which is then used to test the effect of a rapid PI(4,5)P_2_ increase on chromaffin cell exocytosis. Three observations are made for wildtype cell capacitance changes after initiating uncaging. First, there is an immediate burst of a small number of granules, next there is no change in the depolarization-induced immediately releasable pool (IRP), and finally there is significant augmentation of the depolarization-induced readily releasable pool (RRP). These observations imply that PI(4,5)P_2_ can, itself, stimulate a small amount of fusion and that it is a limiting factor controlling the priming for RRP. As might be expected, augmentation of the RRP requires Syt-1 and Munc13-2 but, surprisingly, did not require CAPS1/2.

After group discussion, it was the overall consensus of the referees that the impact of this study lies more in the development and use of a new tool than in a conceptual breakthrough. Previous studies reached similar conclusions using patch pipette delivery of PI(4,5)P_2_. Thus, the improvement of temporal resolution is small for the experiments on IRP and RRP. While uncaging might have the spatial resolution to achieve highly localized changes in PI(4,5)P_2_, the present work has the same limitation of previous work in which the released PI(4,5)P_2_ is present throughout the cell (and may even cause mislocalization of proteins with PI(4,5)P_2_ binding domains). In terms of the biology, two findings stand out: the lack of a role for CAPS1/2 and that PI(4,5)P_2_ is capable of stimulating fusion without Ca^2+^ influx. Interestingly, the latter depended on a dramatically reduced time of PI(4,5)P_2_ exposure (the 1st 1-2msec flash) so it highlights the power of the tool. Nevertheless, as the authors concede, the physiological situation is that exocytosis is triggered by a rise in calcium not a rise in PI(4,5)P_2_ so the result is likely a consequence of something already present in the literature: that PI(4,5)P_2_ may increase of Ca^2+^ affinity of syt-1. Reluctantly, our overall assessment is that the study is better suited to a more specialized journal despite its elegance and convincingness. The reviewers agreed that this will be of broad interest and deserves presentation in a good journal; however, they thought it fell short of what should be presented as an article in *eLife*.

We realize this news will be disappointing but we are confident that the work will be published quickly in another high quality journal.

Reviewer #1:

This is a high quality study that reports the synthesis and application of an optically caged PI(4,5)P_2_ to our understanding of chromaffin cell exocytosis. The authors show clearly a direct connection between Synaptotagmin 1 and Munc13-2 and PI(4,5)P_2_. The work appears to be carried out to a high standard and the following suggestions are offered to enhance the clarity of the story. The exocytosis experts will have more to add regarding the impact of these data on that field.

1) Please comment on the effect of the pluronic in cell viability and secretion efficiency; also, the compounds show strange localizations in cells–please comment. Also, please comment on the absolute capacity of the chromaffin cells to undergo exocytosis after taking up the compound compared with control cells. Finally, it would help the reader if the authors could highlight at the end of the discussion precisely what this probe brings that was not really known before, and other important applications that will benefit from its use.

2. Figure 2—figure supplement 1 could be a regular figure, as could a few others; please consider making your paper 6-8 regular figures and keeping the synthesis details in supplemental

Reviewer #2:

The study by Walter et al. uses impressive chemistry to create a caged, membrane-permeant PI(4,5)P_2_, which is then used to test the effect of a rapid PI(4,5)P_2_ increase on chromaffin cell exocytosis. Three observations are made for wildtype cell capacitance changes after initiating uncaging. First, there is an immediate burst of a small number of granules, next there is no change in the depolarization-induced immediately releasable pool (IRP), and finally there is significant augmentation of the depolarization-induced readily releasable pool (RRP). These observations imply that PI(4,5)P_2_ can, itself, stimulate a small amount of fusion and that it is a limiting factor controlling the priming for RRP. As might be expected, augmentation of the RRP requires Syt-1 and Munc13-2 but, surprisingly, did not require CAPS1/2.

The impact of this study lies more in the development and use of a new tool than in a conceptual breakthrough. In fact, previous studies reached similar conclusions using patch pipette delivery of PI(4,5)P_2_. Thus, the improvement of temporal resolution is small for the experiments on IRP and RRP (both methods appear to have on the order of 1 min lead times). While uncaging might have the spatial resolution to achieve highly localized changes in PI(4,5)P_2_, the present work has the same limitation of previous work in which the released PI(4,5)P_2_ is present throughout the cell (and may even cause mislocalization of proteins with PI(4,5)P_2_ binding domains). In terms of the biology, two findings stand out: the lack of a role for CAPS1/2 and that PI(4,5)P_2_ is capable of stimulating fusion without Ca^2+^ influx. Interestingly, the latter depended on a dramatically reduced time of PI(4,5)P_2_ exposure (the first 1-2msec flash) so it highlights the power of the tool. Nevertheless, as the authors concede, the physiological situation is that exocytosis is triggered by a rise in calcium not a rise in PI(4,5)P_2_ so the result is likely a consequence of something already present in the literature: that PI(4,5)P_2_ may increase of Ca^2+^ affinity of syt-1. Reluctantly, my overall assessment is that the study is better suited to a more specialized journal despite its elegance and convincingness.

Reviewer #3:

Walter et al. describe an interesting study on novel caged PPI_2_ reagents and their application in an analysis of the role of phospholipid signaling in the control of chromaffin granule fusion.

PPI_2_ signaling and lipid second messengers are of eminent importance to essentially all areas of cell biology. A general problem with analysing lipid signaling has been that methods to manipulate lipid second messengers have been intrinsically slow. In the context of PPI_2_ signaling, for instance, the activation of receptors coupled to downstream enzymes or the manipulation of the corresponding enzymes themselves (e.g. inhibition, activation, deletion, or ectopic localization) have mostly been used to modulate PPI_2_ signaling, typically within timeframes of minutes and longer. The caged PPI_2_ compounds described in the present study allow to manipulate PPI_2_ levels in the sub-second timeframe and hence allow in principle to dissect multiple different sub-steps of cell biological processes that are influenced by PPI_2_ signaling.

As far as I am concerned, the first introduction of a caged PPI_2_ compound that actually works is a substantial achievement that deserves prominent publication. The authors did a good job in showing that the caged PPI_2_ s work in principle. Clearly, a few open issues remain – e.g. (i) the question as to what the more or less homogeneous distribution of the caged compound in the cytosol after loading really means (Figure 2), (ii) how sure one can therefore be that it is actually plasma membrane PPI_2_ that causes the effects seen, or (iii) what sub-steps of the secretion process are exactly regulated by the PPI_2_ -Munc13-2 vs. PPI_2_ -CAPS interaction. However, I am willing to cut the authors some slack here, because I am of the opinion that the new caged PPI_2_ compounds and the proof-of-principle data of the present paper will be very interesting to scientists in all areas of cell biology. In fact, the new compounds might be at least as interesting to cell biologists as they may be to people working on neurons and neurosecretory cells.

[Editors’ note: what now follows is the decision letter after the authors submitted for further consideration.]

Thank you for choosing to send your work entitled "Phosphatidylinositol 4,5-bisphosphate optical uncaging potentiates exocytosis" for consideration at *eLife*. Your letter of appeal has been considered by a Senior Editor and a Reviewing editor, and we are prepared to consider a revised submission with no guarantees of acceptance. The revised manuscript should highlight-more clearly- the new biological findings that were not made adequately clear in the initial version. Please be sure to also respond to any of the comments you received in the revised version--in particular, is it at all possible to address whether PPI_2_ effects observed are due to plasma membrane PPI_2_ levels?

---

## [Author Response]

[Editors’ note: the authors’ appeal request in response to the first round of peer review follows.]

[…] The reviewers agreed that this will be of broad interest and deserves presentation in a good journal; however, they thought it fell short of what should be presented as an article in eLife. We realize this news will be disappointing but we are confident that the work will be published quickly in another high quality journal.

The reviewers regard our work as a “high quality study” that “reports the synthesis and application of an optically caged PI(4,5)P_2_”(reviewer #1) – a “substantial achievement” (reviewer #3) using “impressive chemistry” (reviewer #2). Our newly developed tool enabled us to study secretion in the same cell before and after uncaging, which we combined with genetic mutations to accurately decipher the involvement of specific PI(4,5)P_2_ - binding proteins. We use this to provide novel insights into molecular function of PI(4,5)P_2_ in stimulated exocytosis:

• We are the first to provide a genetic analysis showing that the increase in secretion caused by PI(4,5)P_2_ in living cells requires PAGE 2 OF 3 synatotagmin-1 and Munc13-2 (although reviewer #2 seems to have expected this result, this is the first experimental evidence).

• We are the first to show that CAPS-proteins are not involved in the PI(4,5)P_2_ -augmenting effect, a clearly important finding for all researchers studying regulated exocytosis, with mechanistic implications (see below).

• We identify a new, fast effect of PI(4,5)P_2_ to trigger release of a part of the Readily Releasable Pool of vesicles. This is the first time that a manipulation other than membrane depolarization or Ca^2+^ elevation was shown to trigger the release of secretory vesicles. These experiments establish that PI(4,5)P_2_ is more than a vesicle recruitment factor; it is actually involved in fusion itself.

Apart from the biological insights, all reviewers share a consensus view that the tool itself, caged, membrane permeable PI(4,5)P_2_, is a major methodological advance and we understand that the publication of such developments is within the scope of *eLife* (“we welcome the submission of significant technological or methodological advances”): “the work appears to be carried out to a high standard” (reviewer #1), “impressive chemistry” (reviewer #2), “the first introduction of a caged PI(4,5)P_2_ compound that actually works is a substantial achievement that deserves prominent publication”. The compound is “very interesting to scientists in all areas of cell biology” (reviewer #3). Reviewer #2 recognizes the value of our caged compound in experiments where we show that PI(4,5)P_2_ uncaging can trigger rapid vesicle fusion which “highlights the power of the tool” but then detracts from its importance by noting that in the cell a rapid increase in Ca^2+^, not PI(4,5)P_2_, will ordinarily trigger release. However, it is clear to any experimental biologist that the use of exogenous, even unnatural manipulations is key to understand biological function. For instance, Ca^2+^ uncaging experiments (which initially also received skeptical comments) trigger synaptic transmission by unnatural means, yet have profoundly added to our understanding of synapse function, with dozens of papers published in leading journals. Any new tool requires authors, reviewers and editors to leave their comfort zone. As this is particularly difficult, and because any introduction of new methodology requires experiments to validate it, a reflexive notion can tend to be that not much has been learned, or that existing methods would suffice. But this is clearly not the case here. In fact, our compound opens up new avenues for research: using PI(4,5)P_2_ uncaging it is now possible to distinguish between two different functions of PI(4,5)P_2_ -binding proteins: those activated by stoichiometric, obligatory, PI(4,5)P_2_ -binding (according to our data this will include synaptotagmin-1 and Munc13-2), and those operating to localize PI(4,5)P_2_ to the vesicle surroundings. Our experiments indicate that CAPS proteins function in the latter fashion, because their requirement for vesicle priming (which depends PAGE 3 OF 3 on their PH domain) could be bypassed by PI(4,5)P_2_ uncaging, which uncovers PI(4,5)P_2_ beneath the vesicle. Thus, application of our tool enabled us to make this distinction, which is a major finding. Likewise, the striking observation that PI(4,5)P_2_ can actually trigger exocytosis could not have been obtained by any other method. As described in our discussion, we agree with reviewer #2 that this may be due to PI(4,5)P_2_ -dependent increase in the Ca^2+^ affinity of synaptotagmin, which was observed in biochemical experiments. However, the biochemical experiments give no account for the kinetics of PI(4,5)P_2_-dependent activation. We here provide experimental data in living cells showing that this reaction operates on a comparable (millisecond) timescale as Ca^2+^ association, which provides novel biological insights into the vesicle fusion process.

Overall, the editorial decision is not understandable to us. We appreciate your supportive comment that we should have no problem publishing the manuscript in another high-quality journal, but we struggle to see how the additional time and work on the parts of authors, editors and reviewers is justified, when we in fact have already received a remarkably positive evaluation at *eLife*. Given the points made above, we respectfully ask you to reconsider your decision

[Editors’ note: the author responses to the re-review follow.]

Thank you for choosing to send your work entitled "Phosphatidylinositol 4,5-bisphosphate optical uncaging potentiates exocytosis" for consideration at eLife. Your letter of appeal has been considered by a Senior Editor and a Reviewing editor, and we are prepared to consider a revised submission with no guarantees of acceptance. The revised manuscript should highlight-more clearly- the new biological findings that were not made adequately clear in the initial version. Please be sure to also respond to any of the comments you received in the revised version--in particular, is it at all possible to address whether PPI_2_effects observed are due to plasma membrane PPI_2_levels?

We thank the reviewers and editors for their comments. We were happy to hear that the reviewers regard our work as a “high quality study” that “reports the synthesis and application of an optically caged PI(4,5)P_2_” (reviewer #1) – a “substantial achievement” (reviewer #3) using “impressive chemistry” (reviewer #2). In the revised version of our manuscript we have now carefully addressed all concerns raised by the reviewers and the editor. This was done by rewriting of the text, additional figure items and additional experiments. We would like to remain open in the discussion of the advantages and possible difficulties of our novel approach in the manuscript because it is the first time this compound is used in a cellular context.

New biological findings

The novel biological insights gained by our work were not stated clear enough. Now we provide these as bullet points in the Discussion section:

1) First genetic analysis showing that PI(4,5)P_2_ increases exocytosis via synatotagmin-1 and Munc13-2.

We identify a molecular requirement for PI(4,5)P_2_ regulation of exocytosis on these two proteins in genetic analyses. Our data are consistent with mutagenesis experiments that were predicted to interfere with protein lipid interaction in the cell (reviewer #2 also comments on this). However, these mutations are also expected to affect protein functions in additional ways (e.g. the same interface on the Synaptotagmin C2B domain was shown to interact with the neuronal SNAREs (Zhou et al., 2015)), which complicates the interpretation of the mutagenesis data. Here, by selectively studying mouse knockout cells for the relevant proteins, we now provide direct experimental evidence that Syt-1 and Munc13-2 indeed are PI(4,5)P_2_ effector proteins in exocytosis. While this may have been guessed by previous mutagenesis studies, we present crucial experimental evidence.

2) The priming- and PI(4,5)P_2_ -binding CAPS proteins are not involved in the PI(4,5)P_2_ -augmenting effect.

Although CAPS proteins are essential for vesicle priming and well-known PI(4,5)P_2_ interacting proteins, uncaging PI(4,5)P_2_ was still capable to augment exocytosis in the absence of both CAPS isoforms. This functionally distinguishes CAPS from Munc13-2 and Syt-1 and provides insight into differential molecular mechanisms: our data indicate that stoichiometric binding of PI(4,5)P_2_ to Syt-1 and Munc13-2 potentiates exocytosis by activating these proteins. However, CAPS proteins appear to function differently. Loss of CAPS reduces secretion (Liu et al., 2010) and our data now show that PI(4,5)P_2_ uncaging partially reverts this. This is consistent with a function of CAPS to sequester PI(4,5)P_2_ to the vesicle fusion site. Therefore uncaging PI(4,5)P_2_ below the vesicle – as we do here – can bypass the requirement for CAPS. This is the first distinction between two fundamentally different mechanisms (stoichiometric binding for protein activation vs. local PI(4,5)P_2_ recruitment) of lipid interactions regulating exocytosis (see also below).

3) Increasing PI(4,5)P_2_ triggers the rapid release of a part of the Readily Releasable Pool of vesicles.

This is the first time that a manipulation other than membrane depolarization or Ca^2+^ elevation was shown to trigger the release of secretory vesicles. These experiments establish that PI(4,5)P_2_ is more than a vesicle recruitment factor; it is actually involved in fusion itself. Moreover, our data establish that the PI(4,5)P_2_ induced triggering of exocytosis is very fast (milliseconds), demonstrating that PI(4,5)P_2_ can activate the vesicular fusion machinery on similar timescales as Ca^2+^, which is an important mechanistic insight into the vesicle fusion process.

In addition to these specific findings, we now discuss and illustrate (new Figure 8) what the general advantages of caged lipids are. Using uncaging it is possible to distinguish between two different roles of lipid-interacting molecules: those that require specific binding of the lipid to perform their function (‘stoichiometric role’), and those proteins that help localize the lipids locally, in order to align them with the downstream machinery (‘localization role’). Uncaging lipids with a near-homogeneous distribution in the membrane would overcome a localization defect, but not an obligatory, stoichiometric role. Thus, caged lipids can help distinguish between those two roles, as we illustrate here by the contrasting findings upon deletion of Munc13-2 and syt-1 on one hand, and CAPS^-1^/2 on the other.

Localization of caged-PI(4,5)P_2_

All reviewers commented on the localization of our cg-PI(4,5)P_2_. We would like to point out that it was our aim was to generate a caged, membrane permeable PI(4,5)P_2_ variant which would remain undetected in the cell prior to UV-uncaging. However, the fact that our compound is invisible to PI(4,5)P_2_ -recognizing proteins will affect its sub-cellular localization. Proteins which usually establish a strict pattern of phosphoinositide composition on distinct cellular organelles will simply not detect our cg-PI(4,5)P_2_. Therefore, the compound displays broad localization within the cell as we point out in the text. It is important to realize that it will not be possible to both ensure that the caged compound is physiological inert and that it has a similar localization as the physiologically active PI(4,5)P_2_ – those two requirements are irreconcilable. We have now pointed this out in the text. However, this difference can be used to an advantage, because as described before and as can also be seen in Figure 2, cellular PI(4,5)P_2_ typically localizes to clusters at the plasma membrane. In contrast, our compound appears more homogenous on the plasma membrane (Figure 2). Thus, UV uncaging will proportionally result in a much stronger (relative) increase of PI(4,5)P_2_ in areas surrounding endogenous PI(4,5)P_2_ clusters, allowing us to specifically investigate this. Despite the wide-spread cellular distribution of our compound, we are convinced that the elevation of plasma membrane PI(4,5)P_2_ levels is causative for the effects on exocytosis observed here for the following reasons:

1) Line profile analysis where the coumarin fluorescence was compared with a labelling of the plasma membrane demonstrated that cg-PI(4,5)P_2_ was present at the plasma membrane and liberated there (Figure 2).

2) Using TIRF microscopy we show that cg-PI(4,5)P_2_ is present at the plasma membrane and that its uncaging recruited PLC-δ1-PH-GFP, consistent with increased plasmalemmal PI(4,5)P_2_ (Figure 2).

3) Uncaging PI(4,5)P_2_ prevented the dissociation PLC-δ1-PH-GFP in cells where PLC activity was induced (Figure 2). Thus, PI(4,5)P_2_ uncaging counteracted PLC-mediated reduction of PI(4,5)P_2_ at the plasma membrane and stabilized PLC-δ1-PH-GFP in its initial location. Again, the opposite (stronger PLC-δ1-PH-GFP dissociation) would be expected if uncaging predominantly liberated PI(4,5)P_2_ on intracellular organelles.

4) PI(4,5)P_2_ uncaging at the footprint of HEK cells increased the local fluorescence of the actin-marker lifeact (new experiments in Figure 3), indicating the activation of actin polymerization and/or redistribution to the plasma membrane.

5) Bright field uncaging of PI(4,5)P_2_ in chromaffin cells showed that PLC-δ4-PH-GFP re-distributed to peripheral cell regions (Figure 3), in line with PI(4,5)P_2_ enrichment at the plasma membrane (if the uncaging of intracellular cg-PI(4,5)P_2_ had dominated, the opposite – a redistribution to the cell’s interior- would be expected). It might not have been clear that the data in Figure 3 show the ratio between fluorescence in the periphery and the center of the cell – thus an increase represents a redistribution towards the plasma membrane. At the suggestion of reviewer 2, we have now included an example difference picture showing the relocalization directly.

6) Exocytosis depends on a highly localized molecular machinery, particularly as we observe effects on primed RRP vesicles (Figure 7), which are docked via molecular contacts to the plasma membrane. Therefore, the only two membranes that affect exocytosis – within the time frame of our measurements – are the vesicular and plasma membranes.

7) We show that uncaging PI(4,5)P_2_ potentiates exocytosis via synaptotagmin-1 which was previously shown to interact with plasma membrane PI(4,5)P_2_ for fusion (Bai et al., 2004). If PI(4,5)P_2_ uncaging on the vesicle membrane would dominate, it would steer the synaptotagmin C2B to the wrong membrane, leading to an inhibition of exocytosis by PI(4,5)P_2_ uncaging, rather than a stimulation.

In our manuscript we now discuss the issue of cg-PI(4,5)P_2_ localization in more detail, have restructured our figure layout, performed additional analyses and experiments, and included a cartoon summarizing main findings (Figure 8).

Reviewer #1:[…] 1) Please comment on the effect of the pluronic in cell viability and secretion efficiency; also, the compounds show strange localizations in cells–please comment. Also, please comment on the absolute capacity of the chromaffin cells to undergo exocytosis after taking up the compound compared with control cells. Finally, it would help the reader if the authors could highlight at the end of the discussion precisely what this probe brings that was not really known before, and other important applications that will benefit from its use.

We have now pointed out that the overall capacity of the cells to undergo exocytosis in the presence of pluronic and caged PI(4,5)P_2_ is similar to previous experiments. We have also performed additional experiments in which we specifically investigate possible effects of the caged compound itself on exocytosis in chromaffin cells. These data are now provided in Figure 4—figure supplement 1 and revealed no effect of the compound (in the absence of uncaging).

There may have been a misunderstanding in regard to the treatment of the cells. In all experiments in which exocytosis was investigated both groups (uncaging vs. control) received identical treatment with pluronic and the caged PI(4,5)P_2_. All cells were loaded with the compounds, the only difference was whether they were exposed to UV light or not. Therefore, any effect of pluronic would be present in both groups. We have now included an additional statement in the text to clarify this issue.

We comment on the localization and the new findings made possible with the probe above and we have adjusted the text accordingly.

2. Figure 2—figure supplement 1 could be a regular figure, as could a few others; please consider making your paper 6-8 regular figures and keeping the synthesis details in supplemental

These data are now included in Figure 2.

Reviewer #2:[…] The impact of this study lies more in the development and use of a new tool than in a conceptual breakthrough. In fact, previous studies reached similar conclusions using patch pipette delivery of PI(4,5)P_2_. Thus, the improvement of temporal resolution is small for the experiments on IRP and RRP (both methods appear to have on the order of 1 min lead times). While uncaging might have the spatial resolution to achieve highly localized changes in PI(4,5)P_2_, the present work has the same limitation of previous work in which the released PI(4,5)P_2_is present throughout the cell (and may even cause mislocalization of proteins with PI(4,5)P_2_binding domains). In terms of the biology, two findings stand out: the lack of a role for CAPS1/2 and that PI(4,5)P_2_is capable of stimulating fusion without Ca^2+^ influx. Interestingly, the latter depended on a dramatically reduced time of PI(4,5)P_2_exposure (the first 1-2msec flash) so it highlights the power of the tool. Nevertheless, as the authors concede, the physiological situation is that exocytosis is triggered by a rise in calcium not a rise in PI(4,5)P_2_so the result is likely a consequence of something already present in the literature: that PI(4,5)P_2_may increase of Ca^2+^ affinity of syt-1. Reluctantly, my overall assessment is that the study is better suited to a more specialized journal despite its elegance and convincingness.

This reviewer points out that it might be expected that the augmentation of exocytosis by PI(4,5)P_2_ depends on Syt-1 and Munc13-2, based on previous mutagenesis studies of PI(4,5)P_2_ binding sites. However, the introduced mutations are bound to have additional effects. For instance, the residues of the Syt-1 C2B domain interacting with PI(4,5)P_2_ were also shown to interact with the neuronal SNARE complex (Zhou et al., 2015). Our genetic analysis of Syt-1 and Munc13-2 combined with PI(4,5)P_2_ uncaging indicates a functional interaction with PI(4,5)P_2_, which was not shown before. Because any introduction of new methodology requires experiments to validate it, it might appear that not much has been learned, or that existing methods would suffice. But this is clearly not the case here. In fact, our compound opens up new avenues for research: using PI(4,5)P_2_ uncaging it is now possible to distinguish between two different functions of PI(4,5)P_2_ -binding proteins: those activated by stoichiometric, obligatory, PI(4,5)P_2_ -binding (according to our data this will include synaptotagmin-1 and Munc13-2), and those operating to localize PI(4,5)P_2_ to the vesicle surroundings. Our experiments are consistent with CAPS proteins functioning in the latter fashion, because their requirement for vesicle priming (which depends on their PH domain) could be bypassed by PI(4,5)P_2_ uncaging, which uncovers PI(4,5)P_2_ beneath the vesicle. Thus, application of our tool enabled us to make this distinction, which is a major finding. Likewise, the striking observation that PI(4,5)P_2_ can actually trigger exocytosis could not have been obtained by any other method. As described in our discussion, we agree with reviewer #2 that this may be due to PI(4,5)P_2_ -dependent increase in the Ca^2+^ affinity of synaptotagmin, which was observed in biochemical experiments. However, the biochemical experiments give no account for the kinetics of PI(4,5)P_2_ -dependent activation. We here provide experimental data in living cells showing that this reaction operates on a comparable (millisecond) timescale as Ca^2+^ association, which provides novel biological insights into the vesicle fusion mechanism. We cannot see how it is an argument against our approach that exocytosis is normally triggered by a calcium rise; many methods have been devised to stimulate cellular processes in an artificial way so that new insights could be obtained.

We now provide a list of bullet points of biological advance presented in our study (see above and Discussion section). We have also included a cartoon depicting the differential mechanisms of the two types of lipid interactions (stoichimetric, localization-based), which can be distinguished using caged lipids (Figure 8).

Reviewer #3:[…] As far as I am concerned, the first introduction of a caged PPI_2_compund that actually works is a substantial achievement that deserves prominent publication. The authors did a good job in showing that the caged PPI_2_s work in principle. Clearly, a few open issues remain – e.g. (i) the question as to what the more or less homogeneous distribution of the caged compound in the cytosol after loading really means (Figure 2), (ii) how sure one can therefore be that it is actually plasma membrane PPI_2_that causes the effects seen, or (iii) what sub-steps of the secretion process are exactly regulated by the PPI_2_-Munc13-2 vs. PPI_2_-CAPS interaction. However, I am willing to cut the authors some slack here, because I am of the opinion that the new caged PPI_2_componds and the proof-of-principle data of the present paper will be very interesting to scientists in all areas of cell biology. In fact, the new compounds might be at least as interesting to cell biologists as they may be to people working on neurons and neurosecretory cells.

Regarding point (i): we now discuss the issue that the homogeneous localization of the caged PI(4,5)P_2_ is actually expected because the cage makes the compound physiologically inert, preventing the interaction with proteins that usually function to generate organelle-specific phosphoinositide compositions (see also general remarks and remarks to reviewer #2 above).

Regarding point (ii): it is true that owing to the cell-wide distribution it is likely that PI(4,5)P_2_ is also uncaged on intracellular organelles. In a different setting, this may be used to an advantage if combined with local UV illumination. Nevertheless, based on the 7 arguments we list above (general remarks), we think that the observed effects we observe on exocytosis are specific to changes in plasmalemmal PI(4,5)P_2_.

Regarding point (iii): we have now included a cartoon where we illustrate the two distinct functions of PI(4,5)P_2_ we envision for Munc13-2/syt-1 vs. CAPS (Figure 8). Given the requirement of Munc13-2 and Syt-1 for the augmenting effects of PI(4,5)P_2_ uncaging, our data are consistent with a function of CAPS upstream of these two factors to colocalize PI(4,5)P_2_ with the fusion machinery (see also general remarks above).